# Intracranial EEG signals disentangle multi-areal neural dynamics of vicarious pain perception

Huixin Tan[1,2,3], Xiaoyu Zeng [1,2,3], Jun Ni [1,2,3], Kun Liang[4], Cuiping Xu [5], Yanyang Zhang[6], Jiaxin Wang[1,2,3], Zizhou Li[1,2,3], Jiaxin Yang[1,2,3], Chunlei Han[4], Yuan Gao[4], Xinguang Yu[6], Shihui Han [7], Fangang Meng[4,8] ✉ & Yina Ma [1,2,3,8] ✉

Empathy enables understanding and sharing of others' feelings. Human neuroimaging studies have identified critical brain regions supporting empathy for pain, including the anterior insula (AI), anterior cingulate (ACC), amygdala, and inferior frontal gyrus (IFG). However, to date, the precise spatio-temporal profiles of empathic neural responses and inter-regional communications remain elusive. Here, using intracranial electroencephalography, we investigated electrophysiological signatures of vicarious pain perception. Others' pain perception induced early increases in high-gamma activity in IFG, beta power increases in ACC, but decreased beta power in AI and amygdala. Vicarious pain perception also altered the beta-band-coordinated coupling between ACC, AI, and amygdala, as well as increased modulation of IFG high-gamma amplitudes by beta phases of amygdala/AI/ACC. We identified a necessary combination of neural features for decoding vicarious pain perception. These spatio-temporally specific regional activities and inter-regional interactions within the empathy network suggest a neurodynamic model of human pain empathy.

Empathy enables us to quickly perceive and share the experiences and feelings of others, rendering it a powerful catalyst for successful social interactions and prosocial behavior[1–3]. This ability not only enhances our understanding of other individuals' affective states but also equips us with the foresight to predict their future actions, empowering us to take appropriate actions within specific social contexts[4]. Particularly in situations where we witness others' suffering, pain empathy grants us the capacity to vicariously experience their pain and motivates us to provide help[2,3].

Empathy involves dynamic interactions of multiple cognitive, affective, and motivational processes, such as perception and embodied sharing of others' experiences, one's own affective responses, and prosocial motives[5,6], which require neural oscillations and inter-regional communications to coordinate and integrate these diverse processes[7,8]. Therefore, it is crucial to assess the spatio-temporal dynamics of neural oscillations and inter-regional communication modes in empathy-related regions to understand how functionally diverse processes dynamically merge into empathic responses towards others' feelings. However, to date, (i) the temporal order and spectral patterns of neural activity in different empathy-related brain regions, (ii) the rapid functional interactions between different brain regions remains elusive. Moreover, (iii) it is also unclear whether and how vicarious pain perception can be decoded from multiple features of the neural

[1]State Key Laboratory of Cognitive Neuroscience and Learning Beijing Normal University, Beijing, China. [2]IDG/McGovern Institute for Brain Research, Beijing Normal University, Beijing, China. [3]Beijing Key Laboratory of Brain Imaging and Connectomics, Beijing Normal University, Beijing, China. [4]Beijing Tiantan Hospital, Capital Medical University, Beijing, China. [5]Department of Functional Neurosurgery, Xuanwu Hospital, Capital Medical University, Beijing, China. [6]Department of Neurosurgery, Chinese PLA General Hospital, Beijing, China. [7]School of Psychological and Cognitive Sciences, PKU-IDG/McGovern Institute for Brain Research, Peking University, Beijing, China. [8]Chinese Institute for Brain Research, Beijing, China. ✉e-mail: fgmeng@ccmu.edu.cn; yma@bnu.edu.cn

activity and inter-regional communications. The current study aimed to address these questions by recording intracranial electroencephalography (iEEG) signals during the perception of others' pain. This helps us understand how the brain gives rise to empathic responses to vicarious pain, reveal the dynamic organization of the empathy network, and construct a neurodynamic model of empathy.

Functional magnetic resonance imaging (fMRI) studies have identified a core pain empathy network, including the anterior insula (AI), anterior cingulate cortex (ACC), amygdala, and inferior frontal gyrus (IFG)[9,10]. fMRI findings have linked these regions to distinct processes involved in vicarious pain experiences[9–12]. Specifically, IFG activity is responsible for action observation and extracting the painful meaning from these actions, as well as mimicry of others' emotions[13–15]. ACC and AI activities support emotional contagion ("experiencing the emotions of another person") and formation of shared affective representations between oneself and the other experiencing pain[12,16,17]. The AI and amygdala are engaged in differentiating others' emotional states and generating one's own negative affective responses triggered by others' emotional states[18–20]. However, fMRI offers an indirect measure of neural activity with low temporal (seconds) resolution. While EEG and magnetoencephalography (MEG) studies have examined the temporal dynamics of empathic neural responses with a millisecond resolution[21–23], their coarse spatial resolution makes it difficult to precisely localize the source of observed activity, especially the deep brain structures[24].

iEEG measures local field potentials in cortical and deep brain structures with both millimeter and millisecond resolutions, offering the potential to detect rapid-timescale dynamics of neuronal population activity[25]. Several studies have investigated empathic processing using iEEG, providing electrophysiological evidence for the engagement of single neurons in the ACC[26] and broadband activity of insula[27] during the processing of others' pain. However, these studies focused exclusively on a single empathy-related region. To date, it remains elusive how the anatomically distributed and functionally distinct neural activity within the empathy network are temporally and spatially organized during perception of others' pain. The current study sought to reveal the region-specific spectral patterns and the temporal order of empathic neural responses by recording neuronal population activities in four key empathy-related nodes—the ACC, AI, amygdala, and IFG—in 22 epilepsy patients while they viewed pictures depicting painful stimulation applied to others' hands.

Moreover, inter-regional communications have been shown to be critical for empathic responses. Animal research has revealed a key role of projections from the ACC/AI to the basolateral amygdala in vicarious learning of others' emotional states (e.g., pain, fear)[28–30]. iEEG recording allows for the investigation of how neural oscillations support the spatial integration and rapid functional interactions between different brain regions[31–33]. A recent iEEG study[34] reported an association between a questionnaire score of empathy trait and network characteristics of the default-mode network during resting-state; however, this study only examined empathy-irrelevant, non-task-specific resting-state instead of task-related and empathy-specific neural activity. Therefore, it remains unclear how the IFG, ACC, AI, and amygdala communicate and coordinately orchestrate human empathy for others' suffering. Here, we assessed two potential inter-regional communication modes: low-frequency oscillatory coupling and phase-amplitude coupling. Low-frequency coupling has been shown to support long-range interactions across distant brain structures[31,35,36]. Phase-amplitude coupling, the modulation of the amplitude of fast high-frequency activity by the phase of low-frequency oscillations, is recognized as a flexible multi-frequency communication mode to integrate information across multiple spatiotemporal scales[37,38]. Thus we examined both low-frequency coupling and phase-amplitude

coupling to assess how the ACC, AI, amygdala, and IFG interacted when observing other's suffering.

Leveraging the strengths of iEEG and advanced analysis of neurophysiological signals, we provided a spectral-temporal-spatial map of neuronal population activity and inter-regional interaction dynamics that subserved empathic pain experience in humans. In an effort to delineate the specific contributions of these spectral-temporal-spatial specific patterns to vicarious pain perception and identify important neural features, we further investigated how these neural features jointly contributed to the perception of others' pain. Moreover, to assess the associations between empathic responses and critical neural features within the pain empathy network, we tested how these critical neural features were linked to empathy-related behavioral measures, including the strength of overall empathic responses and empathy-related subprocesses (i.e., evaluation of perceived pain intensity and one's own unpleasantness) during perception of other's pain. This investigation was potentially important for empathy research as it would help assess the importance of different types of neural features for vicarious pain perception and identify tailored targets for interventions of empathy-related deficits.

## Results
### Behavioral performance
Twenty-two pre-surgical epileptic patients viewed painful or non-painful pictures in which a person's hand was hurt or not hurt (Fig. 1a) and were instructed to understand and empathize with the target person's affective states during viewing the pictures. Following the picture viewing phase, patients were asked to judge whether the person depicted in the picture experienced pain. Similar stimuli and experimental design have been shown to capture pain empathy neural responses in previous studies[21,39,40]. Patients showed no significant differences in response accuracies (response accuracy difference: $5.91\% \pm 4.73\%$, $t_{21} = 1.26$, $p = 0.221$, Cohen's $d = 0.27$, 95% CI: −0.07, 0.27) and response times (RTs, RT difference: $-0.02 \pm 0.09$, $t_{21} = -0.16$, $p = 0.874$, Cohen's $d = -0.03$, 95% CI: −0.06, 0.05) between painful and non-painful conditions, suggesting comparable attentional engagement and motor responses to painful and non-painful stimuli.

After the iEEG recording, 16 patients (6 patients were unwilling to or failed to participate) completed a post-iEEG session to report subjective ratings of each stimulus regarding (i) the strength of their empathic responses (i.e., "How strongly do you feel empathic towards the target's pain?"), (ii) the intensity of perceived pain in others ("How much pain do you think the target experienced in this situation?"), and (iii) their own unpleasantness ("How unpleasant do you feel when viewing the stimulus") (Methods). All rating scores showed excellent between-rater reliabilities with ICCs > 0.90. Patients reported significantly stronger empathic responses (difference: $63.14 \pm 5.84$, $t_{15} = 10.82$, $p = 1.76 \times 10^{-8}$, Cohen's $d = 2.70$, 95% CI: 50.70, 75.58, Fig. 1b), perceived higher intensity of pain in the target (difference: $61.12 \pm 5.22$, $t_{15} = 11.71$, $p = 6.04 \times 10^{-9}$, Cohen's $d = 2.93$, 95% CI: 49.99, 72.24, Fig. 1c), and experienced more unpleasantness (difference: $51.12 \pm 7.33$, $t_{15} = 6.97$, $p = 4.49 \times 10^{-6}$, Cohen's $d = 1.74$, 95% CI: 35.49, 66.76, Fig. 1d) when seeing the painful (vs. non-painful) stimuli, providing evidence for empathic responses to other's pain in our patients.

Moreover, we assessed whether patients' behavioral responses to vicarious pain were similar to those of healthy individuals. We recruited an independent healthy sample ($n = 22$) whose age and gender were comparable to the patient sample (Methods) and asked them to complete the same pain judgment task and provide subjective ratings on experimental stimuli. Patients and healthy participants showed comparable behavioral performances in the pain judgment task and subjective ratings, indicated by insignificant interactions of pain (painful vs. non-painful) and group (patient vs. healthy) (all $p$ values for interaction >0.05, $F$-tests on linear mixed-effect models; Supplementary Table 1).

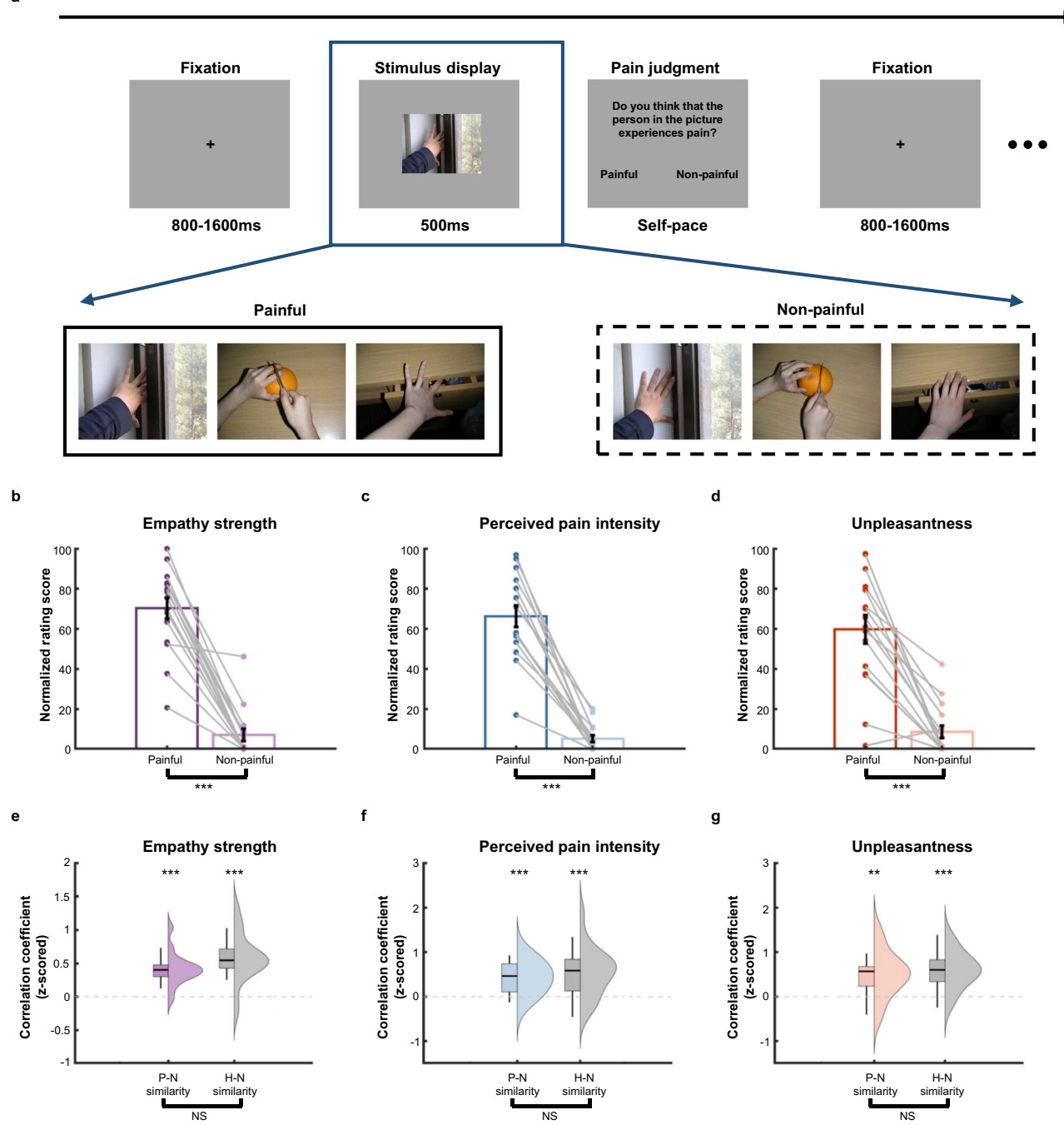

**Fig. 1 | Experimental procedure, stimuli, and behavioral results. a** Illustration of the timeline and stimuli of the pain judgment task. Each trial begins with a fixation of 1200 ms (varying from 800 to 1600 ms), followed by a 500-ms presentation of a painful or non-painful stimulus and a self-paced pain judgment of the stimulus. Each painful stimulus is matched with a neutral, non-painful stimulus sharing a similar context but implying no pain. **b−d** Empathy-related ratings of patients ($n = 16$). Paired-$t$ tests (two-sided) showed that patients reported stronger empathic responses (**b**, $p = 1.76 \times 10^{-8}$), perceived higher intensity of pain in the target (**c**, $p = 6.04 \times 10^{-9}$), and experienced more unpleasantness (**d**, $p = 4.49 \times 10^{-6}$) to painful (vs. non-painful) stimuli. Data are presented as mean values ± SE. Dots represent individual participants and gray lines show the conditional differences for individual participants. The correlations between the ratings of patients ($n = 16$) or healthy participants ($n = 22$) and the normative ratings for empathy strength (**e**),

perceived pain intensity (**f**), and unpleasantness (**g**). Permutation tests (two-sided) showed that the ratings of patients and healthy participants were similar to the normative ratings across all painful stimuli (for patient-normative (P-N) similarity, empathy strength: $p < 0.001$, perceived pain intensity: $p < 0.001$, unpleasantness: $p = 0.003$; for healthy-normative (H-N) similarity, empathy strength: $p < 0.001$, perceived pain intensity: $p < 0.001$, unpleasantness: $p < 0.001$). The P-N similarity was comparable to the H-N similarity (two-sided permutation test, empathy strength: $p = 0.287$, perceived pain intensity: $p = 0.576$, unpleasantness: $p = 0.583$). The split-half violin $p$lots show the probability density of the data and the left boxplots show the median and first and third quartiles with whiskers extended to the most extreme data points that are no more than 1.50 times the interquartile range. All comparisons were FDR-corrected across dimensions. **\*\***$p < 0.01$, **\*\*\***$p < 0.001$, NS, not significant. Source data are provided as a Source Data file.

Next, we examined the similarity in subjective ratings across stimuli between patient group and healthy group. This analysis showed that the aggregated subjective ratings of patient group were highly similar to those of healthy participant group (all $ps < 0.05$ regardless of all stimuli or only painful stimuli were considered, survived FDR correction for multiple comparisons, Pearson correlation; Supplementary Fig. 1, Supplementary Table 2). Moreover, we took individual variations into account to further confirm the similarity of empathy-related ratings between patients and healthy participants by calculating the similarity between each patient's or each healthy participant's ratings and normative ratings. Specifically, similar to previous studies[27,41], we considered the average rating from the healthy control sample as the normative ratings. We then computed correlations between each patient's or each healthy participant's ratings and the normative ratings (*Methods*). Results showed that both ratings from patient individuals and healthy individuals were similar to the normative ratings for each empathy-related measure (patient-normative similarity: all $ps < 0.01$, healthy-normative similarity: all $ps < 0.001$, survived FDR correction for multiple comparisons, no matter when all stimuli or only painful stimuli were considered; permutation tests; Supplementary Table 3). Importantly, the patient-normative similarity was comparable to the healthy-normative similarity in all three dimensions (all $ps > 0.05$ regardless of all stimuli or only painful stimuli were included; permutation tests on the difference between the patient-normative correlations and the healthy-normative correlations; Fig. 1e–g, Supplementary Table 3). Taken together, patient and healthy individuals showed similar response patterns in differentiating painful and non-painful stimuli, and also at a stimulus-wise level. Thus, patients' empathic responses to vicarious pain were comparable with those of healthy individuals.

## Empathy-related neural activity

We examined empathic neural responses in four empathy-related brain regions that were commonly covered by the implanted electrodes, including the AI (98 channels from 18 participants, Fig. 2a), ACC (40 channels from 7 participants, Fig. 2b), amygdala (68 channels from 17 participants, Fig. 2c), and IFG (91 channels from 15 participants, Fig. 2d). The locations of electrodes were exclusively determined by clinical need and the presence of specific channels within the AI, ACC, amygdala, and IFG was ascertained via careful examinations of channel locations in each patient's native anatomical space. To investigate temporal and spectral features of empathy-related neural activity, we performed a time-frequency decomposition of iEEG signals and compared spectral power in response to painful vs. non-painful stimuli for all frequency bands (theta band: 4–8 Hz; alpha band: 8–15 Hz; beta band: 15–35 Hz; gamma band: 35–70 Hz; high-gamma band: 70–150 Hz)[42].

The time-frequency analyses revealed significant differences in the low-frequency power in the AI, amygdala, and ACC between painful and non-painful conditions (two-sided *paired-t* tests for each time-frequency point, corrected $p < 0.01$, corrected for multiple comparisons using cluster-based permutation tests; Fig. 2e–g). Specifically, painful (vs. non-painful) stimuli induced a sustained suppression in power of a broad low-frequency range in the AI (theta/alpha/beta: 4–28 Hz, $p < 0.001$, Fig. 2e) and the alpha band in the ACC (6–18 Hz, $p = 0.006$, Fig. 2f), with power increases (decreases) in the non-painful (painful) conditions over the pre-stimulus baseline period (Supplementary Fig. 2a–d). Perception of others' pain was also associated with a late power suppression mainly in the beta band of the amygdala (16–45 Hz, $p = 0.006$, Fig. 2g; with a beta power increase in the non-painful condition (vs. baseline), which became absent in the painful condition, Supplementary Fig. 2e, f). In contrast, vicarious pain perception induced beta power enhancement in the ACC (19–40 Hz, $p = 0.009$, Fig. 2f), with beta power decrease (increase) in the non-painful (painful) condition compared to the baseline (Supplementary

Fig. 2g, h). Neural responses in the IFG to painful (vs. non-painful) stimuli were characterized by a sustained power increase mainly in the high-gamma band (55–145 Hz, $p < 0.001$, Fig. 2h; with power increases in both conditions (vs. baseline) but to a greater degree in painful condition, Supplementary Fig. 2i, j). These reported neural patterns showed high trial-by-trial consistencies (Supplementary Fig. 3).

We further examined the temporal characteristics of neural responses in each region. We averaged the power in significant frequency bands (e.g., beta band, high-gamma band) and compared it between painful and non-painful conditions for each time point at the post-stimulus time window. The onset time of empathic neural activity was defined as the earliest time point that showed a significant increase (or decrease) in the power to painful (vs. non-painful) stimuli (one-sided *paired-t* tests for each time point, corrected $p < 0.01$, corrected for multiple comparisons using cluster-based permutation tests, at least maintained for 50 ms consecutively to dampen noises[43,44]).

The empathy-induced beta power change emerged as early as 120 ms after stimulus onset in the AI (indexed by a significant decrease at a time window of 120 to 310 ms, $p = 0.005$, Fig. 2i), at 180 ms after stimulus onset in the ACC (a significant increase at 180–320 ms, $p = 0.003$, Fig. 2j; Supplementary Fig. 4 provides temporal profiles of theta/alpha power in the AI and alpha power in the ACC), and at 370 ms after stimulus onset in the amygdala (significant decreases at 370–470 ms, $p = 0.009$, Fig. 2k). These results indicated that early beta oscillations in the AI and ACC and a late beta power decrease in the amygdala were involved in the processing of others' pain. Interestingly, the increased high-gamma activity in the IFG in response to painful (vs. non-painful) stimuli was identified as early as 60 ms after stimulus onset (indexed by significant increases at 60-340 ms, $p < 0.001$, and 380–470 ms, $p = 0.005$, Fig. 2l), showing evidence for early and sustained involvement of high-gamma IFG activity in the processing of vicarious pain.

## Low-frequency synchronization between ACC, AI, and amygdala

Perception of others' pain was associated with low-frequency oscillations in the AI, ACC, and amygdala (Fig. 2e–g). These low-frequency oscillations provided a fundamental temporal scaffolding for inter-regional communication between these regions through low-frequency coupling, as low-frequency coupling was suggested to support long-range interactions across distant brain structures[31,35,36]. We thus examined the coupling of low-frequency oscillations among the AI, ACC, and amygdala. Power correlation was considered as a valid metric to estimate inter-regional low-frequency communication[45,46]. In particular, coupled oscillations of distinct neuronal groups operating at the same frequency can coordinate the rhythmic opening of their communication windows, thereby facilitating effective inter-regional communications[47]. Hence, similar to previous studies[32,45], we calculated power correlations in the overlapping frequency bands of the respective intra-regional power effects for each pair of regions (alpha and beta bands for ACC-AI; beta band for ACC-amygdala and AI-amygdala). For each channel pair and each frequency, power correlation was computed as the Fisher-z-transformed correlation coefficient across the stimulus-presentation window (i.e., 0–500 ms). We then compared the power correlation coefficients between painful and non-painful conditions (two-tailed paired *t*-tests for each frequency, corrected $p < 0.01$, using cluster-based permutation tests to correct for multiple comparisons).

The comparison between the painful and non-painful conditions revealed significant suppression in beta-band power correlations between the ACC and AI (25–32 Hz, $p = 0.007$, Fig. 3a; significant coupling in the non-painful condition, which became absent in the painful condition, Supplementary Fig. 5a, b), as well as between AI and the amygdala (18–24 Hz, $p = 0.001$, Fig. 3b; significant coupling in both conditions, Supplementary Fig. 5c, d). Interestingly, between the amygdala and ACC during the perception of other's pain (vs. non-

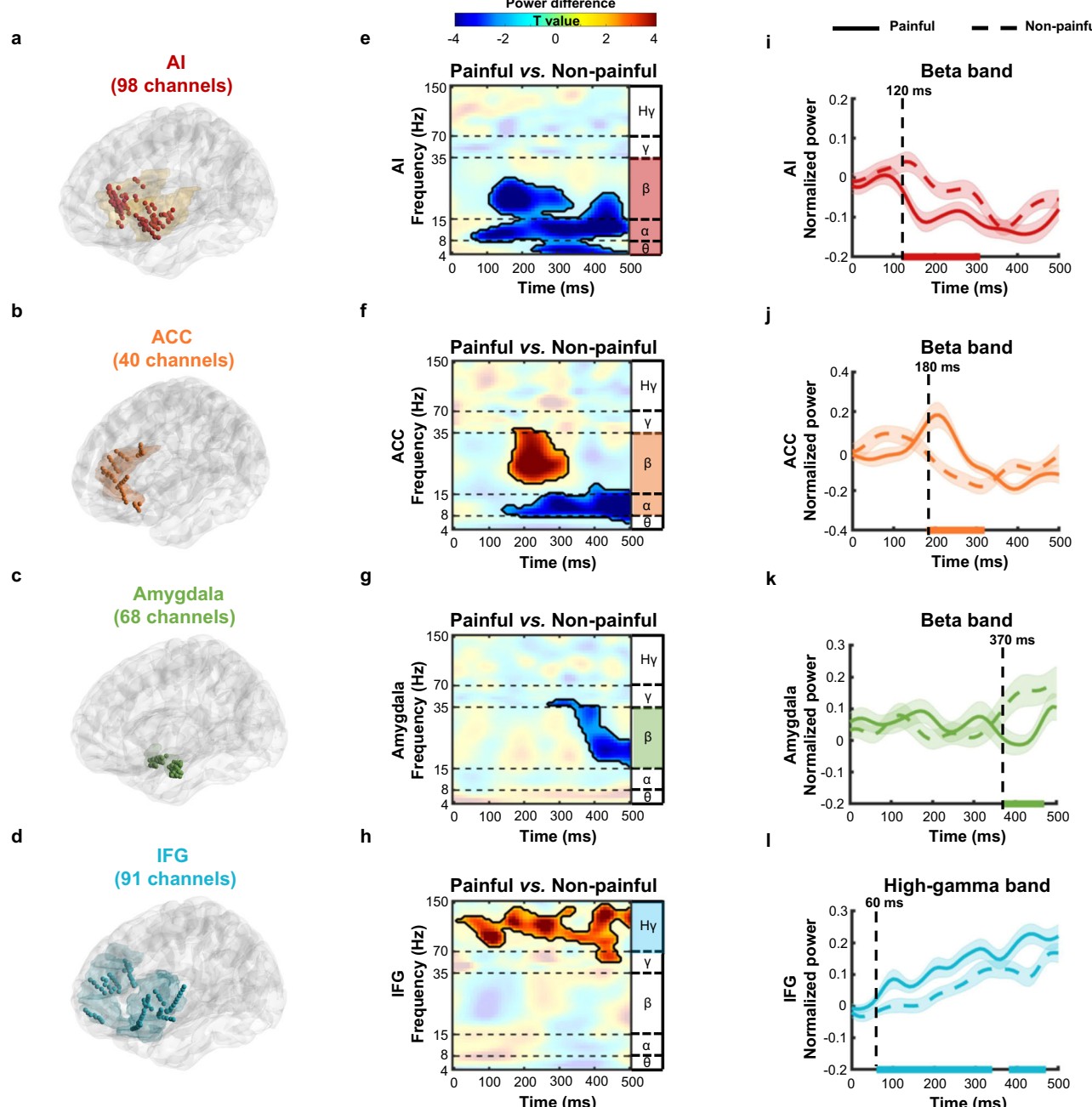

**Fig. 2 | Time-frequency profiles of the empathy-related neural responses.**
Locations of channels for the four brain regions of interest, i.e., the AI (red, **a**, $n = 98$ channels), ACC (orange, **b**, $n = 40$ channels), amygdala (green, **c**, $n = 68$ channels), and IFG (blue, **d**, $n = 91$ channels). Channel locations were determined by careful examinations of each patient's native anatomical space, and rendered onto a Colin27 template brain[89] using BrainNet Viewer[90] for visualization. Time-frequency spectrograms for the contrast of painful vs. non-painful stimuli with significant clusters in the AI (**e**, low-frequency cluster: $p < 0.001$), ACC (**f**, alpha cluster: $p = 0.006$; beta cluster: $p = 0.009$), amygdala (**g**, beta cluster: $p = 0.006$), and IFG (**h**, high-gamma cluster: $p < 0.001$) after the stimulus onset (two-sided *paired-t* tests for each time-frequency point, corrected for multiple comparisons using cluster-based permutation tests). Significant clusters are highlighted with black contours (corrected $p < 0.01$, 1000 permutations) with insignificant time-frequency ranges presented with transparency. Warmer colors indicate higher $t$ values. Horizontal dashed lines indicate boundaries between frequency bands and Hγ represents the

high-gamma band. Temporal characteristics of the AI, ACC, amygdala, and IFG neural activity. The beta or high-gamma power averaged across all channels within each brain region was plotted as a function of time for the AI (**i**, $p = 0.005$), ACC (**j**, $p = 0.003$), amygdala (**k**, $p = 0.009$), and IFG (**l**, early cluster: $p < 0.001$; late cluster: $p = 0.005$). Time points with significant conditional power differences are highlighted with horizontal lines on the x-axis (one-sided *paired-t* tests for each time point, corrected $p < 0.01$, corrected for multiple comparisons using cluster-based permutation tests, 1000 permutations, at least maintained for 50 ms consecutively). Dashed vertical lines indicate the onset timepoint of empathy-related power changes (i.e., a significant increase (or decrease) in the power to painful compared to non-painful conditions). Solid (painful condition) and dashed (non-painful condition) lines indicate the mean power across all channels for each time point, with shading representing the standard error. Source data are provided as a Source Data file.

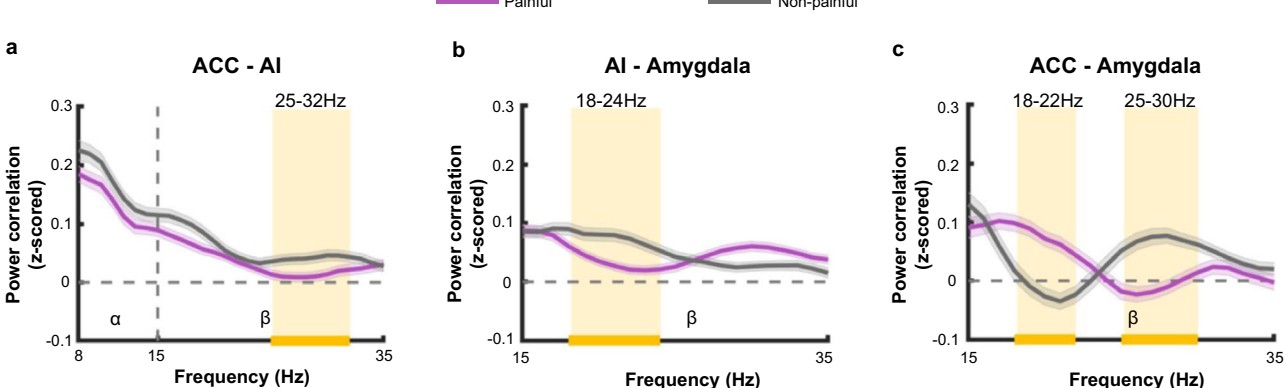

**Fig. 3 | Low-frequency synchronization between the ACC, AI, and amygdala.** Power correlations (z-scored) averaged across all channel pairs are plotted as a function of frequency for ACC-AI (**a**, $n = 234$ channel pairs, 25–32 Hz: $p = 0.007$), AI-amygdala (**b**, $n = 300$ channel pairs, 18–24 Hz: $p = 0.001$), and ACC-amygdala (**c**, $n = 72$ channel pairs, 18–22 Hz: $p = 0.005$, 25–30 Hz: $p = 0.001$) pairs. Purple (painful condition) and gray (non-painful condition) lines (shadows) indicate the mean (standard error) of frequency-resolved power correlations across all channel pairs. Clusters with significant conditional (painful vs. non-painful) differences in power correlations are highlighted with light-orange rectangles and orange horizontal lines on the axis (two-sided *paired-t* tests for each frequency, corrected $p < 0.01$, corrected for multiple comparisons using cluster-based permutation tests, 1000 permutations). Dashed vertical lines indicate boundaries between frequency bands. Source data are provided as a Source Data file.

pain), there were higher power correlations within the lower frequency range of the beta band (18–22 Hz, $p = 0.005$, Fig. 3c; significant ACC-amygdala coupling in the painful not non-painful conditions, Supplementary Fig. 5e, f) but lower power correlations within the upper frequency range of the beta band (25–30 Hz, $p = 0.001$, Fig. 3c; significant ACC-amygdala coupling in the non-painful, but not painful, condition; Supplementary Fig. 5g, h). These patterns were consistent across trials (Supplementary Fig. 6). These results together suggested that the beta oscillation may act as a prominent mediator for inter-regional communications among the ACC, AI, and amygdala during perception of other's pain.

We further disentangled the directionality of the observed empathy-related low-frequency synchronization by computing transfer entropy (TE), which measures the directionality of information flow between two brain regions[48]. For each channel pair, we computed bidirectional TE (e.g., from AI to amygdala and from the amygdala to AI) across the stimulus-presentation time window. We averaged TE values across frequency ranges of the corresponding power correlation effect and submitted them to repeated-measures analyses of variance (ANOVAs) with Pain (painful vs. non-painful stimuli) and Direction as within-subjects factors.

We detected a significant main effect of pain in ACC-AI at 25–32 Hz ($F_{1, 233} = 30.04$, $p = 1.10 \times 10^{-7}$, $\eta_p^2 = 0.11$, Supplementary Fig. 7a, b), suggesting reduced information flow at the beta band in both directions from-ACC-to-AI and from-AI-to-ACC. Interestingly, there was a significant Pain × Direction interaction on TE values in AI-amygdala at 18–24 Hz ($F_{1, 299} = 13.67$, $p = 2.59 \times 10^{-4}$, $\eta_p^2 = 0.04$, Supplementary Fig. 7c, d). Perception of painful stimuli specifically suppressed information transfer from the amygdala to AI (amygdala-to-AI: $t_{299} = -3.25$, $p = 0.001$, Cohen's $d = -0.19$, 95% CI: −0.009, −0.002; AI-to-amygdala: $t_{299} = -0.48$, $p = 0.629$, Cohen's $d = -0.03$, 95% CI: -0.004, 0.003; Supplementary Fig. 7c, d). For ACC-amygdala pairs, we found a significant main effect of pain at 18–22 Hz ($F_{1, 71} = 20.14$, $p = 2.72 \times 10^{-5}$, $\eta_p^2 = 0.22$, Supplementary Fig. 7e, f). For the interactions between the ACC and amygdala at 25–30 Hz, no significant results were found for the main effect of pain or the interaction effect (Main effect of pain: $F_{1, 71} = 0.33$, $p = 0.569$, $\eta_p^2 = 0.01$; Interaction: $F_{1, 71} = 1.83$, $p = 0.191$, $\eta_p^2 = 0.03$).

### Inter-regional phase-amplitude coupling between ACC/AI/amygdala and IFG

As we showed evidence for engagement of both AI/ACC/amygdala low-frequency oscillations and IFG high-gamma activity during the perception of vicarious pain, we further examined the relationship between low-frequency oscillations in the AI/ACC/amygdala and high-gamma activity in the IFG. Previous findings have suggested that the phase of low-frequency oscillations can modulate the amplitude of fast high-frequency activity to integrate neural responses across different spatial and temporal scales[37,38]. We thus tested whether the phase of low-frequency oscillations in the AI/ACC/amygdala modulated high-gamma amplitudes of the IFG during empathic responses by performing inter-regional phase-amplitude coupling (PAC) analysis. The phase and amplitude time series were obtained for frequencies in the frequency bands of the respective intra-regional power effect (phase frequency bands: alpha/beta bands in the ACC, theta/alpha/beta bands in the AI, and beta band in the amygdala; amplitude frequency: 70–150 Hz in the IFG). PAC values were indexed by circular-linear correlation coefficients between the phase time series and the amplitude time series. We then compared the Fisher-z-transformed PAC values across channel pairs between the painful and non-painful conditions (two-tailed paired $t$-tests for each frequency pair, corrected $p < 0.01$, using cluster-based permutation tests to correct for multiple comparisons).

This analysis revealed higher PAC values between the beta phase of the ACC/AI/amygdala and the high-gamma amplitude of the IFG in the painful than non-painful conditions. The IFG high-gamma amplitude was locked to beta phase in the ACC (phase at 16–29 Hz of ACC and IFG amplitude of 76–150 Hz, $p < 0.001$; Fig. 4a), AI (phase at 29–35 Hz of AI and IFG amplitude of 96–142 Hz, $p = 0.006$, Fig. 4b), and amygdala (phase at 22–35 Hz of amygdala and IFG amplitude of 70–116 Hz, $p < 0.001$; Fig. 4c) in response to painful (vs. non-painful) stimuli. Higher PAC value was also observed between the alpha phase of AI and the high-gamma amplitude of IFG in response to painful (vs. non-painful) stimuli (phase at 9–17 Hz of AI and IFG amplitude of 70–120 Hz, $p = 0.004$; Fig. 4b). Further analyses of all these spectral clusters showed that the reported PAC patterns were consistently observed across trials (Supplementary Fig. 8) and significant PAC was specifically present in the painful condition (Supplementary Fig. 9). These findings provided supporting evidence for modulations of the IFG high-gamma amplitude by ACC/AI/amygdala beta oscillations during empathy for pain.

In order to exclude the possibility that the observed empathy-related neural patterns were driven by inter-subject or inter-channel (inter-channel-pair) variations, we performed linear mixed-effect models with patient and channel (channel-pair in coupling analyses) identities as random effects[32]. These analyses replicated all of the

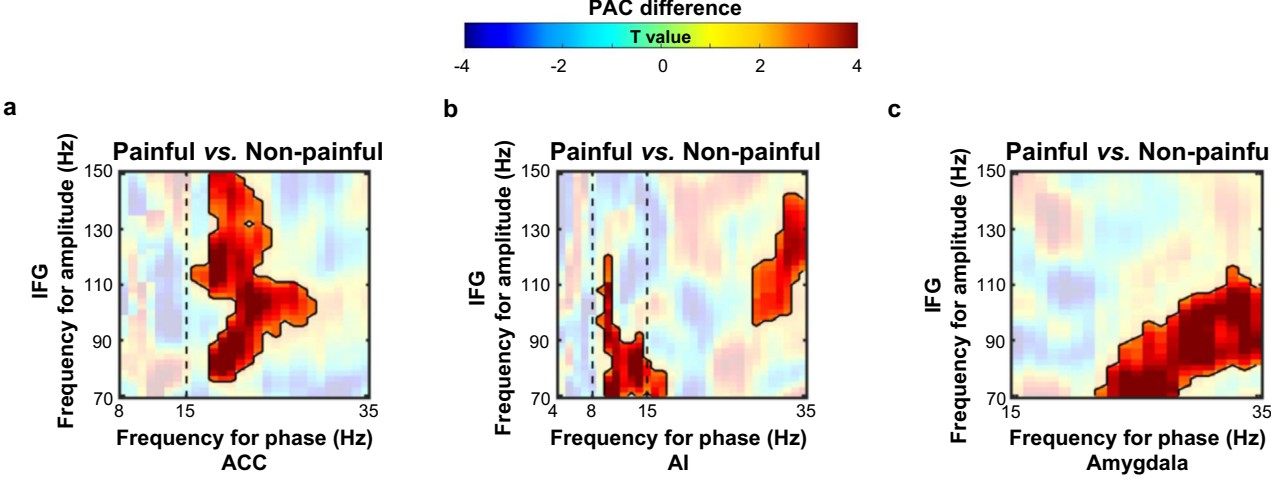

**Fig. 4 | Phase-amplitude coupling (PAC) between the ACC/AI/amygdala and IFG.** The inter-regional PAC comodulogram for differences between painful and non-painful conditions in the ACC-IFG (**a**, n = 219 channel pairs, beta cluster: p < 0.001), AI-IFG (**b**, n = 459 channel pairs, beta cluster: p = 0.006, alpha cluster: p = 0.004), and amygdala-IFG (**c**, n = 254 channel pairs, beta cluster: p < 0.001) pairs. Warmer colors denote higher t values. Significant clusters are highlighted with black contours (two-sided *paired-t* tests for each frequency pair, corrected p < 0.01, corrected for multiple comparisons using cluster-based permutation tests, 1000 permutations) with insignificant frequency ranges presented with transparency. Vertical dashed lines indicate boundaries between frequency bands. Source data are provided as a Source Data file.

observed effects in the spectro-temporal power (Supplementary Fig. 10), power correlations (Supplementary Fig. 11), and PAC (Supplementary Fig. 12).

## Contribution of neural activity and inter-regional communications in empathic pain perception

We tested how the neural activity and neural synchronization in the ACC, AI, amygdala, and IFG contributed to the detection of other's pain. To this end, we employed a support vector machine to construct a decoder to discriminate perception of painful and non-painful stimuli from power, power correlations, and PAC values. First, we split our dataset into two parts at the channel or channel-pair levels (detailed in *Methods*). One feature-selection dataset where we performed time-frequency decomposition of iEEG signals for each region and calculated inter-regional correlations of low-frequency power and PAC value for channel pairs, was used to select relevant features (see *Methods*). The other decoding dataset was used to construct and validate the classification model. As a validity check of our classification model before further analyses, we assessed its classification performance and found that our classification model significantly differentiated labels of painful and non-painful stimuli (classification accuracy = 76.28%, p < 0.001).

Next, we assessed the importance of the identified neural features to better understand how these neural features enabled the classification of perception of painful and non-painful stimuli. We calculated the classification weight for each feature to index its contribution to the classification[49,50]. The ACC beta power were assigned the highest classification weight, followed by ACC alpha power, phase-amplitude coupling between ACC and IFG, AI low-frequency power, and beta coupling between ACC and amygdala (the full list presented in Fig. 5a).

We then ran data simulations (referred to as "virtual disruption" analysis) to assess the necessity of each feature and/or combinations of features in detecting the perception of vicarious pain. Specifically, we performed additional decoding analyses to resemble how "*disruption*" of each feature or combination of neural features affected the vicarious pain perception. Similar to prior studies[51,52], we assessed the extent to which removing specific feature(s) from the model reduced decoding accuracy. The removal of feature(s) simulated the scenarios when a specific feature or feature combination was disrupted. The decrease in decoding accuracies thus provided simple estimations for

determining the necessity of the feature(s). We observed that removal of each individual feature from the classification model only resulted in slight decreases in decoding accuracy, with the model still significantly outperforming chance level (all FDR corrected ps < 0.01). This suggested that no single feature alone was sufficient to *disrupt* the decoding of vicarious pain perception. However, there was a dramatic drop in decoding accuracy when simultaneously removing all 8 top features (the decoding accuracy dropped to 49.78% and did not differ significantly from the chance level, p = 0.617; Fig. 5b, c), indicating that this eight-feature combination is necessary for discriminating between painful and non-painful stimuli. Moreover, the removal of this eight-feature combination led to significantly lower decoding accuracy compared to randomly removing an equal number of features (p < 0.001), further supporting the importance of this eight-feature combination in vicarious pain perception.

We concluded our analysis by examining whether these eight important neural features not only engaged in the detection of others' pain but also captured more fine-grained information about vicarious pain. We assessed the relationship between these neural features and subjective reports of empathic responses. Specifically, we tested for the associations between the conditional differences (painful minus non-painful) in neural features with differential ratings of empathy strength across all matched painful and non-painful pairs of stimuli using linear mixed-effect models which included patient and channel (or channel pair) identities as random effects to control for individual variations (among 16 patients who completed the post-iEEG session). We found that the strength of empathic responses were negatively associated with ACC alpha power and AI low-frequency power (ACC alpha power: $\beta = -0.15$, SE = 0.05, $t_{343} = -2.69$, p = 0.008, 95% CI: −0.25, −0.04; AI low-frequency power: $\beta = -0.16$, SE = 0.04, $t_{658} = -3.55$, p = $4.05 \times 10^{-4}$, 95% CI: −0.24, −0.07; survived FDR correction for multiple comparisons; Fig. 5d; similar analyses on the remaining six neural features failed to find significant relationship with empathy strength, all FDR-ps > 0.05). This suggested that stronger empathic responses were linked to greater suppression of ACC alpha oscillations and AI low-frequency oscillations. These findings further suggested that suppressed ACC alpha power and AI low-frequency power not only facilitated qualitative differentiation between others' pain and non-pain (signaling the presence of other's pain) but also quantitatively tracked the strength of empathic responses.

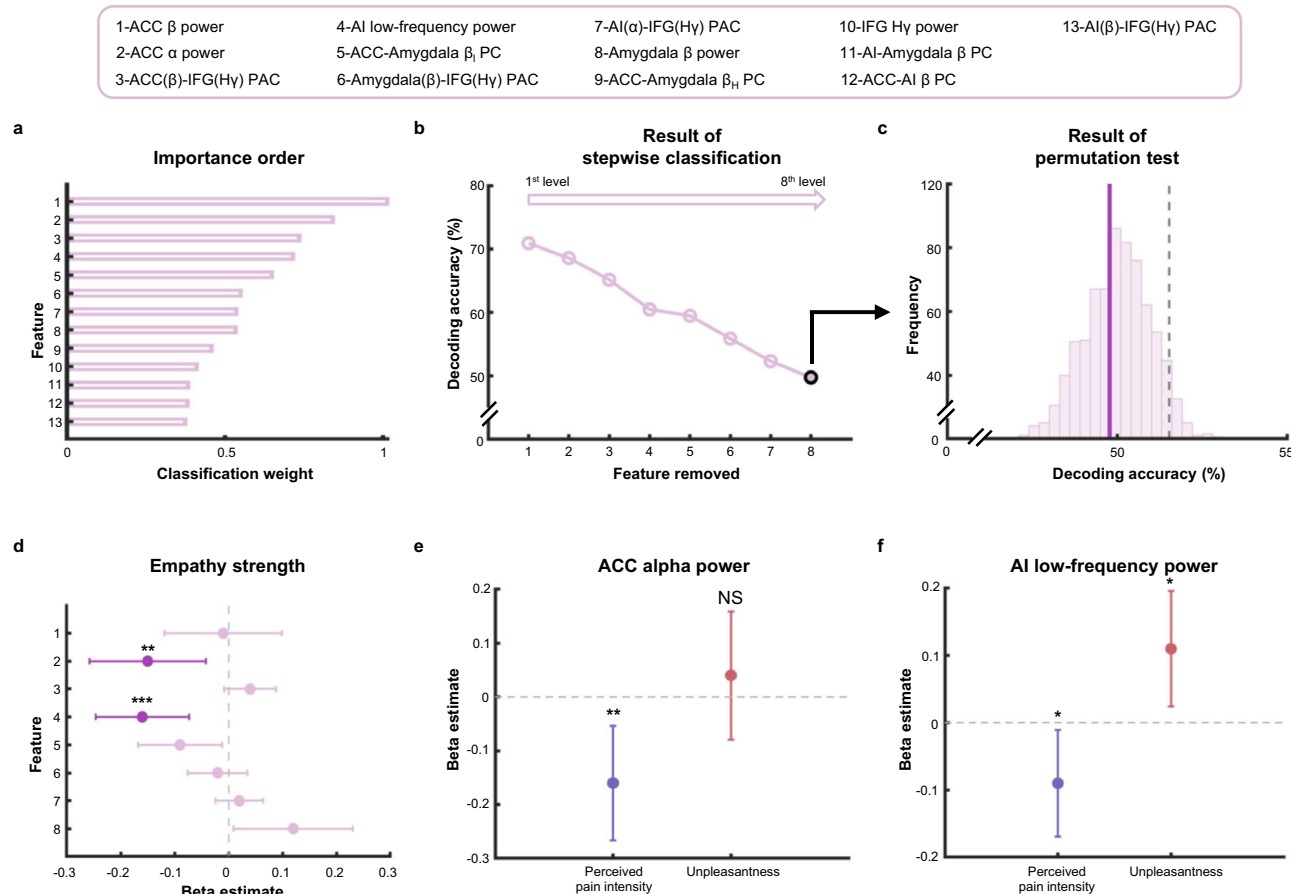

**Fig. 5 | Contribution of regional activity and inter-regional communications in vicarious pain perception. a** The classification weights of each empathy-selective feature. α alpha, β beta, Hγ high-gamma, PC power correlation, PAC phase-amplitude coupling, $\beta_{l/h}$ low (high) frequency range within the beta band. **b** Decoding accuracy for each level of stepwise classification where we sequentially removed features based on their classification weights in (**a**) until the classifier failed to decode vicarious pain perception (highlighted by the black-edged circle). **c** The decoding accuracy after removing the top 8 features (marked by a dark-purple vertical line) located lower than the upper limit of the 95% confidence interval (marked by gray dotted line) of the permutation distribution (1000 permutations, one-sided). **d** The association between empathy strength ratings and the top 8 features (linear mixed-effect model, two-sided, FDR-corrected across features). The beta estimates of ACC alpha power ($n = 345$ trials from 36 channels, $p = 0.008$) and AI low-frequency power ($n = 660$ trials from 70 channels, $p = 4.05 \times 10^{-4}$) survive FDR-correction (shown in dark-purple). The beta estimates of ACC beta power ($n = 345$ trials from 36 channels, $p = 0.820$), ACC-IFG PAC

($n = 1839$ trials from 199 channels, $p = 0.133$), ACC-amygdala beta coupling ($n = 642$ trials from 72 channels, $p = 0.031$), amygdala-IFG PAC ($n = 1735$ trials from 194 channels, $p = 0.537$), AI (alpha phase) -IFG PAC ($n = 2831$ trials from 307 channels, $p = 0.491$) and amygdala beta power ($n = 432$ trials from 46 channels, $p = 0.035$) do not survive FDR-correction (shown in light-purple). The association between subjective reports of perceived pain intensity (own unpleasantness) and ACC alpha power (**e**) and AI low-frequency power (**f**) (linear mixed-effect model, two-sided, FDR-corrected across dimensions). The ACC alpha power ($n = 345$ trials from 36 channels) was associated with perceived pain intensity ($p = 0.003$) but not with unpleasantness ($p = 0.503$). The AI low-frequency power ($n = 660$ trials from 70 channels) was associated with both perceived pain intensity ($p = 0.026$) and unpleasantness ($p = 0.012$). Data in (**d**–**f**) are expressed as $\beta$-estimate ± 95% CI, which are presented as the center circle and corresponding error bars, respectively. *$p < 0.05$, **$p < 0.01$, ***$p < 0.001$, NS not significant. Source data are provided as a Source Data file.

To aid the interpretation of the functional meaning of these associations, we further examined whether and how the neural features encoding the strength of empathic responses were associated with the intensity of perceived pain in others and one's own unpleasantness. We found that ACC alpha power was specifically associated with the intensity of perceived pain ($\beta = -0.16$, SE = 0.05, $t_{343} = -3.01$, $p = 0.003$, 95% CI: −0.27, −0.06; survived FDR correction for multiple comparisons; Fig. 5e) but not with the level of experienced unpleasantness ($\beta = 0.04$, SE = 0.06, $t_{343} = 0.67$, $p = 0.503$, 95% CI: −0.08, 0.16; Fig. 5e), suggesting that higher suppression of ACC alpha power predicted perception of stronger pain in others. AI low-frequency power was positively associated with the level of experienced unpleasantness ($\beta = 0.11$, SE = 0.04, $t_{658} = 2.51$, $p = 0.012$, 95% CI: 0.02, 0.20; survived FDR correction for multiple comparisons; Fig. 5f) but negatively linked to perceived pain intensity ($\beta = -0.09$, SE = 0.04, $t_{658} = -2.23$, $p = 0.026$, 95% CI: −0.17,

−0.01; survived FDR correction for multiple comparisons; Fig. 5f). These results showed that stronger suppression of AI low-frequency power was related to higher intensity of perceived pain in others but a lower level of self-related unpleasantness.

It should be noted that we conducted additional control analyses to exclude the possibility that our neural findings resulted from potential differences in arousal levels between painful and non-painful stimuli. We asked patients to provide ratings of the arousal level for each stimulus after the iEEG recording. We examined the association between the arousal level and ACC alpha power/AI low-frequency power but did not find any significant results ($ps > 0.05$; no significant results even when we checked for spectro-temporal power at all time-frequency points in each brain region, Supplementary Fig. 13). Therefore, the observed neural effects cannot be attributed to potential differences in arousal levels between painful and non-painful stimuli.

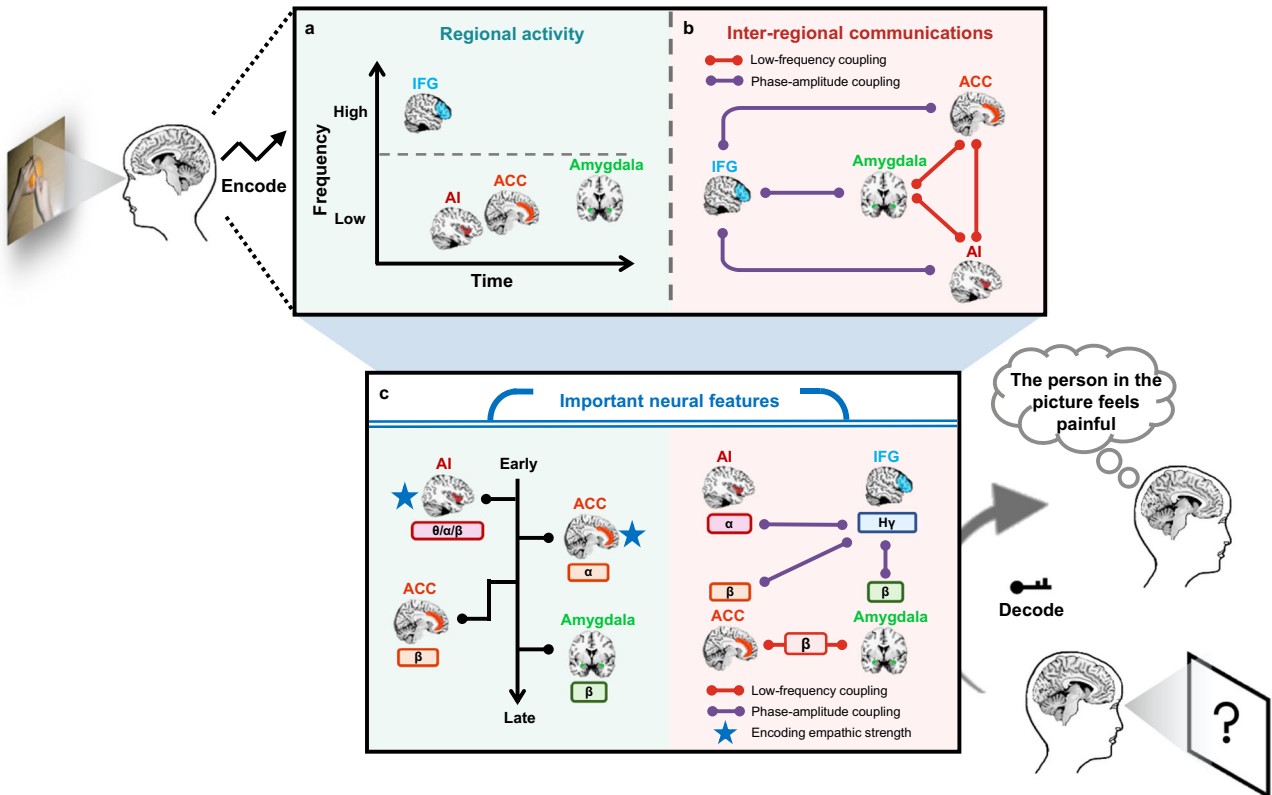

**Fig. 6 | A neurodynamic model of empathy for pain constructed based on our iEEG findings.** Illustration of the empathy-specific spectral-temporal map of the IFG, AI, ACC, and amygdala (**a**) and their inter-regional communications (**b**). **a** Perceived pain in others is encoded by early high-gamma activity in the IFG, early low-frequency activity in the AI and ACC, and late beta power in the amygdala. **b** Two types of inter-regional communications subserve empathic responses to painful stimuli, specified by low-frequency coupling among the ACC, AI, and amygdala and cross-frequency coupling between IFG high-gamma amplitudes and AI/amygdala/ACC low-frequency phases. **c** Illustration of a necessary neural feature combination to decode the perception of vicarious pain, including the low-frequency activity of ACC/AI/amygdala, ACC-amygdala beta coupling, and IFG-amygdala/AI/ACC cross-frequency coupling. The nuanced variations of strength of individuals' empathic responses were encoded by low-frequency activity of ACC (alpha) and AI. θ/α/β/Hγ, theta/alpha/beta/high-gamma. The locations of ROIs were drawn based on the Colin27 template brain[89] using the MRIcron software (http://www.mricro.com).

## Discussion

Empathy is a complex psychological construct arising from dynamic interactions between multiple processes[5,6]. Our iEEG results revealed the precise spectro-temporal characteristics of neural responses in the key regions of the empathy network (i.e., ACC, AI, amygdala, and IFG) and the electrophysiological basis of the inter-regional communications among these regions during perception of others' pain. The oscillatory patterns and inter-regional functional interactions provide insights into our understanding of how different processes dynamically coordinate and merge into empathy for pain.

A neurodynamic model of human empathy for pain emerged from our iEEG findings (Fig. 6). This model disentangles the responding frequency, timing, and functional significance of neural responses in the four key nodes of the empathy network and highlights two interregional communication modes of these nodes. Specifically, perception of others' pain elicited early high-gamma activity in the IFG followed by distinct patterns of low-frequency oscillatory alternations in the AI and ACC and later in the amygdala (Fig. 6a). This model also highlights distinct mechanisms underlying inter-regional information transfer and integration in the empathy network, including the beta coupling between the ACC, AI, and amygdala and increased modulations of IFG high-gamma amplitudes by phases of amygdala/AI/ACC beta oscillations (Fig. 6b). Importantly, our study integrated regionspecific neural activity and inter-regional interactions to decode vicarious pain perception, enabling us to characterize how these neural features jointly contributed to vicarious pain perception. Specifically, the low-frequency power of ACC/AI/amygdala, ACC-amygdala

beta coupling, and IFG-amygdala/AI/ACC cross-frequency coupling consist of a necessary combination to decode the perception of vicarious pain (Fig. 6c). Moreover, ACC alpha and AI low-frequency oscillations also captured more nuanced empathic-pain-related information, such as variations in the strength of individuals' empathic responses.

Co-activations of the ACC and AI have often been observed in previous fMRI studies of empathy for pain[9,16,17]. What remains unclear is whether the neural dynamics of the ACC and AI contribute to different aspects of empathic experiences. Here, we identified spectrotemporally shared and distinct profiles of ACC and AI oscillations in response to perceived pain in others. On the one hand, suppression of alpha oscillations was common for the ACC and AI when differentiating between painful and non-painful stimuli. The alpha suppression observed here resembles the observation that first-hand experienced pain was associated with a reduction in alpha power[8,53]. This finding is consistent with the shared network hypothesis of empathy[16,17] that individuals empathize with others' pain by recruiting similar neural substrates underpinning first-hand pain experience. Our results further specified alpha suppression in the ACC and AI as the shared oscillatory feature of first-hand and vicarious experiences of physical pain. On the other hand, we evidenced region-specific oscillatory patterns to others' pain in the ACC and AI. The ACC response to vicarious pain perception was featured with enhanced beta oscillations, increased low-beta coupling with the amygdala, and represented only the intensity of perceived pain in others. However, a distinct pattern was observed in the AI with decreased beta oscillations,

decreased low-beta coupling with the amygdala, and predictive of both the intensity of other's pain (similarly reported by Soyman and colleagues[27]) and participants' own unpleasantness. These findings were consistent with recent animal studies that revealed distinct functional roles of the ACC and AI in the early formation and consolidation of empathic pain by in vivo electrophysiological recording of the ACC and AI in mice[30].

While animal electrophysiological studies have documented the important role of the amygdala in empathic responses[28–30], the fMRI evidence for the involvement of the amygdala in human empathy has been inconsistent[9,54,55], leaving whether the amygdala is crucial for the processing of human empathy for pain an open question. Our iEEG findings fill the gaps between animal electrophysiological findings and human fMRI studies by providing electrophysiological evidence for the crucial and sophisticated role of amygdala in the perception of vicarious pain. Interestingly, we showed that the amygdala activity responded to others' pain later than the ACC, AI, and IFG. The late amygdala oscillation observed in the current study was less expected, but was consistent with findings of animal research showing that amygdala neurons responded to cues associated with electric shocks to another mouse later than neurons in the ACC[28]. The late amygdala beta suppression and inter-regional communications with other regions likely reflected the processing of late-stage information by the amygdala, such as the integrated neural representation of others' pain resulting from interactions with other regions. This processing may aid in differentiating from others' emotional states and generating one's own negative emotional responses[56,57]. It would be valuable for future research to directly test this possibility.

The discovery of empathy-related inter-regional communications has advanced our mechanistic understanding of the functional organization of the empathy network. We identified two potential pathways for inter-regional communication: beta-band-coordinated coupling between ACC, AI, and amygdala; and cross-frequency coupling between high-gamma IFG and beta ACC/AI/amygdala. Previous animal studies have identified critical functional roles of the AI-amygdala[30] and ACC-amygdala[28,29] circuits in observational learning and the formation of empathic pain. However, to date, it remains unclear how rapid communications between empathy-relevant brain regions support empathic responses in humans. Our iEEG results provided electrophysiological evidence for the engagement of these two circuits in human empathy, suggesting the ACC/AI-amygdala circuit as an evolutionarily conserved mechanism of empathy. Moreover, we identified a different mode of inter-regional communication related to empathy − cross-frequency coupling between high-gamma IFG and beta ACC/AI/amygdala − which points towards new directions for future investigations into empathy-related circuits.

The current study provided evidence for both increased functional interactions (e.g., enhanced coupling between ACC beta phase and IFG gamma amplitude) and decreased inter-regional communications (e.g., attenuated beta coupling between ACC and AI) within the empathy network. These patterns highlighted rapid information flow among different brain regions to coordinate diverse processes of empathy. When processing others' pain, the brain needs to not only enhance functional interactions between specific empathy-related regions (e.g., between ACC and IFG), potentially facilitating their coordination and information integration but also appropriately suppress certain inter-regional communications in order to reduce mutual distractions and increase functional specialization of relevant brain regions (e.g., ACC and AI).

Taking the ACC and IFG as an illustrative example of increased functional interactions. In terms of regional oscillations, while low-frequency oscillations in the ACC responded to perceived pain, we observed a distinct spectro-temporal profile of the IFG with early high-gamma activity. The modulation of low-frequency phases on high-frequency amplitudes is suggested to convert information from slow timescales into fast local processing and integrates functions across spatio-temporal scales[37,38]. Thus, the enhanced modulation of IFG high-gamma amplitude by the ACC beta phase suggests a possible mechanism underlying information integration from ACC and IFG, echoing previous findings on anatomical connections between ACC and IFG[58,59].

In contrast, we observed weaker functional communications between ACC and AI in the painful (vs. non-painful) condition, which may be associated with a reduced exchange of information related to perceiving others' emotional states and generating personal emotional responses. This may facilitate functional specialization and prevent emotional responses from biasing the evaluation of other's pain[60]. These findings on empathy-related inter-regional communication aid in understanding how the brain generates empathic responses towards others' pain.

The behavioral and neural data in the current work were collected from epilepsy patients, however, the disease or treatments might impact our findings to a minimum extent. Patients in our study indeed perceived others in stronger pain, felt more unpleasantness, and reported stronger empathic responses when seeing others in painful (relative to non-painful) situations. Second, patients and healthy controls showed comparable behavioral performances during judgments of others' pain and subjective ratings of each stimulus. Third, patients' neural responses were predictive of the strength of empathic responses, as well as their own ratings of perceived pain intensity and unpleasant feelings of vicarious pain. These results provide consistent evidence for the engagement of empathic response to painful stimuli in patients and the behavioral and neural patterns observed here reflect response profiles of perceived pain in others.

In the current study, the localization of the electrodes was determined exclusively by clinical needs, which limited further investigations into other empathy-related regions (e.g., sensorimotor cortex or middle cingulate cortex) and the functional dissociation of subregions within each brain region. We conducted a preliminary examination of potential hemispheric differences in empathy-related neural activity. While we found similar patterns in the left and right hemispheres of ACC, amygdala, and IFG, there was a right-lateralized processing of vicarious pain in the AI (Supplementary Fig. 14), consistent with previous studies[30,61]. In addition, the encoding of vicarious pain may also vary along the anterior-posterior axis within the insula. This speculation arose from comparing our findings with those reported by ref. 27. Our work focused on the anterior insula and revealed significant power changes specifically within low-frequency bands in AI (but not in the gamma and high-gamma bands). In contrast, ref. 27 explored activity across both the anterior and posterior insula and found that broadband activity, rather than specific frequency bands, was involved in encoding intensity of perceived pain in others. These distinct spectral patterns may imply that the anterior and posterior insula have different response profiles during processing of other's pain. Alternatively, these differences may be attributed to variations in experimental design and stimulus types between our study and that of ref. 27. For example, while our study asked patients to make a dichotomous judgment on whether the person depicted in a static picture experienced pain or not, ref. 27 asked patients to rate the intensity of pain perceived in a person shown in a video clip using continuous evaluation. It would be interesting for future studies with larger sample size and trial number to further investigate if and how the experimental task modulates spectral patterns within different subregions of the insula as well as other empathy-related regions during processing others' pain.

To conclude, our iEEG results revealed frequency-specific patterns in the key nodes of the empathy network and identified two inter-regional communication modes as crucial neurophysiological mechanisms underlying empathy for pain. Moreover, our findings of the beta-band-coordinated synchronization between the AI, ACC, and

amygdala, as well as the cross-frequency coupling between the AI/ACC/amygdala (low-frequency) and IFG (high-gamma), provide insights into how functionally diverse and spatially distributed brain regions communicate and coordinate when perceiving others in pain. These findings contribute to a sophisticated understanding of the neural dynamics of empathy, which may help with the prediction of empathy-motivated prosocial behaviors and the development of therapeutic interventions targeting empathy-related deficits commonly observed across various neuropsychiatric disorders.

## Methods

### Inclusion and ethics

For all patients, electrode localizations were exclusively determined by clinical needs. We prioritized and maintained the integrity of clinical care during conducting the current study. All patients provided informed consent after the experimental procedure had been fully explained, and were acknowledged their right to withdraw at any time during the study. The experimental design and procedures adhered to the standards set by the Declaration of Helsinki and were approved by the local Institutional Review Board of each hospital where the patients were tested (i.e., the Chinese PLA General Hospital: S2021-394-02, Beijing Xuanwu Hospital: ClinRes No.2022018, and Beijing Tiantan Hospital: KY 2020-080-02).

### Patients

Data were recorded from 29 epilepsy patients who were implanted with intracranial depth electrodes and were undergoing intracranial EEG monitoring to localize the seizure onset zone for potential surgical resection. All participants recruited in the current study had no history of psychiatric disorders, head trauma, or encephalitis. Patients did not take pain medication several hours prior to the iEEG recording of the pain judgment task and were not experiencing any physical pain during the iEEG recording. The patient selection was based on two inclusion criteria: (i) having electrodes in the ACC, AI, amygdala, or IFG contralateral to or outside of the epileptogenic zone; and (ii) achieving a response accuracy above 50% in the pain judgment task. Based on these criteria, one patient was excluded due to a low response accuracy (45%) in the pain judgment task, and six patients were excluded because no electrodes were implanted in the regions of interest. The remaining 22 patients were included in the behavioral and neural analysis of the pain judgment task (13 males, age = 25.73 ± 2.07 years old).

Similar to most iEEG studies[31–33], we did not perform a prior sample size estimation. Data from neurosurgical patients in this study were collected over 4 years and our sample sizes are similar to (or larger than) those reported in most of previous iEEG publications[27,31,35]. Moreover, the main analyses were conducted in regions of interest at the channel level (40–98 channels per region) or region pairs of interest at the channel-pair level (72–459 channel pairs), which are similar to or larger than that reported in typical iEEG publications[27,31–33,35]. A post-hoc power analysis (two-sided, paired-$t$ tests, alpha error = 5%) confirmed that we had sufficient power (86.94%) to detect medium effect sizes ($d = 0.5$) even with the minimum number of channels ($n = 40$). We employed stringent thresholds and the main findings are highly significant statistically, and survived correction for multiple comparisons.

### Experimental stimuli

Experimental stimuli consisted of twenty pictures showing others in painful or neutral (non-painful) situations. This set of stimuli was employed by previous studies to successfully induce empathic neural responses[21,62]. The painful pictures described first-person perspective situations in which someone's hand was hurt (e.g., a hand was cut by a knife or caught by a door). Each painful picture was matched with a non-painful picture sharing a similar context but implying no pain (Fig. 1a). The luminance, contrast, and color of the painful and non-painful stimuli were matched. Each stimulus was presented on a gray background of a 21.5-inch color monitor during iEEG recording, subtending a visual angle of 11.33° × 8.51° at a viewing distance of 80 cm.

### Experimental tasks and procedures

During iEEG recording, epileptic patients performed a pain judgment task[21,62] (Fig. 1a). Each trial started with a fixation presenting for 1200 ms (with a duration ranging from 800 to 1600 ms)[21]. Participants were then presented with a painful or non-painful picture with a duration of 500 ms and were instructed to understand and empathize with the emotional states of the person depicted in the pictures. To motivate and monitor engagement in the task, participants were asked to indicate whether the person in each picture experienced pain or not (as specified by Chinese instruction: "请您判断图片中的人是否感到疼痛") by pressing the left (index finger) or right (middle finger) button using their dominant hands after the picture disappeared. Participants made these responses on a self-paced basis and were encouraged to respond as accurately as possible. Here, the picture viewing phase and response phase were separated in order to isolate vicarious pain perception from any potential biasing effects of motor responses on neural responses to others' pain. For each participant, painful and non-painful pictures were presented once in a random order.

Similar to the majority of previous neuroimaging studies[21,63,64], we invited all patients to a post-iEEG session to measure the empathic strength and other empathy-related subjective ratings to perceived pain in others after the iEEG recording. This setting (post-iEEG rating procedure) could avoid potential influence on the empathic neural responses in the pain judgment task caused by self-report empathic ratings (e.g., avoid evoking intentionally controlled empathic processes[65]), and enabled us to separately measure different dimensions of empathy-related ratings (see the procedure of post-iEEG session in Supplementary Fig. 15). No data were excluded, but the subjective ratings of six patients were missing as the six patients were unwilling to or failed to complete the post-iEEG session. Thus, the behavioral and neural analysis of subjective ratings were conducted on the remaining 16 patients (10 males, age = 24.63 ± 2.35 years old). In particular, we asked them to evaluate each stimulus along three dimensions: (i) the strength of their empathic responses to each stimulus ("How strongly do you feel empathic towards the target's pain?"; 0 = not at all, 100 = extremely strong); (ii) the intensity of perceived pain in others ("How much pain do you think the target experienced in this situation?"; 0 = not at all, 100 = extremely painful); and (iii) their own unpleasantness ("How unpleasant do you feel when viewing the stimulus"; 0 = not at all, 100 = extremely unpleasant). Ratings for these dimensions were obtained on separate blocks. In addition, to confirm that the neural findings did not result from possible differences in arousal levels between painful and non-painful stimuli, patients also provided ratings of arousal level for each stimulus ("How intense is your emotional response induced by this picture"; 0 = extremely calm, 100 = extremely strong). Although painful stimuli were associated with higher arousal levels than non-painful stimuli (difference: 55.98 ± 6.00, $t_{15} = 9.33$, $p = 1.23 \times 10^{-7}$, Cohen's $d = 2.33$, 95% $CI$: 43.20, 68.76), the observed tempo-spectral patterns did not vary as a function of arousal levels (Supplementary Fig. 13). These results suggested that the observed neural effects were not due to potential differences in arousal levels between painful and non-painful stimuli.

### Control participants

To demonstrate that patients were not impaired in their responses to vicarious pain, and showed empathy-related behavioral patterns similar to that in healthy controls, we recruited a healthy participant sample whose gender and age distributions were comparable to those of the patient sample ($n = 22$; 9 males, age = 23.18 ± 2.38 years old) (age: $t_{42} = -0.81$, $p = 0.424$, Cohen's $d = -0.24$, 95% $CI$: −8.91, 3.82, two-sided

two-sample $t$-test; gender: $\chi^2(1) = 0.82$, $p = 0.366$, two-sided Pearson's Chi-square test of independence). Healthy participants were asked to complete the same pain judgment tasks and provide subjective ratings of the strength of empathic response, perceived pain in others, and their own unpleasantness for each stimulus.

## Behavioral analysis

To deal with the ceiling effect on the response accuracies, similar to previous studies[66], we applied the arcsine-square-root transformation to accuracy rates before statistical analysis. For the response time (RT) analysis, we first log$_{10}$-transformed the original RTs and then excluded trials with outlier observations (longer or shorter than three median absolute deviations away from the median) or incorrect responses[67,68]. We then performed paired $t$-tests (two-tailed) to compare response accuracies and RTs between the painful and non-painful conditions (Supplementary Table 4 for the full statistical reports of response time and accuracy).

As for the subjective ratings, given variations in participants' rating patterns and preferences (e.g., rating ranges were smaller in some participants but more extensive in others), we first normalized subjective rating scores of each dimension across all stimuli for each participant[69,70], using the Eq. (1):

$$\text{Score}_{norm} = \frac{\text{Score}_{orig} - \text{Score}_{min}}{\text{Score}_{max} - \text{Score}_{min}} \times 100 \tag{1}$$

Where $\text{Score}_{orig}$, $\text{Score}_{min}$, $\text{Score}_{max}$, and $\text{Score}_{norm}$ denote the original, minimum, maximum, and normalized rating scores, all within a range of [0, 100].

Similar to previous studies[71,72], we examined the between-rater reliability for rating scores by calculating an intraclass correlation coefficient (ICC (2, $k$), using the R function ICC). A high ICC reflects that the total variance of ratings is mainly explained by the variance across stimuli, rather than raters. We observed excellent between-rater reliabilities (ICC greater than 0.90)[73] for all dimensions of subjective ratings (details in Supplementary Table 5). We then compared subjective ratings between painful and non-painful stimuli to check empathic responses to others' pain in our patients (two-tailed paired $t$-tests).

We then examined whether patients and healthy participants showed different responses and subjective feelings to vicarious pain. Data were averaged across trials in each condition for the response accuracy and time in the pain judgment task and subjective ratings. To account for individual variations, we examined group differences between patients and healthy participants by constructing linear mixed-effect models with participants as a random effect to control for individual variations. Specifically, the aggregated data were then entered into separate linear mixed-effect models to compare between two groups, which included fixed-effects of pain (painful vs. non-painful stimuli), group (patient vs. healthy) and their interactions, and random effects of participant. We then performed $F$-tests on the fixed effects estimates (two-tailed).

Moreover, we assessed how similar in subjective ratings between patient and healthy groups by calculating the Pearson correlations between patients' and healthy participants' averaged rating scores, across painful and non-painful stimuli, as well as only for painful stimuli. We also took individual variations into account to further confirm the similarity in subjective ratings between patients and healthy participants. Similar to previous studies[27,41], we considered the average ratings from all healthy participants as the normative ratings. We calculated Pearson correlations between each patient's or each healthy participant's ratings and the normative ratings. First, we examined whether ratings of patients and healthy participants were similar to the normative ratings by conducting permutation tests on Fisher-z-transformed patient-normative correlation coefficients or healthy-normative correlation coefficients[74,75]. The null distribution was generated by randomly

flipping the sign of Fisher-z-transformed patient-normative correlations or healthy-normative correlations. Furthermore, we examined the difference between the patient-normative and healthy-normative correlation coefficients (two-tailed permutation tests) by creating a null distribution via randomly shuffling membership between patient group and the healthy participant group.

## iEEG data acquisition and analysis

iEEG data were recorded using amplifiers from a Nicolet electroencephalogram system (256 channel amplifier, the Chinese PLA General Hospital) or a Nihon Koheden system (256 channel amplifier, Beijing Tiantan Hospital, Capital Medical University), or a Micromed system (128 channel amplifier, Xuanwu Hospital, Capital Medical University), with sampling rates of 4096, 2000, and 1024 Hz, respectively. For each participant, 5 to 14 electrodes were implanted. Each electrode was 0.8 mm in diameter and contained 5 to 18 contact leads 2 mm wide and 1.5 mm apart. iEEG data for the pain judgment task were collected when no subclinical or clinical seizures occurred during or immediately before the task.

**Signal preprocessing.** The iEEG data were analyzed using customized scripts of MATLAB and the Fieldtrip toolbox[76]. iEEG data were first down-sampled to 1000 Hz and zero-padded to minimize filter-induced edge effects. The signals were then band-pass filtered between 1 and 250 Hz and band-stop filtered between 47–53, 97–103, 147–153, and 197–203 Hz[77] to remove power-line noise (i.e., 50 Hz and its harmonics frequencies) using a zero-phase Butterworth filter with Hamming window. Channels underwent a quality check and were discarded if any of the following criteria were met[77]: (1) variances were five times greater than the median variance across all channels within the same category (gray matter channels or white matter channels); and (2) the number of jumps between consecutive data points larger than 100 µV was more than three times the median number of such jumps across all channels within the same category. All the remaining channels were also visually inspected to ensure that all bad channels had been removed.

iEEG signals from white matter channels have been shown with low variance and with the same amount of common noise as any other signal; thus, using white matter channels as the reference allows for the removal of common noise with a minimal bias to the re-referenced signal[78]. Therefore, we referenced the recorded iEEG data using white-matter referencing. Specifically, we applied two different white-matter-channel-based methods. We first referenced the iEEG data to the average of all clean channels in white matter (i.e., average white matter reference)[79]. Moreover, using the same reference may bring the fluctuations of this reference to the referenced signals, thereby yielding spurious correlations between two re-referenced signals[80]; thus, we further repeated the analyses with iEEG signals referencing to the nearest white-matter neighbor, yielding a bipolar montage[31,35]. Using this referencing scheme, a majority of cross-regional channel pairs (an average of 84.09% across all region pairs of interest) utilized different white-matter channels as references for their respective brain regions. We reported the results of the white-matter-average referencing scheme in the main text. Moreover, the basic patterns were reserved when all signals were referenced to their nearest white matter neighbors (Supplementary Figs. 16–18).

Epileptic charges were identified via an automatic assessment[81]: (1) the envelope of the unfiltered signal was five standard deviations away from the baseline (i.e., the whole time series); or (2) the envelope of the filtered signal (band-pass filtered between 25–80 Hz) was six standard deviations away from the baseline. Similar to previous studies[82,83], our neural analysis focused on the presentation phase of the painful or non-painful stimuli, and therefore each trial was epoched from 200 ms before to 500 ms after the stimulus onset. Any epoch containing epileptic charges was removed from further analyses and

any channel with more than 30% epochs removed from either painful or non-painful conditions was excluded (see Supplementary Data 1 for the detailed information about the number of remaining epochs). Finally, we visually screened all channels for epileptic charges and removed those with too many remaining artifacts. All visual inspections were performed while blinded to the experimental conditions.

**Electrode localization.** For each patient, the post-implantation computed tomography (CT) image was co-registered to the pre-implantation MRI. Two neurologists independently visually determined the locations of all channels in each patient's native anatomical space. Regions of interest (ROIs) were defined through AAL atlas combined with careful visual examination of anatomical landmarks[84–87]. For visualization, we firstly normalized the MRI images into the standard Montreal Neurological Institute (MNI) space using the Fieldtrip toolbox[76,88] and then rendered channel locations (the MNI coordinates of these channels) from all participants onto the Colin27 template brain[89] based on a high-resolution anatomical atlas using BrainNet Viewer[90]. The channel information of each region of interest is provided in Supplementary Data 2, 3. Note that channels from both hemispheres were collapsed to improve statistical power[31,32,35].

**Time-frequency analysis.** To remove the wavelet-transform-induced edge effect, each epoch was padded with sufficient data before and after the epoch. Using the Fieldtrip toolbox[76], we performed time-frequency decomposition for each epoch using complex Morlet wavelets with adaptive cycles, evenly spaced between 3 cycles (at 1 Hz) and 6 cycles (at 34 Hz) in a 1-Hz step and between 6 cycles (at 35 Hz) and 12 cycles (at 150 Hz) in a 5-Hz step[91]. After time-frequency transformation, the spectral power data were down-sampled to 100 Hz. Since power decreases at increasing frequencies, we pooled all pre-stimulus baselines and normalized the power by subtracting the mean of the pooled baseline and dividing by the standard deviation of the pooled baseline separately for each epoch, frequency, and channel. We then smoothed the power time series with a sliding window of 100 ms after zero-padding to eliminate potential artifacts[42]. To exclude potential baseline-induced bias, we baseline-corrected each epoch by subtracting the mean of the pre-stimulus baseline within that epoch. For each condition, power was averaged across all epochs separately for each frequency and channel. Consistent with the pipelines reported in previous studies[31,35], we statistically compared power changes between the painful and non-painful conditions (two-sided paired-$t$ test) and quantified the significance of the power differences and corrected for multiple comparisons using a non-parametric cluster-based permutation test[92] for all frequencies (4–150 Hz, theta band: 4–8 Hz; alpha band: 8–15 Hz; beta band: 15–35 Hz; gamma band: 35–70 Hz; high-gamma band: 70–150 Hz)[42]. First, for each time-frequency point (at each channel), we randomly shuffled the painful and non-painful condition labels. We then calculated a $t$ value between the shuffled conditions across all channels. Then, time-frequency points with uncorrected $p$ values (<0.05) were clustered based on the temporal and spectral adjacency. We calculated the sum of $t$ values for each cluster as its 'mass' and recorded the most extreme cluster mass among all clusters. All these steps were repeated 1000 times and the most extreme cluster masses were used to form a distribution of the null hypothesis. Finally, we extracted clusters based on the true data and calculated a corrected $p$ value for each true data cluster defined as the percentage of null cluster masses more extreme than the mass of that true data cluster. Any cluster with a corrected $p$ value less than 0.01 was defined as a significant cluster. We adopted this slightly more stringent threshold ($p < 0.01$) to reveal clusters with robust conditional differences. To further characterize the observed spectro-temporal patterns, we conducted two-tailed one-sample $t$-tests separately for painful and non-painful conditions on clusters with significant conditional differences (results shown in Supplementary Fig. 2).

To control for general activities, we adopted a resampling approach to compare the observed spectro-temporal power changes with those obtained from randomly selected channels located outside our four ROIs ("random channels")[27]. We ensured an equivalent number of channels within each ROI and of "random" channels. We extracted the power for the corresponding cluster (i.e., clusters with significant conditional differences in Fig. 2e–h; averaged across all relevant time-frequency points), and then conducted paired-$t$ tests to examine the conditional power differences among the random channels. This procedure was repeated for 1000 times, resulting in 1000 $t$ values to construct the null distribution. The observed conditional power difference was then compared with this null distribution. We found that the observed conditional effects of all these clusters were significantly stronger than those calculated based on random channels (ACC alpha cluster: $p = 0.011$, ACC beta cluster: $p < 0.001$; AI low-frequency cluster: $p < 0.001$; amygdala beta cluster: $p < 0.001$; IFG high-gamma cluster: $p < 0.001$).

**Temporal profile of the empathy-related power change.** Given that the neurobiological origin and functional significance of different frequency bands relative to specific frequencies are relatively well understood[25], we averaged power time series across frequencies of classic frequency bands (i.e., theta band, 4–8 Hz, alpha band, 8–15 Hz, beta band, 15–35 Hz, high-gamma band, 70–150 Hz) to investigate the temporal characteristics of frequency bands of interest, similar to previous iEEG studies[35,42]. This approach can avoid double-dipping issues and draw more general conclusions for the corresponding frequency band. Specifically, we examined the temporal profiles of empathy-related activities in the amygdala/ACC/AI (of the low-frequency, i.e., theta band, 4–8 Hz, alpha band, 8–15 Hz, beta band, 15–35 Hz) and IFG (of the high-frequency, i.e., high-gamma band, 70–150 Hz) by averaging the normalized power time series across frequency range of interest. We then detected time windows with significantly increased (or decreased) power in response to painful (vs. non-painful) stimuli. We conducted paired-$t$ tests to compare the power between the painful and non-painful conditions across all channels for each time point and used similar non-parametric cluster-based permutation tests to correct for multiple comparisons[35]. The null distribution of differences between conditions was generated by randomly shuffling condition labels 1000 times for each channel at each timepoint. These comparisons were conducted as follow-up analyses for the time-frequency analyses in order to further reveal the onset time for empathic neural activity. Since the directionality of the conditional differences was already shown in the time-frequency power analysis, i.e., whether there was higher or lower power in the painful vs. non-painful contrasts (Fig. 2e–h), one-sided paired $t$ tests were applied for these analyses (corrected $p < 0.01$, 1000 permutations; all results were maintained even when we applied two-sided paired-$t$ tests). We adopted a 50 ms threshold for contiguous significance to avoid using spurious transient power increase or decrease as markers for activity onset[43,44]. The onset time of empathy-induced neural activity was defined as the first time point within time ranges showing a significant increase (or decrease) in the power to painful (vs. non-painful) stimuli. It should be noted that we found similar temporal characteristics of the ACC/amygdala/AI/IFG when we smoothed the power time series with a smaller sliding window (from 100 ms to 50 ms) to attenuate potential confound associated with temporal smoothing, on the one hand, and to increase signal-to-noise ratios, on the other hand (Supplementary Fig. 19).

**Power correlation analysis.** We assessed functional communications between the AI, ACC, and amygdala by performing power correlations analyses. Similar to previous studies[32,45], we investigated the power correlations between each region pair of the AI, ACC, and amygdala in overlapping frequency bands that significantly differentiated between painful and non-painful stimuli (including the alpha and beta bands for

ACC-AI and the beta band for ACC-amygdala and AI-amygdala). To eliminate the potential influence of individual differences, we only considered pairs of channels within each participant (i.e., the two channels of each pair from the same participant) and included participants with at least one channel pair.

Power correlation analyses were conducted among amygdala, AI, and ACC (234 ACC-AI pairs, 72 ACC-amygdala pairs, and 300 AI-amygdala pairs, Supplementary Data 3) based on the power time series smoothed with a smaller sliding window of 50 ms. Considering the non-normality of the power time series, we calculated the Spearman correlation coefficient across the post-stimulus time window (i.e., 0–500 ms) for each epoch, each frequency, and each channel pair. For each condition, the correlation coefficients were Fisher-z-transformed (note that the reason to perform this transformation was to satisfy the assumption of normal distribution for statistical evaluation and this transformation did not introduce any distortions to the original power correlation pattern that was shown in Supplementary Fig. 20) and averaged across epochs. We then compared the power correlation values between painful and non-painful conditions by performing cluster-based permutation tests (corrected $p < 0.01$, two-tailed), in which we randomly shuffled the condition labels for each channel pair 1000 times to create the null distribution. To aid a better understanding of conditional differences in power correlation values, we also conducted two-sided one-sample $t$-tests to examine whether the power correlation values (averaged across the spectral clusters showing significant conditional differences) significantly deviated from 0 separately for painful and non-painful conditions (results shown in Supplementary Fig. 5).

**Transfer entropy analysis.** Given that power correlation was an undirected measure which cannot convey directional information about inter-regional interactions[45,93], we further estimated the directionality of power correlation results by computing TE values using the neuroscience information theory toolbox[48]. The choice of using TE to index effective connectivity was based on previous work[94], which showed that TE, as compared to Granger Causality and other model-based approaches, is a model-free measure of effective connectivity based on information theory and better at detecting effective connectivity for nonlinear interactions. TE also requires no a priori assumptions on interaction patterns and is suitable for electrophysiological data[95–97]. Since previous research has not examined the electrophysiological basis for effective connectivity within human empathy network, and we had no clear assumptions about the interaction patterns, we thus took the advantages of TE and its exploratory nature to reveal the directions of the observed inter-regional low-frequency coupling. Specifically, TE measured the mutual information exchange between Y at lag $t_1$ and X at $t_0$ given the information about Y at $t_0$[48]. We computed bidirectional TE (i.e., TE from the AI to the amygdala, as well as from the amygdala to the AI) of the minimally smoothed power data (smoothed with a 50 ms time window) across the stimulus-presentation time window for each epoch, each frequency, and each channel pair at a lag of 10 ms. The TE values were then averaged across epochs and frequencies (among the frequency range showing significant conditional differences in power correlations, Fig. 3a–c). We conducted repeated-measures ANOVAs with Pain (painful vs. non-painful stimuli) and Direction (e.g., AI-to-amygdala vs. amygdala-to-AI) as within-subjects factors, followed by the planned two-tailed paired $t$-tests to compare painful and non-painful conditions separately in each direction. We also assessed the sensitivity of our TE results by conducting TE at different lags, with lag values ranging from 10 ms to 50 ms in increments of 10 ms. This analysis showed that the observed patterns in TE remained relatively stable as the lag increased (Supplementary Fig. 21).

**Phase-amplitude coupling analysis.** Given that the contrast of painful vs. non-painful stimuli showed low-frequency oscillatory

alternations in the amygdala/ACC/AI but high-frequency power increases in the IFG, we thus examined cross-frequency interactions by computing the inter-regional phase-amplitude coupling (PAC) values[31,35]. The PAC value can be used to index directional, cross-frequency coupling between two regions (i.e., whether the low-frequency phase at region A modulated the high-frequency amplitude at region B)[37,38]. The iEEG data were zero-padded and filtered in the low-frequency bands to obtain instantaneous phase (phase frequency: 4–35 Hz, in 1-Hz steps) and high-gamma band to obtain instantaneous amplitude (amplitude frequency: 70–150 Hz, in 2-Hz steps). The phase time series were computed by taking the angle of the Hilbert transform of the bandpass filtered data around phase frequency with a bandwidth of $\pm 1\,Hz$[98]. The amplitude time series were acquired by taking the magnitude of the Hilbert transform of the bandpass filtered data around amplitude frequencies with a bandwidth equivalent to the corresponding phase frequency (i.e., a frequency window of amplitude frequency $\pm$ coupling phase frequency[99,100]). For example, when centered at an amplitude frequency bin of 90 Hz and a phase-frequency bin of 30 Hz, the band-pass filter passes data in a range between 90 and 30 = 60 Hz and 90 + 30 = 120 Hz to obtain the analytic amplitude. The employment of varying bandwidth rather than fixed bandwidth for the amplitude component was to ensure that the band of amplitude component was sufficiently large to contain the peaks produced by the low-frequency phase[99,100]. Consistent with previous studies[31,35], the inter-regional PAC values were computed as the circular-linear correlation coefficient between the phase time series of the amygdala/AI/ACC and the amplitude time series of the IFG (across the post-stimulus time window, 0–500 ms) and Fisher-z-transformed and averaged across all epochs in each condition.

Similarly, we conducted a non-parametric cluster-based permutation test to compare the Fisher-z-transformed PAC values between the painful and non-painful conditions (two-tailed paired-$t$ test for each frequency pair, corrected $p < 0.01$, 1000 permutations). The distribution of the null hypothesis was estimated by randomly shuffling the condition labels within each channel pair at each frequency pair and clustering the supra-threshold frequency pairs based on their spectral adjacency. To better understand the observed PAC patterns, we further examined the PAC values within each condition (averaged across spectral pairs within clusters showing significant conditional differences). It is important to note that PAC values were indexed by circular-linear correlation coefficients, thus the PAC values were non-negative (i.e., larger than 0)[31,35]. Therefore, we assessed the significance of PAC values for each condition using permutation tests (one-sided)[101], in which we shuffled data by cutting the amplitude time series at a random time-point into two parts and then reversing the order of these two parts, and recomputed the PAC values to generate the null distribution[101] (results shown in Supplementary Fig. 9).

### Decoding analysis
We conducted a decoding analysis to discriminate the perception of painful stimuli from that of non-painful stimuli based on regional power and inter-regional communications within and across ACC, AI, amygdala, and IFG. The dimensionality of these neural variables was extremely high, which can potentially lead to the overfitting problem[102]. To address this issue, we first performed feature selection to identify important empathy-related features. We then decoded stimulus types based on these selected features[103]. To avoid information leakage problem[104–106], we performed feature selection and model construction using separate datasets. Specifically, we randomly split our data into two independent sub-datasets at the channel (or channel pair) level, with 70% of the data as the feature-selection dataset to identify informative features, and the remaining 30% of the data as the decoding dataset to construct the classification model to decode stimulus types.

Within the feature-selection dataset, we repeated the time-frequency analysis, power correlation analysis, and PAC analysis. The resulting univariate conditional difference maps were used to identify informative neural features of regional activity (i.e., the time-frequency points of power) or inter-regional communication (i.e., spectral ranges for the power correlation and PAC), but with a less stringent (i.e., the classic cluster threshold of corrected $p < 0.05$) to reduce risk of false negatives resulting from the smaller number of channels or channel pairs[107,108]. The use of a separate sub-dataset, instead of the full dataset, to select relevant features has been suggested as a way to avoid potential problems of information leakage[104–106].

Within the decoding dataset, we performed a decoding analysis using support vector machine (SVM) classifiers for binary classification with a linear kernel to test whether and how these neural features could predict the perception of painful and non-painful stimuli. First, based on the identified critical time-frequency points of power and spectral ranges of the power correlation and PAC, we extracted the corresponding neural features for the decoding dataset. For each feature, we randomly sampled $m$ channels/channel pairs ($m$ = the minimal number of channels or channel pairs across all features) and averaged it across all sampled channels/channel pairs (this sampling was repeated for 200 times)[109]. This was done to integrate all features into a single decoding model in a way that the number of channel/channel pairs have been balanced across features. To ensure that features with larger numeric ranges did not outweigh those with smaller numeric ranges, similar to previous work[110], we linearly scaled each feature to the range [0, 1] before applying SVM.

SVM classifiers were trained to find the optimal hyperplane that distinguished neural patterns between painful and non-painful conditions (using the fitsvm function in the MATLAB). SVM (C = 1[111,112]) algorithm was employed because it was relatively insensitive to the sample size[113,114] and suitable for decoding analysis with a small sample size[115] (for illustrative purpose, we depicted the variations of decoding performance across different sample sizes in our dataset, Supplementary Fig. 22). A linear kernel was employed to reduce model complexity and minimize the likelihood of overfitting[116]. Similar to previous studies[117,118], the classification performance of the SVM classifier was evaluated by a five-fold cross-validation procedure. We randomly divided all trials into 5 folds. Each fold was used in turn for testing. For each iteration, the classifier was trained on the remaining 4 folds and then tested on the held-out testing fold. The decoding accuracy of the confusion matrix (i.e., (true positive + true negative)/ total observation) was calculated based on the average of five iterations. Due to the instability of accuracy, we calculated the average accuracy across all iterations and resamples as a result.

We assessed the statistical significance of the decoding accuracy using a permutation test[104]. In each permutation, we randomly shuffled the class labels and the entire decoding procedure detailed above was repeated. After 1000 permutations, we obtained the null distribution of decoding accuracy, which can well-capture the variability of decoding accuracy corresponding to the current sample size. We controlled for the influence of limited sample size on the variations of decoding accuracy by comparing the true decoding accuracy with this null distribution (one-tailed, with the threshold of $p < 0.01$).

We further compared the importance of the neural features in the discrimination between painful and non-painful stimuli by computing the averaged classification weights[49,50]. Next, we assessed whether each feature or feature combinations were necessary to detect perception of vicarious pain by conducting data simulations (a "virtual disruption" analysis)[51,52] to estimate the removal of which feature(s) led to failure in the decoding of vicarious pain perception. Here, the operations to respectively remove each feature from the model resembled the scenarios when a specific feature was disrupted. As none of the neural features can individually determine the success of the decoding, we further searched for feature combinations which were necessary for successful decoding. We performed stepwise classification by removing features sequentially and evaluating the classification performance of the remaining features until the decoding accuracy was reduced to the chance level. Those removed features consisted of a necessary feature combination for successful decoding. To lend further evidence to strengthen the importance of the detected feature combination, we took into account the influence of feature numbers and compared the decoding accuracy when removing this feature combination to that when randomly removing an equal number of features (random removing for 1000 times, with the threshold of $p < 0.01$, two-tailed).

### Relationship between empathy-related ratings and important neural features

To test whether the top 8 neural features also captured elaborated information related to empathic responses to other's pain, we examined the relationship between these neural features and empathy-related ratings. For each of the eight neural features, we first averaged this feature in the corresponding time-frequency points/spectral change that significantly differed between painful and non-painful stimuli in the whole dataset (Fig. 2e–g; Fig. 3c; Fig. 4a–c). We conducted linear mixed-effect models to examine the associations between the conditional differences (painful minus non-painful) in neural features (dependent variable) and differential ratings of normalized empathic strength ratings of the corresponding participant (independent variable) across all matched painful and non-painful pairs of stimuli[83] with patient and channel (channel-pair in coupling analyses) identities included as the random effect to control for inter-patient and inter-channel or inter-channel-pair variations. For each feature encoding empathic strength, we conducted similar analyses to explore whether and how this feature was associated with perceived pain intensity in others and one's own unpleasantness when witnessing others in pain. All the independent and dependent variables were standardized before entering in the model and all the two-tailed $p$-thresholds were adjusted via FDR-correction based on the number of statistical tests. Given the limited number of patients who completed post-iEEG ratings, we applied a less stringent threshold of FDR corrected $p < 0.05$ for all these analyses to reduce risk of false negatives. Finally, we also checked the associations between the neural features encoding empathy strength and arousal ratings to further demonstrate their specificity in the encoding of empathy-related behavioral responses.

### Reporting summary

Further information on research design is available in the Nature Portfolio Reporting Summary linked to this article.

## Data availability

The raw and preprocessed iEEG data generated in this study have been deposited in a local database. The iEEG data are available under restricted access as they contain personally identifiable information and patients have not consented to data distribution. Access can be obtained from the corresponding author upon request. Source data are provided with this paper.

## Code availability

The code to perform wavelet transform and compute power correlations and phase-amplitude coupling is publicly available (https://github.com/Huixin-Tan/iEEG_empathy). The access of other codes is available from the corresponding author upon request.

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

## Acknowledgements

We thank C. Hao and W. Yuan for their assistance in data collection. This work was supported by the National Natural Science Foundation of China (Projects 32125019 to Y.M.; 32230043 to S.H.); STI 2030—Major Projects 2022ZD0211000 to Y.M.; the Fundamental Research Funds for the Central Universities (2233300002 to Y.M.); and the start-up funding from the State Key Laboratory of Cognitive Neuroscience and Learning, IDG/McGovern Institute for Brain Research, Beijing Normal University (to Y.M.).

## Author contributions

Y.M. conceived of the project and designed the experiments; X.Z., Z.L., J.Y., K.L., Y.Z., C.X., C.H., Y.G., X.Y., and F.M performed the experiments and collected data; H.T., J.N., and J.W. analyzed the data under the supervision of Y.M.; H.T. and Y.M. wrote the original draft. S.H. provided critical revisions. All authors approved the final version of the manuscript for submission.

## Competing interests

The authors declare no competing interests.
