## [Peer Review File · Nature Communications]

Intracranial EEG signals disentangle multi-areal neural dynamics of vicarious pain perceptionREVIEWER COMMENTS

Reviewer #1 (Remarks to the Author):

Summary: Tan and colleagues reported results of neural dynamics within an empathy network, including anterior insula (AI), anterior cingulate (ACC), amygdala (AMY), and inferior frontal gyrus (IFG) in supporting the perception of other's pain. The data was collected in 22 drug-resistant epilepsy patients while they were undergoing seizure monitoring for potential surgical treatment at the hospital. Participants were presented with a series of images stimulation applied to other's hands (e.g., a hand was/wasn't cut by a knife) and were instructed to make a judgement of painful or non-painful decision. 16/22 participants also completed the post-iEEG session, during which they report subjective ratings of each stimulus, including their empathic responses, intensity of perceived pain in others and their own unpleasantness. Overall, authors examined the oscillatory power, power correlation, inter-regional directionality and phase amplitude among this empathy network, and evaluate the predictive power of the observed neural features with participants' painfulness judgement. In particular, they found:

- 1) An early and sustained high-gamma power increase in IFG, an early beta oscillatory increase in ACC, but decreased beta oscillation in AI and AMY for painful compared to non-painful conditions.
- 2) Power correlation in beta band was increased within ACC-AMY and AI-AMY pairs for painful condition compared to non-painful cases.
- 3) Phase amplitude coupling between IFG high-gamma activity and beta phases of AMY/AI/ACC increased for painful compared to non-painful conditions.
- 4) Several neural features, including AMY/ACC/AI low frequency powers, their power correlation and interaction with IFG high gamma activity can predict participants' strength of empathic responses.

The dataset is unique and precious, with unparalleled spatiotemporal resolution in understanding the perception of others' pain in humans. The results are interesting and potentially provide circuit-level understanding for the proposed empathy network. However, the manuscript is not very well written, missing clear theoretical motivation, clarification of method selection, and interpretation of findings. I have listed several major concerns and some minors as well that need to be addressed to meet the quality standard to be published in Nature Communication.

Major:

1. The introduction needs to be better written to: 1) include more information that inspires and is related to this study. For example, more literatures of anatomical/functional connectivity within this empathy network, similar to Line 106-107, can help readers better understand the motivation of

looking into these four regions. 2) the content in each paragraph should have a clear topic. For example, Line 73- 81 and its following paragraph seem to contain redundant information, please consider rephrase the paragraph and make the description consistently and concisely. 3) clarify the findings from previous literatures and how it inspires this study. For example, line 70-71, what does the prior knowledge mean in this sentence “The ACC activity also mediates the top-down modulation of empathic responses to others’ pain (e.g., prior knowledge27).”

2. The task design has some potential pitfalls that raise concerns about whether it is suitable to address the scientific questions tested here.

a. As authors reported (line 753-754), there were only 10 painful stimuli and 10 non-painful stimuli in this task, that only presented once. The actual trial numbers included in this task would be even smaller, after excluding the trials with incorrect pain judgement and inter-ictal discharges. It is questionable whether such limited trial number ($n < 10$ per condition) has enough statistical power, especially for SVM decoding analyses (see Major point 5 as well).

b. A subgroup of participants (16/22) provided their subjective ratings, including empathy strength, intensity of perceived pain in others, own unpleasantness during the post-iEEG period, which were likely collected several days after participants performed the original task recording. How representative these subjective ratings were in reflecting participants’ internal states when performing the original task? Author should provide evidence to support the utilization of such subjective ratings that were collected far away from the original experiment. Also, the subjective ratings are collected during the second exposure to the stimuli that might be influenced by the adaptation effect. Therefore, the results in Fig. 6 that use the neural features collected during real experiment and ratings during post-iEEG period is hard to interpret.

3. Almost all the results reported here are based on the statistical testing between painful and non-painful conditions. However, to claim that the observed neural features are due to the perception of others’ pain, it is also important to demonstrate the significance level within each condition. Because a given neural signature might show significant difference across conditions but not significant within condition. Below are listed several analyses are related to this concern.

a. Figure 2E-H, is power in painful or nonpainful condition significantly different from the baseline? It is worth extending the analysis window to baseline to ensure that the conditional differences are due to the process of painful vs nonpainful stimuli.

b. Fig. 3A-C, what’s the actual correlation value without Fisher-z-transfer? Also, whether this is significant before comparing across conditions The following statement can be hardly supported with the current figure: “AI-amygdala pairs showed significant alpha/beta power correlations in both painful (8-35 Hz; Fig. 3C) and non-painful (8-34 Hz; Fig. 3C) conditions.” Is the significance measured simply against to zeros?

c. Fig. 4A-C, what phase-amplitude coupling look like for painful or non-painful condition separately? Are they significant within condition as well?

4. The inclusion/exclusion criteria for specific analyses in this paper lacks clear justification.

a. Fig. 3A-C and Line 999-1001, the power correlation analysis was focused on the channels showing significant power difference between painful and non-painful conditions. However, the computation of power correlation analysis itself was done individually on each condition first and then compare across. Then why significant power differences should be a selection criterion here for the power correlation analysis? Also, even with this selection criteria, why the power correlation analysis are not covering theta band, which AI (Fig. 2E) shows significant power difference between painful versus nonpainful conditions?

b. Fig. 3D-E, is there any evidence to support the exclusion criteria for the Transfer entropy analysis? Authors wrote that (line 324 -326) “situations without significant power correlation indicated that no meaningful functional interaction was involved and further investigation on the information transfer flow was unwarranted.” First, various of methods that quantifies inter-regional functional interactions are phase based (e.g., phase locking value) that does not require strong power correlation. Therefore, power correlation is not necessary for meaningful functional interaction. Second, based on the definition of transfer entropy as the author described (Line 1031 -1033), it is possible to have it without significant increased power correlation. The directionality could be totally independent from this. So why significant power correlation should be a selection criterion here?

5. Question about the SVM analyses

a. Line 1087-1089, it seems like authors split the dataset for training and testing at the channel level. Does that mean authors were decoding the subjective ratings across participants? If so, I found this very problematic, as the subjective ratings across participants might not consistent with each other, and also the neural signatures might contained individual variations. If not, I found it hard to decode within participant with such limited trial number ($n < 10$ trials per condition, see Major point 2 as well).

b. Fig. 5B, why does the decoding accuracy drop after integrating more features (>8)? Isn't it supposed to get higher or at least stay the same?

c. Fig. 5C, it is expected to have higher decoding accuracy with more features. If authors tried to emphasize the importance of top 6 features for decoding, a better comparison here should be with models excluding randomly 6 features.

d. Fig. 5D, how to understand the negative correlation value between features 2-5 and Empathy strength?

6. Some analytic approaches used in this paper might not be suitable and needs additional controls.

d. Line 177, Is the similarity simply the correlation of rating scores from patient group and control group? If so, it might be problematic as the ratings here (e.g., Fig. S1) are averaged across all the patients or across all healthy subjects without taking into account the variations across subjects. A better statistical method needs to be used.

e. Line 846-848, what is the pairwise Euclidean distance between the corresponding rating differences for each participant pair? Even the two subject population have comparable age and gender, pairing subjects across groups seems nonsense.

7. The discussion section is supposed to summarize the key findings in the paper, provide reasonable interpretation, and inspire additional research directions. However, it is really hard for readers to learn from the reported results in this paper and how that could strengthen our knowledge about the empathy network.

a. For example, Line 620-624, author makes the claim that “increased ACC beta oscillations may support the top-down modulation of vicarious pain perception by prior knowledge or experience while decrease AI beta oscillations may associate with bottom-up affective responses triggered by perceived pain in others.” How does this hypothesis fits into this task?

b. Line 641 and 642, then what is the functional role of amygdala in perception of other’s pain? Especially with such late response

c. Line 645 to 661, what information the inter-regional communication can provide in addition to the power increase/decrease?

d. How to understand the results from the directionality analyses?

e. Line 680- 683, based on what results, that authors conclude that IFG might play a role in understanding the target’s action itself and automatic action simulation? Also, how does this connect to the perception of others’ pain?

8. Authors should be mindful of using terms like “first” when describing the results. For example, Line 566 -568, authors claim that “our study is the first attempt to integrate region-specific neural oscillations and inter-regional interactions to decode vicarious pain perception, enabling us to characterize how these neural features jointly contributed to viscarious pain perception”. Also, Line 1028-1030, “our study is among the first to examine the electrophysiological basis for effective connectivity within human empathy network, and no clear assumptions on interaction pattern existed.”

Minor:

1. It will be helpful to provide a schematic plot, similar to Fig. 1A, to demonstrate the post-iEEG session as well. Also, how much time away from pain judgement and post-iEEG session?
2. $XX \pm XX$, standard deviation vs. standard error mean needs to be stated in the manuscript.
3. Line 142 to 144, it is helpful to show the actual value of response accuracy and response time along with the t-test.
4. I assume that the data was collected with Chinese instruction. We appreciate that the authors have translated the instruction to English (Figure 1A). It is also informative to show the original task instruction in a supplementary Figure.
5. Maybe Colorbar label in Figure 2E-H should not be Power, but instead a power difference (t value)?
6. Figure 2I-L, is the plotted power here normalized? The value seem to be around zero before $t = 0$. If so, please update the y axis to normalized power instead.
7. Fig. 2J, the figure title "Beta band" seems to be smaller than the rest.
8. Are the results (Fig. 3 and Fig. 4) only include electrode pairs within the same subject and same hemisphere? If so, please clarify in the text.
9. The thickness of the lines in Fig. 3A-C seems not very consistent, is this on purpose?
10. In Fig. 3D, What's the significance level for AI-amygdala within alpha band between painful and nonpainful conditions?
11. It seems like transfer entropy is larger for Non-painful compared to painful conditions? What does that mean?
12. What is the difference between Fig.4 and Fig. S6? What is the varying vs fixed bandwidth? Also, why the results between IFG and amygdala are so different between Fig 4 and Fig. S6?
13. Line 787- 789, please provide details for how the arousal assessment was done in patients. Categorically measured or continuously measured?
14. Line 793-794, author mentioned that they have recruited the gender-matched healthy control group for the study, which has 22 subjects in total with 9 males while the patient group has 13 males. So it's not entirely matched. Please make sure the consistency of description.
15. Authors mentioned that the spikes (defined as $>100\mu\text{V}$ changes between consecutive samples) were removed from original data. Spikes can also refer to single neuron activation. It might be less confusing to change it to a different term
16. Line 1015 and Line 1019, what is the cluster-based permutation here, each trial has only one correlation coefficient?

Reviewer #2 (Remarks to the Author):

The paper revolves around the neural underpinnings of empathy for pain using intracranial electroencephalography (iEEG) to elucidate with high-resolution the spatio-temporal profiles of neural oscillatory activity and inter-regional communications within the empathy network during the perception of others' pain. While the research protocol for testing empathic reactivity to pain is a standard one, the technique used is a sophisticated one and the analytic approach state-of-the-art. While this paper may advance our understanding of the neural dynamics that underpins empathy for pain there are several points that need to be clarified

A point by point list of comments is provided below.

1. The introduction does not provide a detailed overview of the importance of empathy in social interactions and its neural basis. In particular, the paper does not provide a thorough and well-defined overview of the existing literature that utilizes intracranial electroencephalography (iEEG) to explore empathy for pain. Notably, the works of Soyman et al. (2022) and Mo et al. (2022) are absent in the manuscript, leading to an incorrect assertion of the technique's novelty within the field. This is a major point of weakness. While this may be considered as a lack of scholarship, it may correspond to a simple overlook. However, discussing the above papers is fundamentally important for comparing the findings of the present research with what is already known.
2. The methods section detailing the use of iEEG in epilepsy patients is well-described. However, additional information regarding patient selection criteria, ethical considerations, and potential confounders would enhance the methodological robustness of the study. Moreover, it is not clear to me whether 6 patients were excluded because failed to complete the post iEEG session or whether they did not have electrodes implanted in the regions of interest. Both sentences are written in the manuscript, and it is hard to understand why 6 patients were excluded from the data (or if in different moment different 6 patients were excluded). Moreover, it might be interesting to indicate for each patient where the electrodes were implanted (e.g., patient 1, X electrodes in AI, X electrodes in ACC, etc).
3. A main concern is whether the authors have data about areas which are not involved in the empathic network to control for general activities (see Soyman et al., for a comparison between the insular electrodes and “random” electrodes). It seems that the authors use signal from white matter as reference but there isn't an actual comparison between areas which shouldn't be implicated in processing empathic responses.
4. Related to this, the introduction lacks the rationale of selecting AI, ACC, IFG and the amygdala and no other areas, such as the motor or the somatosensory cortex or the mCC (Fallon et al., 2020).
5. In the neural features analysis, it is not clear why authors selected 7 features and then excluded the seventh one and select 6 as sufficient. Please explain this choice.
6. Other analytical choices are not clearly motivated. On page 12, line 241, for example, it is not clear why the authors adopt one-sided (rather than two sided) comparisons.

7. The discussion is difficult to read. I would like to suggest that the authors help the readers by adding title for each section (as they did for the result).

8. Previous studies hinting at the dynamic nature of brain rhythms during empathy for pain should be quoted and briefly discussed even if they do not use iEEG (e.g. Betti et al, 2009; Zebarjadi et al, 2021)

Minor points

a) Some of the information in the analysis part repeated in the method section. I suggest avoiding presenting them twice and to leave the information sufficient to understand the analysis in the results section and all the details in the methodology.

b) The results are presented in a clear and logical manner. However, the manuscript is very dense, presenting several analyses in a way that it is hard to follow. Some sections with many statistics could be organized into tables.

c) Line 511: it is not clear to me why one behavioral component has been removed from this analysis.

d) Line 190: I would suggest the authors to state that the empathic responses to vicarious pain were comparable to healthy controls – more than representative of a general population.

e) Line 123- 130: this sentence needs to be rephrased

f) Line 140: felt painful

g) Supplementary tables with the subjective data of patients and controls rather than just similarity indices may be useful

Reviewer #3 (Remarks to the Author):

This work utilizes intracranial electrophysiological recordings in humans to examine in detail the complex interplay between the anterior insula, the anterior cingulate cortex, the amygdala and the inferior frontal gyrus across different oscillation frequencies at a millisecond timescale during vicarious pain perception. Highly valuable iEEG data collected from pre-clinical epilepsy patients in a well-established pain judgment task was analyzed with advanced intra- and inter-regional neurophysiological analysis techniques and the results were presented in a highly organized and

clear manner. In addition, the relative importance, the necessity, and the sufficiency of a wide array of neural activity markers were assessed via a decoding algorithm to construct a neurodynamic model of empathy for pain. Overall, this is a very challenging, and thus highly valuable work that enhances our understanding of the interactions between critical brain regions that underlie vicarious pain representations.

I have a number of comments/recommendations that I believe will increase the impact and the understandability of the manuscript.

In the third paragraph of the Introduction section, immediately after discussing the limitations of studying empathic neural responses using fMRI and EEG/MEG and the advantages of using intracranial methods, the goal of the present study is introduced. This gives the feeling that vicarious pain responses have never been investigated before using intracranial recordings in the four regions of interest in the human brain. However, this is not true. The authors should refer to studies such as

Hutchison, W. D., Davis, K. D., Lozano, A. M., Tasker, R. R., & Dostrovsky, J. O. (1999). Pain-related neurons in the human cingulate cortex. *Nature neuroscience*, 2(5), 403-405.

Soyman, E., Bruls, R., Ioumpa, K., Müller-Pinzler, L., Gallo, S., Qin, C., ... & Gazzola, V. (2022). Intracranial human recordings reveal association between neural activity and perceived intensity for the pain of others in the insula. *Elife*, 11, e75197.

In the last paragraph of the Introduction section, different sentences use different tenses. For readability, please stick with the present or past tense consistently throughout the paragraph.

The fact that iEEG participants did not show any significant differences in accuracies and reaction times between the painful and non-painful stimuli during the pain judgment task is shown in sufficient detail. However, the average, the standard deviation, the minimum, and the maximum of these accuracies and reaction times for painful and non-painful stimuli are not shown in the Results or the Methods section. These values must be clearly reported so that the reader can grasp what type of performances the participants showed in the task.

In the sentence starting at line 257, it is stated that a smaller sliding window was used for the analysis reported in the Supplementary Figure 3. Although, it is reported clearly in the earlier pages that the first analysis was conducted with a 100-ms window and this analysis is conducted with a 50-ms window, the reader has to go between several pages to fully grasp what is being changed in this analysis. The reader would substantially benefit if the authors clearly state here the change was from the earlier 100-ms to 50-ms time window.

In the sentence starting at line 257, the authors state that the smaller sliding window analyses showed similar temporal profiles to the longer sliding time window. However, as seen in Supplementary Figure 3G, there was a very critical difference in that the latency of the IFG high-gamma responses were 160 ms, as opposed to the 60-ms latency in the analyses with the longer smoothing window. This latency difference challenges the neurodynamic model proposed in Figure 6, which present very early IFG responses followed by other brain regions. The authors must discuss this critical difference openly in the manuscript.

In the paragraph starting at line 308, the authors use sentences as “The ACC and amygdala synchronized to a greater degree in the painful than non-painful conditions...”. It is as if there was synchronization in both conditions and one condition was higher than the other one. In fact, the analyses show that there was no synchronization at all in the non-painful condition. All the sentences in this paragraph must be rephrased to capture this critical information in order to prevent such misinterpretations.

In Figure 5D and the related analyses in the text, the statistical significance of the correlations between empathic strength and neural features are reported in detail. However, quite importantly, some of these correlations are positive and some are negative, which is not stated or discussed in the manuscript. Especially for neural features 4, 5, and 6, the direction of the correlations are quite surprising given the analyses reported in earlier sections of the manuscript. To take neural feature 6 as an example, the amygdala beta oscillation analyses reported in Figure 2K shows that these oscillations are significantly decreased during vicarious pain. Since empathic processing would increase during vicarious pain, one would expect a negative correlation between these two measures, whereas Figure 5D shows a positive correlation. I do understand that these ladder analyses are conducted on painful-nonpainful conditional difference scores, and thus do not argue that there is necessarily an analytic error here. But these nuanced differences in analyses must first be explicitly stated in the Results section and then also how they should be interpreted, as well as their potential explanatory power in terms of the neurodynamical processes underlying empathic processing, must be openly discussed in the Discussion section.

In the sentences starting at line 540, the authors report that the neural activity did not vary as a function of arousal levels. However, whether there was a significant difference between the painful and non-painful stimuli in these arousal levels is not reported. The results of such an analysis must be reported here.

The Discussion section would benefit substantially if the authors discuss their anterior insula findings in relation to the findings of the Soyman et al. (2022) study, which analyzed intracranial

recordings from the insula while the participants engaged in a vicarious pain perception task similar to the one used in this study.

In the sentence starting at line 769, the authors state that the participants pressed the left or right button for reporting their answers for the presence or the absence of pain. Was the assignment of the buttons to the responses randomized across trials or participants? If not, wouldn't the systematic association of one motor behavior with one response lead to systematic biases in the neural responses?

In the sentences starting at line 774, the authors state that the 20 pictures were shown once, which means that the maximum number of trials that could go into any analysis was 20. Furthermore, in the sentence starting at line 913, it is stated that any channel with more than 30% epochs removed from either painful or non-painful conditions was excluded. This means that it was possible for a channel to be included in the analyses if it had 7 painful and 7 non-painful trials. For electrophysiological analyses, these are surprisingly low numbers of trials. In addition, some of the analyses in the manuscript are conducted by taking the conditional difference between matched pairs of painful and non-painful stimuli. If unmatched trials were rejected, these analyses would be conducted with even lower numbers of matched trials. For each analysis, the author must report the mean, the standard deviation, the minimum, and the maximum number of trials (across channels) that were taken into account when computing that particular metric or conducting that analysis for transparent reporting of the analytical procedures.

The MNI coordinates of all channels included in the final analyses must be reported in a supplementary table or file. In addition, this file must have, for each channel, the MNI coordinates of the nearest white-matter neighbor reference channel used in the bipolar montage analyses. In the Methods section, around the sentence starting at line 899, the authors must clearly state, in this bipolar montage referencing, whether the same nearest white-matter reference channel was used for channels that were placed in different regions in the brain. The purpose of the bipolar analyses is to overcome the potential problem of increasing the estimated correlations between different brain regions that can stem from having the same reference signal for those brain regions. Thus, the authors must clearly state that indeed different reference channels are used for channels in different brain regions in the bipolar montage.

In the sentence starting at line 1039, the authors state that transfer entropy was computed at a lag of 10 ms. Considering that the spectral power data was down-sampled to 100 Hz and a 100-ms sliding time window was used for sliding, responses at a 10-ms lag would be heavily influenced by a wide temporal window that intersects at a substantial amount of earlier and later time points in the two channels. The transfer entropy analysis must be conducted at various lags starting from 0 ms to 100 ms to assess the sensitivity of the reported results.

Reviewer #4 (Remarks to the Author):

I co-reviewed this manuscript with one of the reviewers who provided the listed reports as part of the Nature Communications initiative to facilitate training in peer review and appropriate recognition for co-reviewers.

➤ **Responses to Reviewer #1's comments:**

Summary: The dataset is unique and precious, with unparalleled spatiotemporal resolution in understanding the perception of others' pain in humans. The results are interesting and potentially provide circuit-level understanding for the proposed empathy network. However, the manuscript is not very well written, missing clear theoretical motivation, clarification of method selection, and interpretation of findings. I have listed several major concerns and some minors as well that need to be addressed to meet the quality standard to be published in Nature Communication.

General response: We appreciated the questions about the methodology and analysis details, and constructive suggestions for the introduction and discussion sections. In preparing the revision, we made appropriate and significant revisions to the *Introduction* and *Discussion* to include discussion of relevant key literature, make our research background, question, and motivation clearer, and provide interpretation of the main findings. Second, we have conducted all the required and additional analyses. We are happy to report that these new analyses confirmed our initial findings and added to our understanding of the neural dynamics of vicarious pain perception. Specifically, we collected new data to demonstrate the stable subjective rating and verify our observation between neural features and subjective ratings. We also conducted a series of analyses to show the robustness of our results and results from data simulation indicated that our decoding accuracy was reliable and insensitive to the number of trials. Third, we provided a comprehensive explanation for our rationale behind each analysis, elaborated on the experimental procedure, clarified the analysis pipelines, and provided additional details about our methodology in the revised *Method* and *Supplementary Materials*.

#Point 1. The introduction needs to be better written to: 1) include more information that inspires and is related to this study. For example, more literatures of anatomical/functional connectivity within this empathy network, similar to Line 106-107, can help readers better understand the motivation of looking into these four regions. 2) the content in each paragraph should have a clear topic. For example, Line 73- 81 and its following paragraph seem to contain redundant information, please consider rephrase the paragraph and make the description consistently and concisely. 3) clarify the findings from previous literatures and how it inspires this study. For example, line 70-71, what does the prior knowledge mean in this sentence "The ACC activity also mediates the top-down modulation of empathic responses to others' pain (e.g., prior knowledge²⁷)."

Response: We thank Reviewer #1 for these very helpful suggestions to improve the *Introduction*. Accordingly, we have re-written the *Introduction* (*Pages 3-6 Lines 51-148*) to make our research background, question, and motivation clearer. Specifically, we **i)** provided a more detailed literature overview of empathy-related neuroscience findings; **ii)** illustrated how these lines of literature motivated research questions of the current study; and **iii)** re-structured the *Introduction* so that the topic of each paragraph was clear to the audience. We hope that you agree that these changes helped to create an improved, clearer *Introduction*.

#Point 2. The task design has some potential pitfalls that raise concerns about whether it is suitable to address the scientific questions tested here.

2a. As authors reported (line 753-754), there were only 10 painful stimuli and 10 non-painful stimuli in this task, that only presented once. The actual trial numbers included in this task would be even smaller, after excluding the trials with incorrect pain judgement and inter-ictal discharges. It is questionable whether such limited trial number ($n < 10$ per condition) has enough statistical power, especially for SVM decoding analyses (see Major point 5 as well).

Response: We thank Reviewer #1 for raising this question, which provides us with the opportunity to clarify this issue. Accordingly, we **i)** presented evidence from the literature justifying the *exceptionally high signal quality of iEEG* as a means to illustrate the feasibility of utilizing limited number of trials (Ball et al., 2009; Parvizi & Kastner, 2018; Mercier et al., 2022); **ii)** provided evidence from previous studies (Shao & Lunetta, 2012; Stelzer et al., 2013; Valizadeh et al., 2019) supporting that our choice of *decoding approach and decoding parameters* can *mitigate* the potential impact of low trial numbers on the neural results (*Page 44 Lines 1236-1241, Lines 1244-1251*); and **iii)** *performed data simulation* to directly examine how changes in trial numbers affected the decoding performance in our dataset, thus confirming the robustness of our findings. Detailed clarifications and results were elaborated below. In addition, we have now explicitly acknowledged the potential limitation of low trial numbers and encouraged future studies to examine empathic neural responses with larger trial numbers in the revised *Discussion* (*Page 28 Lines 759-763*).

i) As pointed out the reviewer, the number of trials in our study was relatively limited compared to fMRI and scalp EEG studies. We made this choice due to *a)* the clinical and hospital constraints associated with intracranial EEG recording and *b)* the practical consideration that patients were donating their time and energy during a challenging period (Parvizi & Kastner, 2018; Mercier et al., 2022). Therefore, we minimized the experimental duration (Mercier et al., 2022) and reduced the number of trials accordingly. However, this potential limitation in trial numbers can be compensated for by the *exceptional signal-to-noise ratio* (SNR) observed in iEEG data compared to fMRI or scalp EEG data (Parvizi & Kastner, 2018; Mercier et al., 2022). For example, when simultaneously recording scalp EEG and iEEG signals, Ball and colleagues (2009) found that the signal quality of iEEG data was **20 to over 100 times better** than that of scalp EEG. Furthermore, recent iEEG studies have demonstrated that, with the high SNR of iEEG data, particularly in functionally specialized brain regions, stimulus-evoked neural responses can be **reliably observed using fewer trials** (Mercier et al., 2022; Lachaux, 2023). For example (**Fig. R1**, adapted from Mercier et al., 2022, Fig. 11), broadband and high-frequency neural activity can be reliably observed with just 2 trials while maintaining consistent patterns across a range of 2 to 59 trials.

Figure R1. Neural responses with different number of averaged trials. The broadband activity (A) and the magnitude envelope of high-frequency activity (40-150 Hz, B) showed similar patterns across a range of 2 to 59 trials (adapted from Mercier et al., 2022, Fig. 11).

ii) When conducting the decoding analysis, we also took into consideration the limited number of trials and carefully selected an appropriate approach and model parameters to mitigate the potential impact caused by a small trial number. We opted for the support vector machine (SVM) approach, which has been demonstrated to be suitable for decoding analysis with a small sample size (LaConte et al., 2005; Fan et al., 2007; Linn et al., 2016). The SVM was found to be *less sensitive to the size of the training sample* compared to other classification algorithms such as multilayer perceptron neural networks, and classification and regression trees (Shao & Lunetta, 2012; Valizadeh et al., 2019). For example, researchers found that there was only around a 3% difference in decoding accuracies achieved by SVM when using 20 samples vs. 800 samples to train the model (Fig. R2, adapted from Shao & Lunetta, 2012, Fig. 2). The SVM has been widely used in studies with limited sample sizes, demonstrating relatively high decoding accuracy even when applied to small datasets (e.g., Bisenius et al., 2017; Levitt et al., 2020; Wu et al., 2023). In addition, we opted for a “*linear*” kernel instead of a “*non-linear*” kernel when constructing the SVM classifier to reduce model complexity and minimize the likelihood of overfitting due to the small sample size (Han & Jiang, 2014).

Moreover, we employed *permutation testing* to assess the statistical significance of the decoding accuracy, which provides robust statistical control over the decoding accuracy, particularly when dealing with small sample sizes (Stelzer et al., 2013; Combrisson & Jerbi, 2015; Varoquaux, 2018). By implementing multiple random shuffling of class labels, permutation testing allows us to estimate chance-level decoding accuracy while *considering sample size* and *controlling for its impact* on the variability of decoding accuracy (Combrisson & Jerbi, 2015). Specifically, we randomly shuffled class labels (painful vs. non-painful) for multiple times while

maintaining other characteristics of the SVM model unchanged, including the sample size (trial number). This generated an empirical distribution of chance-level decoding accuracy that well-captured the variability of decoding accuracy corresponding to the current sample size. We assessed the significance of our observed decoding accuracy based on this permutation distribution. Therefore, we utilized permutation testing as a reliable approach to determine statistical significance and ensure the reliability of our result, especially when dealing with low trial numbers as in the current study.

Figure R2. The classification performance for SVM, multilayer perceptron neural networks (NN), and classification and regression trees (CART) using a range of training sample sizes (adapted from Shao & Lunetta, 2012, Fig. 2).

iii) Besides carefully considering methodological factors and drawing empirical supports from existing literature, we conducted additional data simulation analyses based on our own dataset to assess the impact of trial numbers on decoding accuracy and further validate the stability of our findings. Similar to previous data simulation work (Chu et al., 2012; Nieuwenhuis et al., 2012; Valizadeh et al., 2019), we performed random resampling of trials and channels/channel-pairs to create different data pools differing in sample sizes of training data (ranging from 7-50 trials per class, 200 resampled datasets for each training sample size). For each resampled dataset, we repeated the classification procedure using a linear SVM algorithm combined with five-fold cross-validation to classify painful and non-painful stimuli (the same procedure as our original analysis). We found that the classification performance showed **low variability** across different resampled datasets (relatively small standard errors) in each sample size (Fig. R3). Moreover, the results obtained from our data simulation analysis also suggested **relatively stable** classification performances across different training sample sizes (Fig. R3), as changing the number of trials containing similar information as our original dataset would only result in slight changes (5.5% difference in decoding accuracy between 7 and 50 trials) in the decoding accuracy. These results aligned with previous SVM studies (Shao & Lunetta, 2012; Valizadeh et al., 2019) and further demonstrated that our decoding results was insensitive to variations in trial numbers.

Finally, although the number of trials only slightly influenced the decoding accuracy, we admitted that a larger number of trials will be beneficial and recommended using datasets with larger trial numbers to further examine observed empathy-related neural features in the revised *Discussion* (Page 28 Lines 759-763).

Figure R3. The decoding accuracy for different training sample sizes (trial numbers) in the data simulation analysis of our own dataset. We observed only slight increases in the overall decoding accuracy when the training sample size (per class) increased from 7 to 50 (5.5% difference in decoding accuracy). Error bars show the standard errors.

#Point 2b. A subgroup of participants (16/22) provided their subjective ratings, including empathy strength, intensity of perceived pain in others, own unpleasantness during the post-iEEG period, which were likely collected several days after participants performed the original task recording. How representative these subjective ratings were in reflecting participants' internal states when performing the original task? Author should provide evidence to support the utilization of such subjective ratings that were collected far away from the original experiment. Also, the subjective ratings are collected during the second exposure to the stimuli that might be influenced by the adaptation effect. Therefore, the results in Fig. 6 that that use the neural features collected during real experiment and ratings during post-iEEG period is hard to interpret.

Minor #point 1. It will be helpful to provide a schematic plot, similar to Fig. 1A, to demonstrate the post-iEEG session as well. Also, how much time away from pain judgement and post-iEEG session?

Response: In response to questions regarding the post-iEEG ratings, we **i)** provided a comprehensive explanation for our rationale behind using the post-iEEG rating procedure (Page 31 Lines 842-849); **ii)** elaborated on the experimental procedure used to collect post-iEEG ratings (revised Fig. S11); **iii)** presented our interpretation of the correlation between neural features and subsequent post-iEEG ratings; and **iv)** recruited *an independent cohort of participants* to validate the reliability of ratings collected on different days.

The decisions to employ a pain-judgment task during iEEG recording and to collect subjective ratings related to empathy after, but not during, iEEG-recording were made

based on several considerations. *a*) The majority of previous neuroimaging studies on empathy for other's pain have asked participants to passively view painful (and non-painful) stimuli or perform pain judgment during EEG or fMRI signal recordings and have successfully elicited empathic neural responses, and collected subjective ratings after neural recordings (Avenanti et al., 2005; Fan & Han, 2008; Xu et al., 2009; Morelli & Lieberman, 2013; Chen et al., 2014; Feng et al., 2016; Huang et al., 2023). *b*) Moreover, it has been shown that reporting subjective experience of empathy can evoke intentionally controlled empathic processes instead of voluntarily focusing their empathy on others (de Greck et al., 2012). Therefore, by utilizing a simple pain-judgment task during iEEG recording and the post-iEEG rating procedure, we aimed to capture spontaneous neural processing of empathy for others' pain rather than intentional, response-type modulated empathy and *avoid potential influence* on the empathic neural responses caused by self-report empathic ratings. *c*) This experimental design enabled us to *separately* measure different dimensions of empathy-related ratings. These advantages led us to choose collecting empathy-related ratings after (rather than during) iEEG recording. As pointed out by the reviewer, we also had considered the potential adaptation effect caused by immediate repeated exposure to experimental stimuli. To *minimize potential adaptation effects*, we had decided to collect the empathy-related ratings on separate days. In addition, we provided a schematic plot to elaborate on the procedure of the post-iEEG rating session (revised Fig. S11).

It is important to note that, despite the separate acquisition of our empathy-related neural features and subjective ratings, we indeed observed significant correlations between these neural features and subjective ratings. Furthermore, these correlations were *robustly* observed in different neural features and survived a *stringent* statistical correction for multiple comparisons. Consistent with our observation, such neural-subjective rating associations have been previously reported in studies where participants' ratings were also collected after the neural recordings (Fan & Han, 2008; Han et al., 2008; Chen et al., 2014; Feng et al., 2016; Gonzalez-Liencrez et al., 2016; Huang et al., 2023). Therefore, we believe that the observed associations between neural features and post-iEEG ratings are reliable. We hypothesize that this may be attributed to the relative stability of participants' empathy-related subjective feelings towards targets' pain over time or across different sessions.

We then *empirically tested this possibility* by examining the consistency of empathy-related subjective ratings over time with experimental data. To this end, we recruited a new sample of healthy participants ($n = 29$; 18 males, age = 22.24 ± 7.99 years old) and asked them to provide empathy-related ratings (including subjective ratings of empathy strength, intensity of perceived pain in others, and one's own unpleasantness) for our experimental stimuli on separate days with a relatively long interval. We found that empathy-related subjective ratings demonstrated *considerable stability* across time, providing further evidence supporting our hypothesis about the associations between neural responses and subjective ratings on separate days.

Specifically, we conducted two sets of analyses to assess the similarity between ratings collected on separate days. *a*) We assessed the differences between ratings from two sessions. More importantly, *b*) we examined the cross-stimulus correlation between ratings from two sessions. Each set of analysis was performed among all stimuli (including painful and non-painful stimuli), followed by further examinations of only painful stimuli to check the robustness.

a) We constructed linear mixed models to test the significance of *differences* in rating scores between Time 1 and Time 2 while controlling for individual variations among participants (*Model*: Rating difference (Time 1-Time2) $\sim 1 + (1|\text{subject})$). This analysis suggested that there were *no significant differences* in subjective ratings across the two sessions for each dimension (all stimuli, empathy strength: FDR-corrected $p = 0.108$; perceived pain intensity: $p_{FDR} = 0.711$; unpleasantness: $p_{FDR} = 0.711$). Similar patterns were observed when only painful stimuli were included (empathic strength: $p_{FDR} = 0.942$; perceived pain intensity: $p_{FDR} = 0.888$; unpleasantness: $p_{FDR} = 0.678$).

b) Moreover, consistent with previous studies (Knoll et al., 2015; Lin et al., 2021), we constructed linear mixed models to test for the *correlation* between ratings of Time 1 and ratings of Time 2 while adding participants as a random effect to control for individual variations among participants (*Model*: Rating(Time2) \sim Rating(Time1) + (1|subject)). This analysis showed that subjective ratings of two time points were *highly correlated* with each other (all stimuli, empathy strength: $\beta = 0.92$, SE = 0.02, $t_{479} = 61.08$, $p_{FDR} < 0.001$; perceived pain intensity: $\beta = 0.97$, SE = 0.01, $t_{481} = 92.88$, $p_{FDR} < 0.001$; unpleasantness: $\beta = 0.99$, SE = 0.01, $t_{486} = 84.57$, $p_{FDR} < 0.001$; even when only painful stimuli were considered: empathy strength: $\beta = 0.55$, SE = 0.05, $t_{265} = 10.68$, $p_{FDR} < 0.001$; perceived pain intensity: $\beta = 0.59$, SE = 0.05, $t_{256} = 12.94$, $p_{FDR} < 0.001$; unpleasantness: $\beta = 0.66$, SE = 0.05, $t_{260} = 14.61$, $p_{FDR} < 0.001$).

These results suggested that the empathy-related subjective ratings were relatively stable over time, providing important evidence supporting the reliability of our post-iEEG ratings and relevant findings.

#Point 3. Almost all the results reported here are based on the statistical testing between painful and non-painful conditions. However, to claim that the observed neural features are due to the perception of others' pain, it is also important to demonstrate the significance level within each condition. Because a given neural signature might show significant difference across conditions but not significant within condition. Below are listed several analyses are related to this concern.

Response: This is an excellent point, and we now have tested the significance of each condition. In the revised manuscript, we have made adjustment to the reporting structure of our results. For each analysis, we first reported the results of the contrast between painful and non-painful conditions. We then *separately* presented the statistical significance of the corresponding neural index for painful and non-painful conditions in relation to significant conditional differences.

#Point 3a. Figure 2E-H, is power in painful or nonpainful condition significantly different from the baseline? It is worth extending the analysis window to baseline to ensure that the conditional differences are due to the process of painful vs nonpainful stimuli.

Response: Following suggestions of the reviewer, we examined the significance of spectro-temporal power within each condition by conducting two-tailed one-sample *t*-tests separately for painful and non-painful conditions on clusters with significant conditional differences (revised *Methods*, *Pages 37-38 Lines 1044-1047*). The results revealed that all these clusters exhibited **significant power changes in at least one condition** (results were reported in *Pages 10-11 Lines 250-267*; see *Fig. R4*, revised *Fig. S2*).

Second, per the reviewer's suggestion, we extended the analysis window to include the baseline period (i.e., 200 ms prior to stimulus onset). Our findings indicated that **i**) no clusters with significant conditional differences were observed before stimulus onset and **ii**) all results reported in the original manuscript remained **unchanged** (*Fig. R5*).

These two lines of evidence confirmed that the observed conditional differences were indeed associated with the processing of others' pain (*vs.* non-pain). We thank the reviewer for this suggestion, as it has allowed us to gain a better understanding of empathic neural responses by unveiling distinct patterns to painful and non-painful conditions in different brain regions. Our findings indicated an increase in oscillatory power for the low-frequency bands of AI (*Fig. R4A, B*), ACC alpha band (*Fig. R4C, D*), and amygdala beta band (*Fig. R4E, F*) in the non-painful condition, while the perception of other's pain suppressed the oscillatory power. In contrast, we observed a decrease in beta power for ACC in the non-painful condition, and empathic pain perception enhanced ACC beta power (*Fig. R4G, H*). In the IFG, we found a high-gamma power increase for both painful and non-painful conditions, with a more substantial increase in the painful condition (*Fig. R4I, J*).

Figure R4. *Spectro-temporal power for each condition in the AI, ACC, amygdala, and IFG.* Split-half violin plots indicate the probability density of the averaged power across time-frequency points that exhibited significant conditional differences (significant clusters in Fig. 2e-h) in each condition for the AI low-frequency cluster (A, B), ACC alpha cluster (C, D), amygdala beta cluster (E, F), ACC beta cluster (G, H), and IFG high-gamma cluster (I, J). The boxplots showed the interquartile range of 50% with lower and upper quartile limits at 25% and 75%, respectively. Within the boxplot, the middle line represents the median and whiskers are extended to the most extreme data points that are no more than 1.50 times the interquartile range. * $p < 0.05$, ** $p < 0.01$, *** $p < 0.001$, NS, not significant.

Figure R5. *Spectro-temporal power differences in the AI, ACC, amygdala, and IFG when including the baseline period.* The spectro-temporal power differences between the painful and non-painful condition from 200 ms before to 500 ms after the stimulus onset for the AI (A), ACC (B), amygdala (C), and IFG (D). Significant clusters are highlighted with black contours (corrected $p < 0.01$, 1000 permutations, survived the cluster-based permutation test for multiple comparisons) with insignificant time-frequency ranges presented with transparency. Warmer colors indicate higher t values. Horizontal dashed lines indicate boundaries between frequency bands and Hy represents the high-gamma band.

#Point 3b. Fig. 3A-C, what's the actual correlation value without Fisher-z-transfer? Also, whether this is significant before comparing across conditions The following statement can be hardly supported with the current figure: "AI-amygdala pairs showed significant alpha/beta power correlations in both painful (8-35 Hz; Fig. 3C) and non-painful (8-34 Hz; Fig. 3C) conditions." Is the significance measured simply against to zeros?

#Point 4a. Fig. 3A-C and Line 999-1001, the power correlation analysis was focused on the channels showing significant power difference between painful and non-painful conditions. However, the computation of power correlation analysis itself was done individually on each condition first and then compare across. Then why significant power differences should be a selection criterion here for the power correlation analysis? Also, even with this selection criteria, why the power correlation analysis are not covering theta band, which AI (Fig. 2E) shows significant power difference between painful versus nonpainful conditions?

Response: The points #3b and #4a, both pertaining to the analyses of power correlations (Fig. 3a-c), were collectively addressed here. In summary, **i)** regarding the Fisher-z-transformation for power correlation values, we provided a detailed explanation for the *necessity* of Fisher-z-transformation before conducting further statistical analyses on the coefficients. Additionally, we presented the raw power correlations without Fisher-z-transformation to indicate that this transformation *did not introduce any distortions* to the original result pattern. **ii)** We clarified how we identified frequency bands of interest for the power correlation analysis (revised *Results, Page 13 Lines 319-322, Lines 325-331*). **iii)** To address concerns about single conditions, per reviewer's suggestion, we examined the significance of power correlations within each condition in frequency ranges showing significant conditional differences in power correlations (revised *Methods, Page 40 Line 1118-1123*; results were reported in revised *Results, Pages 13-14 Lines 338-351*).

➤ **i) Regarding the Fisher-z-transformation for the power correlation values**

According to the suggestion of Cohen (2014), we applied the Fisher-z transformation to the power correlation values in order to *satisfy the assumption of normal distribution* for statistical evaluation. Correlation coefficients do not follow a normal distribution; instead, they have a bounded distribution between -1 and +1. Therefore, it is necessary to transform correlation coefficients prior to statistical evaluation (Cohen, 2014). The Fisher-z transform, as the most typical transform for correlation coefficients (Bichot et al., 2015; Caggiano et al., 2016; Mann et al., 2017), can effectively "stretch out" the data range to achieve a broader and more normal-looking distribution (Fig. R6, adapted from Cohen, 2014, Fig. 27.3). This transformation was commonly employed in previous studies for assessing the significance of correlation values (Oehrle et al., 2018; Sterpenich et al., 2021). For illustrative purpose, here we also presented the original power correlation values (without Fisher-z transformation), which exhibited an extremely high level of similarity with those obtained after applying the Fisher-z-transformation (Fig. R7).

Figure R6. The distribution of correlation coefficients without (A) or with (B) Fisher-z-transformation. The distribution of Fisher-z-transformed correlation coefficients (B) was more normal-looking (adapted from Cohen, 2014, Fig. 27.3).

Figure R7. Power correlations between ACC, AI and amygdala without (A-C) or with (D-F) Fisher-z-transformation. Purple (painful condition) and gray (non-painful condition) lines (shadows) indicate the mean (standard error) of frequency-resolved power correlations across all channel pairs. Dashed vertical lines indicate boundaries between frequency bands.

➤ **ii) Frequency bands of interest for power correlation analysis**

Below and in the revision (revised *Results*, *Page 13 Lines 319-322, Lines 325-331*), we have provided further clarification on how we identified the frequency bands of interest for power correlation analysis. Specifically, this identification was based on a comprehensive review of studies investigating the neurophysiological basis of inter-regional interactions and low-frequency coupling, as well as the results of our time-frequency analysis. **i)** Local neural oscillations have been found to be ***closely associated*** with inter-regional functional interactions (Fries, 2005; Donner & Siegel, 2011; Siegel et al., 2012; Snyder et al., 2015). For example, Snyder et al. (2015) found that EEG oscillation amplitude was related to an index of functional connectivity at the

neuronal level (i.e., correlation of spike count across neurons). Siegel et al. (2012) pointed out that local neural oscillations provided a fundamental temporal scaffolding for inter-regional communications and therefore, the frequency of these oscillations may be an important factor in determining the frequency of inter-regional interactions. Therefore, most studies assessing inter-regional interactions focused on frequencies of **functionally relevant** neural oscillations (Oehm et al., 2018; Griffiths et al., 2019; Chen et al., 2021). **ii)** The research on low-frequency coupling predominantly focused on cross-region, **same-frequency** coupling (Kam et al., 2019; Li et al., 2022; Manssuer et al., 2022). This focus was supported by a plausible mechanistic explanation that synchronized oscillations of distinct neuronal groups operating at the same frequency can coordinate the rhythmic opening of their communication windows, thereby facilitating effective inter-regional communications (Fries, 2005). Due to the emphasis on same-frequency coupling, previous investigations primarily measured low-frequency coupling in **overlapping frequency bands** across different brain regions (Kam et al., 2019; Chen et al., 2021).

Based on these two lines of evidence, our power correlation analysis focused on the empathy-relevant frequency bands that overlapped between each pair of brain regions. Specifically, we focused on the overlapping frequency bands that significantly differentiated painful from non-painful conditions in different brain regions, namely the beta band for ACC-amygdala and AI-amygdala pairs, as well as the alpha and beta bands for the ACC-AI pairs (see updated results in Fig. R8, revised Fig. 3a-c).

For illustrative purposes only, we also presented here power correlation results including all low-frequency bands, which further confirmed the robustness of our findings (still surviving correction for multiple comparisons) and not contingent upon our criterion for frequency bands of interest (Fig. R9).

Figure R8. Power correlations between ACC, AI and amygdala in the frequency bands of interest. (A-C) Power correlations (z-scored) averaged across all channel pairs are plotted as a function of frequency for ACC-AI (A), AI-amygdala (B), and ACC-amygdala (C) pairs.

Figure R9. Power correlations between ACC, AI and amygdala for all low-frequency bands. (A-C) Power correlations (z-scored) averaged across all channel pairs are plotted as a function of frequency for ACC-AI (A), AI-amygdala (B), and ACC-amygdala (C) pairs.

➤ **iii) Power correlations within each condition**

We followed the same logic as shown in response to Point #3a. We first identified the frequency ranges with significant conditional differences in power correlations (revised Fig. 3a-c). Subsequently, we assessed the significance of power correlation within each condition for these frequency ranges (revised *Methods*, Page 40 Line 1118-1123). This analysis indicated that all the frequency ranges (with significant conditional differences) were accompanied by significant power correlations in at least one condition (results were reported in *Pages 13-14 Lines 338-351*; see Fig. R10, revised Fig. S4). Specifically, we found significant synchronization between the ACC and AI at 25-32 Hz and between AI and amygdala at 18-24 Hz in the non-painful condition, but this synchronization was inhibited or absent during the perception of others' pain (Fig. R10A-D). While the ACC-amygdala synchronization at 18-22Hz was only observed in the painful condition but not in the non-painful condition (Fig. R10E, F), the ACC-amygdala synchronization at 25-30Hz was only observed in the non-painful condition, not in the painful condition (Fig. R10G, H).

Figure R10. Power correlations between ACC, AI and amygdala within each condition. Purple (grey) split-half violin plots indicate the probability density of the averaged Fisher-z-transformed power correlation values across frequency ranges that significantly differentiated between painful and non-painful stimuli (significant clusters in Fig. 3a-c) in the painful (non-painful) condition for ACC-AI (A, B), AI-amygdala (C, D), and ACC-amygdala (E-H). The boxplots showed the interquartile range of 50% with lower and upper quartile limits at 25% and 75%, respectively. Within the boxplot, the middle line represents the median and whiskers are extended to the most extreme data points that are no more than 1.50 times the interquartile range. *** $p < 0.001$, NS, not significant. $\beta_{low/high}$, low (high) frequency range within the beta band.

#Point 3c. Fig. 4A-C, what phase-amplitude coupling look like for painful or non-painful condition separately? Are they significant within condition as well?

Response: Per reviewer's suggestion, we conducted additional analyses to assess the significance of phase-amplitude coupling (PAC) within each condition (revised *Methods*, Pages 42-43 Lines 1186-1195). Following the same logic as shown in response to Point #3a & 3b, we examined the PAC within each condition in the spectral pairs showing significant conditional differences in PAC. It is important to note that PAC values were indexed by circular-linear correlation coefficients (see **formula (1)**), thus the PAC values were non-negative (i.e., larger than 0, Zheng et al., 2017, 2019). Thus, we could not simply conduct one-sample t -tests to test the significance of PAC values for each condition.

$$\rho_{\phi_a} = \sqrt{\frac{r_{ca}^2 + r_{sa}^2 - 2r_{ca}r_{sa}r_{cs}}{1 - r_{cs}^2}} \quad (1)$$

Where $r_{ca} = c(\cos\phi[n], a[n])$, $r_{sa} = c(\sin\phi[n], a[n])$ and $r_{cs} = c(\cos\phi[n], \sin\phi[n])$, with $c(x, y)$ equal to the Pearson correlation between x and y , $\phi[n]$ equals to the instantaneous phase, and $a[n]$ equals to the instantaneous analytic amplitude.

To solve this issue, we assessed the significance of PAC values for each condition using permutation tests. Within each permutation, we cut the amplitude time series at a random time-point into two parts and then reversed the order of these two parts to generate the permuted amplitude time series (Zhang et al., 2017; Hülsemann et al., 2019). Same analyses of PAC were then applied to the permuted data and the PAC values were averaged across all channel-pairs to generate a distribution of permuted PAC values (200 permutations). The PAC value of painful or non-painful condition was compared to this permutation distribution, which was defined as significant if it was larger than 95% of this permutation distribution (Mukamel et al., 2014; Zhang et al., 2017). Results showed significant PAC between high-gamma amplitude of IFG and the beta phases of ACC/AI/amygdala or alpha phase of AI specifically in the painful condition (results were reported in *Page 16 Lines 425-426*; Fig. R11, revised Fig. S5).

Figure R11. Phase-amplitude coupling (PAC) between ACC/AI/amygdala and IFG within each condition. Bar plots indicate the averaged Fisher-z-transformed PAC values across spectral pairs that significantly differentiated between painful and non-painful stimuli (significant clusters in Fig. 4a-c) in each condition for ACC-IFG (A, B), AI-IFG (C-F), and amygdala-IFG (G, H). Horizontal dashed lines correspond to a threshold for a p value of 0.05. Error bars represent standard errors. ** $p < 0.01$, NS, not significant.

#Point 4b. Fig. 3D-E, is there any evidence to support the exclusion criteria for the Transfer entropy analysis? Authors wrote that (line 324 -326) “situations without significant power correlation indicated that no meaningful functional interaction was involved and further investigation on the information transfer flow was unwarranted.” First, various of methods that quantifies inter-regional functional interactions are phase based (e.g., phase locking value) that does not require strong power correlation. Therefore, power correlation is not necessary for meaningful functional interaction. Second, based on the definition of transfer entropy as the author described (Line 1031 -1033), it is possible to have it without significant increased power correlation. The directionality could be totally independent from this. So why significant power correlation should be a selection criterion here?

Point 10. In Fig. 3D, What’ s the significance level for AI-amygdala within alpha band between painful and nonpainful conditions?

Response: We agree with Reviewer 1’s perspective that power correlations analysis and transfer entropy (TE) analysis could be performed as separate analyses, and we have removed the aforementioned sentence from the revised manuscript. Instead, in the revision (revised *Results*, *Page 14 Lines 353-355*), we clarified that the purpose of employing transfer entropy analysis was a ***follow-up analysis*** to the power correlation analysis. This choice was made because our specific research question in the current study aimed to reveal the direction of inter-regional interactions already exhibiting significant power correlation, rather than addressing a general question about the overall direction of all low-frequency coupling. Therefore, we decided to perform TE analysis on frequency bands demonstrating significant power correlations.

Unlike PAC, which can reflect directional information indicating whether the low-frequency phase of one brain region entrains or modulates the high-frequency amplitude of another region (Jensen & Colgin, 2007; Canolty & Knight, 2010; Zheng et al., 2019), power correlation itself *cannot* convey directional information about inter-regional interactions (Cohen et al., 2014). Therefore, we aimed to uncover ***additional*** directional information through transfer entropy analysis to enhance readers’ understanding of our power correlation results (a follow-up analysis rather than an independent analysis) (revised *Methods*, *Page 40 Lines 1125-1128*). Based on this rationale, we only computed the transfer entropy within the frequency ranges that exhibited significant conditional differences in power correlations (Fig. R12, revised Fig. 3d-f).

In addition, in response to Reviewer 1’s questions, we conducted an exploratory analysis on transfer entropy across all overlapping empathy-relevant frequency bands for each pair of brain regions. This analysis yielded the ***same*** patterns of main effects or interactions (Fig. R13) as those revealed using our ‘power-correlation based’ frequency bands. Moreover, the significant frequency ranges identified in this analysis ***largely overlapped*** with the frequency range of interest in our ‘power-correlation based’ transfer entropy analysis (Fig. R12, Fig. R13). This analysis provided further support for the robustness and reliability of our transfer entropy findings.

Figure R12. Transfer entropy (TE) for ACC-AI (A), AI-amygdala (B), and ACC-amygdala (C). Purple (painful) and gray (non-painful) violin plots indicate the probability distribution of TE values, with inner boxplots showing the interquartile range of 50% (lower and upper quartile limits are 25% and 75%).

Figure R13. Transfer entropy (TE) for all overlapping empathy-relevant frequency bands for each region-pair. Transfer entropy averaged across all channel pairs are plotted as a function of frequency for the main effect of pain in ACC-AI (A), the interaction effect in AI-amygdala (B) and the main effect of pain in ACC-amygdala (C).

#Point 5. Question about the SVM analyses

a. Line 1087-1089, it seems like authors split the dataset for training and testing at the channel level. Does that mean authors were decoding the subjective ratings across participants? If so, I found this very problematic, as the subjective ratings across participants might not consistent with each other, and also the neural signatures might contain individual variations. If not, I found it hard to decode within participant with such limited trial number ($n < 10$ trials per condition, see Major point 2 as well).

Response: We appreciate Reviewer 1 for raising this question, which provides us with an opportunity to clarify the methodological details of our decoding analysis. Please allow us to clarify here and in the revision, that we did not split the dataset at the channel level for training and testing, nor did we preform cross-participant decoding. In the revision (revised *Methods*, Page 43 Lines 1199-1210, Page 44 Lines 1228-1231), we clarified that we employed a two-step data split procedure for the decoding analysis.

➤ The rationale and methodological details of the *first-step* data split: Similar to previous studies (Weston et al., 2000; Vandana & Chikkamannur, 2021), we performed the feature selection *independent* of the learning of classifier parameters. We divided our data set into two parts: i) a *feature-identifying* dataset (70%) to identify neural features that were selective for painful stimuli; ii) a *decoding* dataset (30%) to construct the classification model. This data split was performed at the channel or channel-pair levels. Please note that this split was done *prior to* conducting the decoding analysis, and both decoding model training and testing was carried out using *only* the second dataset. This first-step data split was taken in order to ensure that the neural features were identified independent of the decoding dataset (Kriegeskorte et al., 2009; Pereira et al., 2009).

➤ The rationale and methodological details of the *second-step* data split: We constructed the classification model using the decoding dataset. Within this dataset, we computed the average of each identified feature across channels (or channel-pairs) to obtain the neural features for each trial. The decoding analysis was then performed across trials (not participants) to decode the stimulus type (painful vs. non-painful). Then a second-step data split was implemented: i) training and ii) testing datasets. This involved utilizing a five-fold cross-validation approach, where all trials were divided into five separate folds, with each fold used in turn for testing and the remaining folds for training (Pereira et al., 2009; Quandt et al., 2012). Regarding concerns about the trial number for decoding analysis, we have provided a detailed response in Point #2a.

#Point 5b. Fig. 5B, why does the decoding accuracy drop after integrating more features (>8)? Isn't it supposed to get higher or at least stay the same?

Response: It is not necessarily the case that decoding accuracy will increase or remain the same when incorporating additional features. The '*Curse of dimensionality*' (also known as the Hughes phenomenon, Hughes, 1968) has clarified the relationship between the number of features and decoding performance. According to the Hughes phenomenon, with a given number of training samples, only when the number of features does not exceed the optimal feature number would we observe an improvement in decoding performance with increased feature numbers (Hughes, 1968). However, beyond this point, adding more features would actually deteriorate classifier performance (Hughes, 1968). This phenomenon occurs because an excessive number of features could lead to overfitting given a fixed amount of training data (Hughes, 1968; Pallarés et al., 2018). This overfitting problem would result in increased generalization errors and consequently cause a decline in decoding accuracy.

#Point 5c. Fig. 5C, it is expected to have higher decoding accuracy with more features. If authors tried to emphasize the importance of top 6 features for decoding, a better comparison here should be with models excluding randomly 6 features.

Response: We thank Reviewer 1 for suggesting a comparison between the decoding accuracy when removing top features and removing (same number) random features (revised *Methods*, *Page 45 Lines 1267-1271*). Please note that, as a result of the modifications made in response to *Main Point #3b* and *Minor Point #12*, we updated our decoding results that the top 8 features formed a necessary feature combination (the decoding accuracy dropped to 49.78% and did not significantly differ from chance level, $p = 0.617$, *Fig. 5b, c*). By incorporating the suggested comparison, we demonstrated that the removal of top 8 features led to significantly lower decoding accuracy compared to removing an equal number of random features ($p < 0.001$, reported in the revised *Results*, *Page 19 Lines 488-491*). This finding further supports the importance of this eight-feature combination in vicarious pain perception.

#Point 5d. Fig. 5D, how to understand the negative correlation value between features 2-5 and Empathy strength?

Response: We thank Reviewer #1 for raising this inquiry. Accordingly, we have presented our understanding of the correlations between empathy strength and neural features (revised *Results*, *Page 19-20 Lines 511-530*).

First, please note that we have updated our correlation results based on modifications made in response to *Main Point #3b* and *Minor Point #12* and the implementation of linear mixed-effect models to account for individual variations. In the updated results, only two neural features (ACC alpha oscillations and AI low-frequency oscillations) were reliably associated with empathic strength (revised *Results*, *Page 20 Lines 519-525*, revised *Fig. 5d*). The correlations with other features reported in the original submission did not survive multiple correction under the examination of linear-mixed effect models. Therefore, in the revision, we only focused on correlation results between empathy strength and these two features. The negative association between ACC alpha power/AI low-frequency power and empathy strength indicated that stronger empathic responses were linked to greater suppression of ACC alpha oscillations and AI low-frequency oscillations. These observed correlation patterns were consistent with the patterns of decreased low-frequency power observed in the “painful vs. non-painful” contrast (*Fig. 2e, f*; *Fig. R4A, B*). Together, suppressed low-frequency power in the ACC (alpha) and AI not only facilitated **qualitative differentiation** between others’ pain and non-pain (signaling the presence of other’s pain), but also **quantitatively** tracked the strength of empathic responses.

To aid in the interpretation of the functional meaning of low-frequency power in the ACC (alpha) and AI, we further investigated their association with perceived pain

intensity in others and one's own level of unpleasantness (revised *Results*, *Page 20 Line 532-547*). We revealed a negative association between ACC alpha oscillations and perceived pain intensity (revised *Fig. 5e*), suggesting that higher suppression of ACC alpha power predicted stronger perceived pain. Consistent with this, a recent iEEG study showed that decreased ACC alpha power predicted higher pain states in participants (Shirvalkar et al., 2023). Thus, suppression of ACC alpha power was implicated in encoding both the intensity of pain experienced by oneself (Shirvalkar et al., 2023) and vicariously by others (current study). For the AI, we observed a negative correlation between AI low-frequency oscillations and perceived pain intensity (similarly reported by Soyman et al., 2022), but also a positive correlation with one's own unpleasantness (revised *Fig. 5f*). This suggested that, during empathy for other's suffering, the suppression of AI low-frequency power may enhance other-related processing (e.g., the perception of intensity of others' pain) while inhibited self-related processing (e.g., the generation of stronger personal unpleasantness).

#Point 6. Some analytic approaches used in this paper might not be suitable and needs additional controls.

a. Line 177, Is the similarity simply the correlation of rating scores from patient group and control group? If so, it might be problematic as the ratings here (e.g., Fig. S1) are averaged across all the patients or across all healthy subjects without taking into account the variations across subjects. A better statistical method needs to be used.

b. Line 846-848, what is the pairwise Euclidean distance between the corresponding rating differences for each participant pair? Even the two subject population have comparable age and gender, pairing subjects across groups seems nonsense.

Response: We thank the reviewer for raising these questions, which provide us with the opportunity to explain the rationale behind our analysis and elucidate the methodological details of our analysis on empathy-related ratings here and in the revision. Specifically, **i)** we would like to clarify that we examined whether the empathy-related ratings between patients and healthy participants were comparable by **two sets** of analyses that: **a)** assessed whether patients and healthy participants showed different subjective ratings to vicarious pain and **b)** examined how similar in subjective ratings between patient and healthy participants. For both sets of analyses, we indeed employed analytical methods that **considered inter-individual variations** (revised *Methods*, *Page 33 Lines 913-915*, *Page 34 Lines 929-942*, revised *Results*, *Page 7 Lines 180-190*, *Page 7-8 Lines 197-213*, *Fig. 1e-g*). **ii)** Per reviewer's suggestion, we have replaced the patient-healthy pairing approach with the comparison between different individuals and the normative rating. This modification was similar to established method used in previous studies for assessing similarity between patient and healthy groups (Yang et al., 2020; Soyman et al., 2022). **iii)** Regarding the Euclidean distance index, we agreed with the reviewer that this index might be less intuitive and less easy-to-understand. Thus, we have opted for a **more straightforward** measurement, i.e., the Pearson correlation coefficient (Abrams et al., 2013; Conroy et al., 2013).

- The analysis to test whether patients and healthy participants showed *different* subjective ratings to vicarious pain

Consistent with previous studies (Chen et al., 2019; Legendre et al., 2019), we constructed linear mixed-effect models to examine the group differences in empathy-related ratings *while including participants as a random effect to control for individual variations* (Model: Rating ~ Group + Pain + Group×Pain + (1|participant)). Patients and healthy participants showed comparable subjective ratings in differentiating painful and non-painful stimuli, as indicated by insignificant interactions of pain (painful vs. non-painful) and group (patient vs. healthy) for all dimensions of empathy-related ratings (all p -values for interaction > 0.05, F-tests on linear mixed effects models; detailed statistical information in the revised Table S1). Moreover, comparable subjective ratings between patients and healthy participants were observed in each stimulus pair (all FDR-corrected ps for interaction > 0.05).

- The analysis to indicate how *similar* in subjective ratings between patient and healthy participants

Next, we examined the similarity in subjective ratings across stimuli between patient group and healthy group. We have taken into account Reviewer 1's concerns regarding the pairing of participants across groups. Accordingly, we have employed an alternative approach to assess the similarity in the empathy-related subjective ratings between patients and healthy participants. This method has been previously used to assess the similarity between patient and healthy groups (Yang et al., 2020; Soyman et al., 2022). Specifically, the average rating from all healthy participants was considered as the normative rating (Libkuman et al., 2007; Soyman et al., 2022). To account for individual variations in subjective ratings, we calculated the similarity between *each* patient's or *each* healthy participant's ratings and the normative ratings. We conducted permutation tests (Chen et al., 2016; Hyon et al., 2020) and found that ratings from patient individuals and healthy individuals were similar to the normative ratings for each empathy-related measure (patient-normative similarity: all ps < 0.01, healthy-normative similarity: all ps < 0.001, survived FDR correction for multiple comparisons, no matter when all stimuli or only painful stimuli were considered; Fig. R14, revised Fig. 1e-g, detailed statistical information in the revised Table S3). Moreover, the patient-normative similarity was comparable to the healthy-normative similarity (all ps > 0.05 regardless of all stimuli or only painful stimuli were included; Fig. R14, revised Fig. 1e-g, detailed statistical information in the revised Table S3), indicating that patients and healthy participants indeed provided similar empathy-related ratings on the experimental stimuli.

Figure R14. The similarity between ratings of patients and healthy participants and the normative ratings across painful stimuli for empathy strength (A), perceived pain intensity (B), and unpleasantness (C). The split-half violin plots show the probability density of the data and the left boxplots show the 25th, 50th, and 75th percentiles with whiskers extended to the most extreme data points that are no more than 1.50 times the interquartile range. ** $p < 0.01$, *** $p < 0.001$, NS, not significant.

#Point 7. The discussion section is supposed to summarize the key findings in the paper, provide reasonable interpretation, and inspire additional research directions. However, it is really hard for readers to learn from the reported results in this paper and how that could strengthen our knowledge about the empathy network.

Response: We appreciated the helpful and constructive suggestions for the discussion section and made significant and appropriate revisions to the *Discussion* accordingly. In the revised *Discussion* (Pages 21-28 Lines 561-776), we summarized the main findings of empathy-related oscillatory power and inter-regional interactions (paragraphs 1 and 2), presented our understanding of observed regional oscillatory power changes (paragraphs 3 and 4), indicated the importance of observed inter-regional communications in unraveling the functional organization of empathy network (paragraphs 5-9), and discussed other considerations as well as pointed towards new directions for future studies (paragraphs 10-11).

#Point 7a. For example, Line 620-624, author makes claim that “increased ACC beta oscillations may support the top-down modulation of vicarious pain perception by prior knowledge or experience while decrease AI beta oscillations may associate with bottom-up affective responses triggered by perceived pain in others.” How does this hypothesis fit into this task?
 e. Line 680- 683, based on what results, that authors conclude that IFG might play a role in understanding the target’s action itself and automatic action simulation? Also, how does this connect to the perception of others’ pain?

Response: We thank Reviewer #1 for bringing these points to our attention. We agree that these arguments cannot be directly inferred from our findings or align with the current task, as our experimental task did not differentiate between top-down and bottom-up processes nor directly measure the understanding processing. Accordingly, we have now removed these claims.

#Point 7b. Line 641 and 642, then what is the functional role of amygdala in perception of other's pain? Especially with such late response

Response: We thank Reviewer #1 for raising this inquiry. Accordingly, we have discussed the possible functional role of this late amygdala response in the revised *Discussion* (Page 24 Lines 630-655).

Page 24 Line 630-655: “While animal electrophysiological studies have documented the important role of the amygdala in empathic responses²⁸⁻³⁰, the fMRI evidence for the involvement of the amygdala in human empathy has been inconsistent^{9,56,57}, leaving whether the amygdala is crucial for the processing of human empathy for pain an open question. Our iEEG findings fill the gaps between animal electrophysiological findings and human fMRI studies by providing electrophysiological evidence for the crucial and sophisticated role of amygdala in the perception of vicarious pain. Specifically, the late decrease in beta oscillations in the amygdala, beta-band-coordinated coupling between ACC and amygdala, and cross-frequency coupling between IFG and amygdala were found to be pivotal for decoding vicarious pain perception. Interestingly, we showed that the amygdala oscillatory activity responded to others' pain later than the ACC, AI, and IFG. The late amygdala oscillation observed in the current study was less expected, but was consistent with findings of animal research showing that amygdala neurons responded to cues associated with electric shocks to another mouse later than neurons in the ACC²⁸. The late amygdala beta suppression and inter-regional communications with other regions likely reflected the processing of late-stage information by the amygdala, such as the integrated neural representation of others' pain resulting from interactions with other regions. This processing aids in differentiating from others' emotional states and generating one's own negative emotional responses. Two lines of neural evidence supported this possibility. First, previous studies on animal and humans have identified the amygdala and its interaction with other brain regions (e.g., ACC) as important neural features that facilitated the subjective experience of negative affective responses, including pain and fear^{28,29,58,58}. Second, the amygdala beta suppression has been linked to negative emotional states, such as heightened levels of anxiety⁶⁰ and more severe depressive symptoms⁶¹. It would be valuable for future research to directly test this possibility.”

Point #7c. Line 645 to 661, what information the inter-regional communication can provide in addition to the power increase/decrease?

Point #7d. How to understand the results from the directionality analyses?

Minor Point #11. It seems like transfer entropy is larger for Non-painful compared to painful conditions? What does that mean?

Response: We're very grateful to Reviewer #1 for these questions, which motivate us to think further about the significance and interpretations of the inter-regional

communication. Accordingly, we elaborated on how we understood the findings of inter-regional communications, and how these findings furthered our understanding of the functional organization of the empathy neural network below and in the revised *Discussion* (Pages 24-27 Lines 657-723).

Pages 24-27 Lines 657-723: “The discovery of empathy-related inter-regional communications has advanced our mechanistic understanding of the functional organization of the empathy network. Previous iEEG studies on empathy^{26,27} and our findings on regional power changes (e.g., ACC alpha power suppression and IFG gamma power increase) have reflected cognitive operations within individual brain regions. However, these findings did not inform us whether the regions within the empathy network responded independently or interact with each other⁶². The current study highlighted rapid inter-regional communications within the human empathy network, indicating that empathic responses cannot solely be attributed to isolated operations within single brain regions but also require dynamic interactions across multiple regions involved in empathy processes.

Furthermore, we identified two potential pathways for inter-regional communication: beta-band-coordinated coupling between ACC, AI, and amygdala; and cross-frequency coupling between high-gamma IFG and beta ACC/AI/amygdala. These two distinct inter-regional communication mechanisms support cross-spatiotemporal organization of the empathy network. Previous animal studies have identified critical functional roles of the AI-amygdala³⁰ and ACC-amygdala^{28,29} circuits in observational learning and the formation of empathic pain. However, to date, it remains unclear how rapid communications between empathy-relevant brain regions support empathic responses in humans. Our iEEG results provided electrophysiological evidence for the engagement of these two circuits in human empathy, suggesting the ACC/AI-amygdala circuit as an evolutionarily conserved mechanism of empathy. Moreover, we identified a new mode of inter-regional communication related to empathy — cross-frequency coupling between high-gamma IFG and beta ACC/AI/amygdala — which points towards new directions for future investigations into empathy-related circuits.

Moreover, the current study provided evidence for both increased functional interactions (e.g., enhanced coupling between ACC beta phase and IFG gamma amplitude) and decreased inter-regional communications (e.g., attenuated beta coupling between ACC and AI) within the empathy network. These patterns highlighted rapid information flow among different brain regions to coordinate diverse processes of empathy. When processing others’ pain, the brain needs to not only enhance functional interactions between specific empathy-related regions (e.g., between ACC and IFG), potentially facilitating their coordination and information integration, but also appropriately suppress certain inter-regional communications in order to reduce mutual distractions and increase functional specialization of relevant brain regions (e.g., ACC and AI).

Taking the ACC and IFG as an illustrative example of increased functional interactions. In terms of regional oscillations, while low-frequency oscillations in the ACC responded to perceived pain, we observed a distinct spectro-temporal profile of the IFG with early high-gamma oscillatory activity. Previous studies have suggested that low-frequency oscillations are entrained across distant regions and subserved long-range interactions, whereas high-gamma activity mainly reflects local neuronal responses and serves as a signature of local encoding⁶³. This suggested fundamentally distinct engagement and functions of ACC low-frequency oscillations and IFG high-gamma oscillations in empathic responses. The cross-frequency coupling mechanism, i.e., phase-amplitude coupling with low-frequency phases modulating high-frequency amplitudes, serves to convert information from slow timescales into fast local processing and integrates functions across spatio-temporal scales^{39,40}. Thus, the finding of the enhanced modulation of IFG high-gamma amplitude by the ACC beta phase suggests a possible mechanism underlying information integration from ACC and IFG, echoing previous findings on anatomical connections between ACC and IFG^{64,66}.

In contrast, we observed weaker functional communications between the ACC and AI, which may be associated with a reduced bidirectional exchange of information related to perceiving others' emotional states and generating personal emotional responses. This may facilitate functional specialization and prevent emotional responses from biasing the evaluation of other's pain⁶⁷. This observation may be supported by the bi-directional structural connections between the ACC and AI⁶⁸, and was consistent with a recent iEEG study showing the important role of bidirectional connectivity between the ACC and AI in emotional processing⁶⁹. Thus, these findings on empathy-related inter-regional communication aid in understanding how the brain generates empathic responses towards others' pain.”

Point #7d. How to understand the results from the directionality analyses?

Response: We thank Reviewer #1 for raising this question. Accordingly, to help the audience better understand the directionality results, we summarized the information conveyed by the results of directionality analysis in the revised *Results* section (*Page 14 Lines 362-376*) and (briefly) discussed how these results aided in understanding the patterns observed in low-frequency coupling analysis and whether they were consistent with previous anatomical and fMRI findings in the revised *Discussion* (see detailed illustrations in the responses to *Major Point #7c*, reported in *Pages 26-27 Lines 714-721*). Please note that the directionality analysis served as a supplementary analysis to the power coupling analysis and was not the primary focus of the current study. In addition, we lacked direct evidence to indicate the functional meaning of the observed directionality. Therefore, we chose not to extensively elaborate on these results in order to maintain emphasis on our main research questions.

revised *Results* (*Page 14 Lines 362-376*): “We detected a significant main effect of pain in ACC-AI at 25-32 Hz ($F_{1, 233} = 30.04$, $p = 1.10 \times 10^{-7}$, $\eta_p^2 = 0.11$, **Fig. 3d**),

suggesting reduced beta information transfer in both directions from-ACC-to-AI and from-AI-to-ACC. Interestingly, the effect of pain was direction sensitive in AI-amygdala at 18-24 Hz as indicated by a significant Pain \times Direction interaction on TE values ($F_{1, 299} = 13.67, p = 2.59 \times 10^{-4}, \eta_p^2 = 0.04$, Fig. 3e). Perception of painful stimuli specifically suppressed information transfer from the amygdala to AI (amygdala-to-AI: $t_{299} = -3.25, p = 0.001$, Cohen's $d = -0.19$, 95% CI: -0.009, -0.002; AI-to-amygdala: $t_{299} = -0.48, p = 0.629$, Cohen's $d = -0.03$, 95% CI: -0.004, 0.003; Fig. 3e). For ACC-amygdala pairs, we found a significant main effect of pain at 18-22 Hz ($F_{1, 71} = 20.14, p = 2.72 \times 10^{-5}, \eta_p^2 = 0.22$, Fig. 3f), reflecting enhanced information transmission between ACC and amygdala during the processing of others' pain, irrespective of the specific directions. For the interactions between the ACC and amygdala at 25-30 Hz, no significant results were found for the main effect of pain or the interaction effect (Main effect of pain: $F_{1, 71} = 0.33, p = 0.569, \eta_p^2 = 0.01$; Interaction: $F_{1, 71} = 1.83, p = 0.191, \eta_p^2 = 0.03$). ”

Point #8. Authors should be mindful of using terms like “first” when describing the results. For example, Line 566 -568, authors claim that “our study is the first attempt to integrate region-specific neural oscillations and inter-regional interactions to decode vicarious pain perception, enabling us to characterize how these neural features jointly contributed to vicarious pain perception”. Also, Line 1028-1030, “our study is among the first to examine the electrophysiological basis for effective connectivity within human empathy network, and no clear assumptions on interaction pattern existed.”

Response: We thank Reviewer #1 for this reminder. In the revision, we have removed these terms from the manuscript.

Minor:

2. $XX \pm XX$, standard deviation vs. standard error mean needs to be stated in the manuscript.
3. Line 142 to 144, it is helpful to show the actual value of response accuracy and response time along with the t-test.
4. I assume that the data was collected with Chinese instruction. We appreciate that the authors have translated the instruction to English (Figure 1A). It is also informative to show the original task instruction in a supplementary Figure.
5. Maybe Colorbar label in Figure 2E-H should not be Power, but a power difference (t value)?
6. Figure 2I-L, is the plotted power here normalized? The value seem to be around zero before $t = 0$. If so, please update the y axis to normalized power instead.
7. Fig. 2J, the figure title “Beta band” seems to be smaller than the rest.
8. Are the results (Fig. 3 and Fig. 4) only include electrode pairs within the same subject and same hemisphere? If so, please clarify in the text.
9. The thickness of the lines in Fig. 3A-C seems not very consistent, is this on purpose?
11. It seems like transfer entropy is larger for Non-painful compared to painful conditions? What does that mean?
13. Line 787- 789, please provide details for how the arousal assessment was done in patients. Categorically measured or continuously measured?

Response: We are very grateful to the reviewer for these detailed suggestions and the time you put into the manuscript, highly appreciated. We took these suggestions and have now accordingly addressed these points in the revision. Specifically, we have now included the statistical information (point #2; *Page 6 Lines 157-162, Page 7 Lines 172-175*); the values of accuracy and response time (point #3; *Page 6 Lines 157-162*); provided the original Chinese instructions in the revised *Methods* (point #4; *Page 30 Lines 832-834*); the information about channel-pairs (point #8; *Page 36 Lines 1010-1011, Page 39 Lines 1103-1106*); provided an understanding of the transfer entropy pattern (point #11; see detailed illustrations in the responses to *Major Point #7c,d*; *Page 14 Lines 362-364, Page 26-27 Lines 714-721*); details arousal measurement (point #13; *Page 31 Lines 860-864*). We have also revised the colorbar, label of y-axis, font of Fig. 2 (points #5-7; *revised Fig. 2*) and the line thickness of Fig. 3 (point #9; *revised Fig. 3*) according to the reviewer's suggestions.

Point #3 (revised *Results, Page 6 Lines 157-162*): “Patients showed no significant differences in response accuracies (response accuracy difference: $5.91\% \pm 4.73\%$, $t_{21} = 1.26$, $p = 0.221$, Cohen's $d = 0.27$, 95% *CI*: -0.07, 0.27) and response times (RTs, RT difference: -0.02 ± 0.09 , $t_{21} = -0.16$, $p = 0.874$, Cohen's $d = -0.03$, 95% *CI*: -0.06, 0.05) between painful and non-painful conditions, suggesting comparable attentional engagement and motor responses to painful and non-painful stimuli.”

Point #4 (revised *Methods, Page 30 Lines 832-834*): “participants were asked to indicate whether the person in each picture experienced pain or not (as specified by in Chinese instruction: “请您判断图片中的人是否感到疼痛”) ”

Point #8 (revised *Methods, Page 36 Lines 1010-1011, Page 39 Lines 1103-1106*): “Note that channels from both hemispheres were collapsed to improve statistical power^{33,34,37}.” “To eliminate the potential influence of individual differences, we only considered pairs of channels within each participant (i.e., the two channels of each pair from the same participant) and included participants with at least one channel pair.”

Point #13 (revised *Methods, Page 31 Lines 860-864*): “In addition, to confirm that the neural findings did not result from possible differences in arousal levels between painful and non-painful stimuli, patients also provided ratings of arousal level for each stimulus (“How intense is your emotional response induced by this picture”; 0 = extremely calm, 100 = extremely strong).”

Point 12. What is the difference between Fig.4 and Fig. S6? What is varying vs fixed bandwidth? Also, why the results between IFG and amygdala are so different between Fig 4 and Fig. S6?

Response: In response to this point, we first clarified that the only difference between Fig. 4 and Fig. S6 lied in the bandwidth used to filter the amplitude signal in the phase-amplitude coupling analysis (PAC). Specifically, Fig. 4 utilized fixed-bandwidth filtering while Fig. S6 used varying-bandwidth filtering. Below, we clarified the

primary differences between these two filtering approaches and how these differences contributed to variation in PAC result patterns.

To further examine *whether these approaches yielded qualitative differences*, we tested whether the significant cluster identified using the fixed-bandwidth method remained significant with the varying-bandwidth method, and vice versa. We found similar amygdala-IFG PAC patterns using both methods. Notably, spectral clusters identified as significant with one approach remained significant when testing the conditional differences by averaging the spectral pairs with another approach (Fig. R15). In addition, previous studies have indicated that the varying-bandwidth method *outperforms* the fixed-bandwidth method in investigating PAC (Berman et al., 2012; Aru et al., 2015; Zandvoort & Nolte, 2021). Therefore, based on its greater reliability and recommendation within relevant literature, we decided to only report results obtained through varying-bandwidth filtering in the revised manuscript (revised *Methods*, Page 42 Lines 1165-1174, revised Fig. 4a-c). This reporting strategy would also help prevent potential confusion arising from inconsistency.

➤ Fixed-bandwidth vs. varying-bandwidth filtering in PAC

These are two approaches for filtering the amplitude signal in the PAC (Berman et al., 2012; Aru et al., 2015). Specifically, the fixed-bandwidth method uses a constant bandwidth 2 Hz (Stangl et al., 2021), while the varying-bandwidth method employs a bandwidth that varies according to the coupling phase frequency (a frequency window of \pm phase frequency; Berman et al., 2012; Zandvoort & Nolte, 2021). Here, we take the example of an amplitude-frequency bin centered at 90 Hz to illustrate the difference. With the fixed-bandwidth approach, the band-pass filter only allows components in the frequency between $90 - 2 = 88$ Hz and $90 + 2 = 92$ Hz, regardless of the coupling phase frequency. In contrast, with the varying-bandwidth approach, the specific bandwidth used for filtering the amplitude signal depends on each corresponding phase frequency.

For example, when centered at a phase-frequency bin of 30Hz, the band-pass filter passes components in a range between $90 - 30 = 60$ Hz and $90 + 30 = 120$ Hz; but when centered at a phase-frequency bin of 20Hz, it passes components in a range between $90 - 20 = 70$ Hz and $90 + 20 = 110$ Hz. Thus, the varying-bandwidth (vs. fixed-bandwidth) approach included a much wider range of frequency components into the amplitude signal, which could account for the differences in the PAC pattern.

Figure R15. Phase-amplitude coupling (PAC) between amygdala and IFG when employing the fixed-bandwidth (A/D) or varying-bandwidth method (B/C). The spectral pairs deemed significant in the fixed-bandwidth (A) or the varying-bandwidth (C) method remained significant in the varying-bandwidth (B) or the fixed-bandwidth (D) method if we averaged the spectral pairs in the significant clusters. Error bars show the standard errors. ** $p < 0.01$, *** $p < 0.001$.

Point 14. Line 793-794, author mentioned that they have recruited the gender-matched healthy control group for the study, which has 22 subjects in total with 9 males while the patient group has 13 males. So it's not entirely matched. Please make sure the consistency of description.

Response: We appreciate Reviewer #1 for bringing up this concern. We acknowledge that our patient sample and healthy sample were not identical, and we have removed any inappropriate statements from the manuscript accordingly. However, we want to emphasize that despite not being identical, our healthy sample and patient sample had comparable gender and age distributions, as evidenced by insignificant differences in age (age: $t_{42} = -0.81$, $p = 0.424$, Cohen's $d = -0.24$, 95% CI: -8.91, 3.82, two-sided two-sample t -test) and gender (gender: $\chi^2(1) = 0.82$, $p = 0.366$, two-sided Pearson's Chi-square test of independence) between the two groups. In the revised manuscript, we rephrased our statement ("we recruited a healthy participant sample whose gender distribution and age distribution were comparable to those of the patient sample") and provided the above-mentioned statistical information to support this statement (Page 32 Lines 873-877).

Point 15. Authors mentioned that the spikes (defined as >100uV changes between consecutive samples) were removed from original data. Spikes can also refer to single neuron activation. It might be less confusing to change it to a different term.

Response: This is a very valid point, thanks. Accordingly, we have replaced "spikes" with "jumps between consecutive data points larger than 100 μV " (Page 35 Lines 963-965).

Point 16. Line 1015 and Line 1019, what is the cluster-based permutation here, each trial has only one correlation coefficient?

Response: We thank Reviewer #1 for raising this question. Accordingly, we have detailed the cluster-based permutation test for power correlation analysis in the revised *Methods* (Page 40 Lines 1114-1118). Similar to previous iEEG studies, for each frequency and each channel-pair, every trial yielded a correlation coefficient between the two corresponding channels' time series. Within each condition, the correlation coefficients were Fisher-z-transformed and averaged across trials. In the cluster-based permutation test, within each frequency, we randomly shuffled the labels of painful and non-painful conditions for each channel-pair. This enabled us to calculate a t -value between the shuffled conditions across all channel-pairs using paired- t tests. Frequencies with uncorrected p -value (< 0.05) were then clustered based on spectral adjacency (Maris & Oostenveld, 2007). We calculated the sum of t -values for each cluster as its 'mass' and recorded the most extreme cluster mass among all clusters. These steps were repeated 1000 times to generate a null distribution of differences between conditions. Finally, significant clusters (corrected $p < 0.01$) were identified by comparing the cluster of the true data against this null distribution.

➤ **Responses to Reviewer #2's comments:**

The paper revolves around the neural underpinnings of empathy for pain using intracranial electroencephalography (iEEG) to elucidate with high-resolution the spatio-temporal profiles of neural oscillatory activity and inter-regional communications within the empathy network during the perception of others' pain. While the research protocol for testing empathic reactivity to pain is a standard one, the technique used is a sophisticated one and the analytic approach state-of-the-art. While this paper may advance our understanding of the neural dynamics that underpins empathy for pain there are several points that need to be clarified. A point by point list of comments is provided below.

General response: We thank Reviewer #2 for the positive evaluations of our manuscript. We have taken the opportunity – both here below and in the revised manuscript – to address the constructive suggestions.

Point 1. The introduction does not provide a detailed overview of the importance of empathy in social interactions and its neural basis. In particular, the paper does not provide a thorough and well-defined overview of the existing literature that utilizes intracranial electroencephalography (iEEG) to explore empathy for pain. Notably, the works of Soyman et al. (2022) and Mo et al. (2022) are absent in the manuscript, leading to an incorrect assertion of the technique's novelty within the field. This is a major point of weakness. While this may be considered as a lack of scholarship, it may correspond to a simple overlook. However, discussing the above papers is fundamentally important for comparing the findings of the present research with what is already known.

Response: We greatly appreciate the suggestions provided by Reviewer 2, which are extremely helpful in improving the *Introduction* section of our manuscript. In the revised *Introduction*:

i) The first paragraph have now placed greater emphasis on the ***significance of empathy*** in social interactions (*Page 3 Lines 51-57*): “Empathy enables us to quickly perceive and share the experiences and feelings of others, rendering it a powerful catalyst for successful social interactions and prosocial behavior¹⁻³. This ability not only enhances our understanding of other individuals’ affective states but also equips us with the foresight to predict their future actions, empowering us to take appropriate actions within specific social contexts⁴. Particularly in situations where we witness others’ suffering, pain empathy grants us the capacity to vicariously experience their pain and motivates us to provide help^{2,3}.”

ii) We have now **reviewed** previous iEEG studies related to empathy, including Hutchison et al., 1999; Mo et al., 2022; Soyman et al., 2022 (*Page 4 Lines 98-105, Page 5 Lines 118-122*).

On one hand, we underscored that these iEEG studies have yielded **valuable insights** into the neural basis of empathy. Specifically, Hutchison et al. (1999) conducted single-neuron recordings of ACC and found that single neurons in the ACC responded when participants witnessed others’ fingers being pin-pricked, providing

electrophysiological evidence of the engagement of ACC in the processing of others' pain. Soyman et al. (2022) adopted iEEG to characterize the electrophysiological responses of insula to others' pain, with broadband activity of insula encoding other's pain intensity. Mo and colleagues (2022) examined the recurrence-related characteristics of the default-mode network during resting-state and the relationship with a questionnaire score of empathy trait.

On the other hand, we acknowledged their **limitations** such as focusing exclusively on a single brain region (the ACC in Hutchison et al., 1999; AI in Soyman et al., 2022), or examining empathy-irrelevant, non-task-specific resting-state instead of task-related and empathy-specific neural activity (i.g., associating empathy-related traits with resting-state in Mo et al., 2022).

iii) Furthermore, we clarified that **significant gaps** existed between findings from these studies and the understanding of multi-areal neural dynamics in vicarious pain perception. These gaps served as **motivation** for our current study, which aims to investigate both temporal order and spectral characteristics of neural oscillatory activity within empathy-related brain regions, as well as the rapid-scale information flow among these regions through recording iEEG signals at multiple empathy-related regions with millimeter and millisecond resolutions during perception of others' pain.

Point 2. The methods section detailing the use of iEEG in epilepsy patients is well-described. However, additional information regarding patient selection criteria, ethical considerations, and potential confounders would enhance the methodological robustness of the study. Moreover, it is not clear to me whether 6 patients were excluded because failed to complete the post iEEG session or whether they did not have electrodes implanted in the regions of interest. Both sentences are written in the manuscript, and it is hard to understand why 6 patients were excluded from the data (or if in different moment different 6 patients were excluded). Moreover, it might be interesting to indicate for each patient where the electrodes were implanted (e.g., patient 1, X electrodes in AI, X electrodes in ACC, etc).

Response: We thank Reviewer #1 for raising these questions, which provide us with the opportunity to clarify these methodological details in the revision. Accordingly, we have now provided details regarding patient exclusion criteria (*Page 29 Lines 788-790, Page 29 Lines 794-801, Page 31 Lines 842-853*), ethical considerations (*Page 29 Lines 778-786*), and the measures we implemented to exclude potential confounders (*Page 29 Lines 790-792, Page 29 Lines 792-794, Page 30 Lines 821-822, Page 6 Lines 157-162, Page 21 Lines 549-557*) in the revised *Methods* section. We also provided the channel information of each patient in Table R1 (Supplementary Data 3).

- Regarding the number of patients included in each analysis, we clearly stated the criteria and the number of patients in each analysis in the revised *Methods*:
Page 29 Lines 788-790: “Data were recorded from 29 epilepsy patients who were implanted with intracranial depth electrodes and were undergoing intracranial EEG

monitoring to localize the seizure onset zone for potential surgical resection.”

Page 29 Lines 794-801: “The patient selection was based on two inclusion criteria: i) having electrodes in the ACC, AI, amygdala, or IFG contralateral to or outside of the epileptogenic zone; and ii) achieving a response accuracy above 50% in the pain judgment task. Based on these criteria, one patient was excluded due to a low response accuracy (45%) in the pain judgment task, and six patients were excluded because no electrodes were implanted in the regions of interest. The remaining 22 patients were included in the behavioral and neural analysis of the pain judgment task (13 males, age = 25.73 ± 2.07 years old).”

Page 31 Lines 842-853: “Similar to the majority of previous neuroimaging studies^{21,72,73}, we invited all patients to a post-iEEG session to measure the empathic strength and other empathy-related subjective ratings to perceived pain in others after the iEEG recording. This setting (post-iEEG rating procedure) could avoid potential influence on the empathic neural responses in the pain judgment task caused by self-report empathic ratings (e.g., avoid evoking intentionally controlled empathic processes⁷⁴), and enabled us to separately measure different dimensions of empathy-related ratings (see the procedure of post-iEEG session in Supplementary Fig. 11). No data were excluded, but the subjective ratings of six patients were missing as the six patients were unwilling to or failed to complete the post-iEEG session. ”

- We also clarified relevant ethical considerations of the current study:

Page 29 Lines 778-786: “Electrode localizations were exclusively determined by clinical needs. We prioritized and maintained the integrity of clinical care during conducting the current study. All patients provided informed consent after the experimental procedure had been fully explained, and were acknowledged their right to withdraw at any time during the study. ”

- In the revised *Methods*, we also clarified the potential confounders we have considered in the current study, including other neurological disorders, pain medications, pain levels, physical characteristics of stimuli, behavioral performance, and arousal levels related to the experimental stimuli:

Page 29 Lines 790-792: “All participants recruited in the current study had no history of psychiatric disorders, head trauma, or encephalitis.”

Page 29 Lines 792-794: “Patients did not take pain medication several hours prior to the iEEG recording of the pain judgment task and were not experiencing any physical pain during the iEEG recording.”

Page 30 Lines 821-822: “The luminance, contrast, and color of the painful and non-painful stimuli were matched.”

Page 6 Lines 157-162: “Patients showed no significant differences in response

accuracies (response accuracy difference: $5.91\% \pm 4.73\%$, $t_{21} = 1.26$, $p = 0.221$, Cohen's $d = 0.27$, 95% CI : -0.07, 0.27) and response times (RTs, RT difference: -0.02 ± 0.09 , $t_{21} = -0.16$, $p = 0.874$, Cohen's $d = -0.03$, 95% CI : -0.06, 0.05) between painful and non-painful conditions, suggesting comparable attentional engagement and motor responses to painful and non-painful stimuli.”

Page 21 Lines 549-557: “It should be noted that we conducted additional control analyses to exclude the possibility that our neural findings resulted from potential differences in arousal levels between painful and non-painful stimuli. We asked patients to provide ratings of the arousal level for each stimulus after the iEEG recording. We examined the association between the arousal level and ACC alpha oscillations/AI low-frequency oscillations but did not find any significant results ($ps > 0.05$; no significant results even when we checked for spectro-temporal power at all time-frequency points in each brain region, Supplementary Fig. 9). Therefore, the observed neural effects cannot be attributed to potential differences in arousal levels between painful and non-painful stimuli.”

Point 3. A main concern is whether the authors have data about areas which are not involved in the empathic network to control for general activities (see Soyman et al., for a comparison between the insular electrodes and “random” electrodes). It seems that the authors use signal from white matter as reference but there isn't an actual comparison between areas which shouldn't be implicated in processing empathic responses.

Response: This is a very valid point, thank you. Accordingly, we adopted a similar approach as described in Soyman et al. (2022) to compare the observed effect (clusters with significant conditional differences in Fig. 2e-h) with that obtained from randomly selected channels located outside the empathic network (i.e., not within the four ROIs) (*Page 38 Lines 1049-1062*). We ensured an equivalent number of channels within each ROI and of ‘random’ channels. Consistent with our previous analyses, we conducted paired- t tests to examine the conditional power differences among the random channels. This procedure was repeated for 1,000 times, resulting in 1000 t -values used to construct the null distribution. We found that the observed conditional effects of all these clusters were significantly stronger than the conditional differences calculated based on random channels (ACC alpha cluster: $p = 0.011$, ACC beta cluster: $p < 0.001$; AI low-frequency cluster: $p < 0.001$; amygdala beta cluster: $p < 0.001$; IFG high-gamma cluster: $p < 0.001$). The analysis further confirmed that the empathy-related spectro-temporal power observed in different brain regions indeed reflected region-specific (rather than general activity) spectral patterns.

Point 4. Related to this, the introduction lacks the rationale of selecting AI, ACC, IFG and the amygdala and no other areas, such as the motor or the somatosensory cortex or the mCC (Fallon et al., 2020).

Response: We thank Reviewer 2 for this suggestion. Accordingly, we clarified here, as

well as in the revised manuscript (revised *Introduction*, *Page 4 Lines 81-88*; revised *Results*, *Page 10 Lines 236-238*) the rationale of selecting AI, ACC, IFG, and amygdala as the regions of interest. This choice was made based on two aspects of considerations:

i) We reviewed previous neuroimaging studies on empathy for other's pain to identify empathy-related brain regions (which served as a basis for selecting ROIs in the current study). The AI, ACC, IFG, and amygdala, along with other brain regions (such as the somatosensory cortex and the mCC mentioned by the reviewer), have **frequently** been reported in the animal electrophysiological and human fMRI studies investigating pain empathy (Fan et al., 2011; Allsop et al., 2018; Timmers et al., 2018; Fallon et al., 2020; Zhang et al., 2022).

ii) Moreover, the iEEG recording also imposed **practical constraints** on our selection of ROIs. Similar to previous iEEG studies (Parvizi & Kastner et al., 2018; Oehr, 2023), the current study also had to consider the limited coverage of implanted electrodes in the brain. Due to this concern, we adopted the same approach as previous iEEG studies (Zheng et al., 2019; Chen et al. 2021; Sonkusare et al., 2023), focusing on brain regions that were commonly covered by the implanted electrodes. In our dataset, we had an adequate number of channels in each of the four targeted brain regions (40-98 channels), but not in other empathy-related brain regions (e.g., only 4 patients had a total of 21 channels located in the MCC).

Based on these two considerations, we have chosen the AI, ACC, amygdala, and IFG as our brain regions of interest. In relation to other empathy-related brain regions, we encouraged future studies to conduct further investigations in the revised *Discussion* (*Page 28 Lines 759-763*).

Point 5. In the neural features analysis, it is not clear why authors selected 7 features and then excluded the seventh one and select 6 as sufficient. Please explain this choice.

Response: We apologized for bringing in this confusion. Accordingly, we have elaborated the analytical procedure and analysis rationale of our decoding analysis here and in the revision (*Page 45 Lines 1260-1276*).

In the decoding analysis, we first searched for a necessary feature combination that was necessary for the classification between painful and non-painful stimuli. We found that the top 8 features consisted of a necessary feature combination, i.e., removing these features resulted in a decoding accuracy of 49.78% that did not significantly differ from chance level. (ps: the number "8" here corresponds to the 7 features in the original version as we updated the results in response to Reviewer #1's *Main Point #3b* and *Minor Point #12*).

Second, after identifying the necessary combination, we individually assessed whether

each feature within this necessary combination could sufficiently discriminate between painful and non-painful stimuli by itself. If a feature by itself was able to significantly distinguish between two types of stimuli, this feature was considered as a sufficient feature. This analysis revealed that each of the 8 features was a sufficient feature (significantly discriminate between painful and non-painful stimuli, all FDR-corrected $p_s < 0.01$; *Page 45 Lines 1271-1276*). (ps: this corresponds to the 6 features in the original version as we updated the results in response to Reviewer #1's *Main Point #3b* and *Minor Point #12*).

Point 6. Other analytical choices are not clearly motivated. On page 12, line 241, for example, it is not clear why the authors adopt one-sided (rather than two-sided) comparisons.

Response: We appreciate the reviewer's suggestion. Accordingly, we have provided a clear explanation of our choice to use one-sided testing (*Page 39 Lines 1082-1087*). Specifically, we clarified that these comparisons were conducted as follow-up analyses for the time-frequency analyses in order to further reveal the onset time for empathic neural activity shown in *Fig. 2e-h*. In the time-frequency power analysis, we already showed the directionality of the conditional differences, i.e., whether there was higher or lower power in the painful vs. non-painful contrasts (*Fig. 2e-h*). Therefore, our aim in this follow-up analysis was to determine the earliest time point exhibiting the pattern observed in *Fig. 2e-h*. Hence, this specific objective motivated us to conduct one-sided statistical tests.

Additionally, addressing the reviewer's potential concern, we also conducted two-sided statistical tests and showed that these results remained robust and significant under two-sided testing (*Fig. R16*). However, considering the hypothesis-driven nature of this follow-up analysis, we decided to retain the use of one-sided comparisons.

Figure R16. The oscillatory power difference between the painful and non-painful condition in the AI, ACC, amygdala and IFG when conducting one-sided statistical tests (A-D) or two-sided statistical tests (E-H). Time points with significant conditional power differences are highlighted with horizontal lines (one-sided statistical tests) or cross symbols (two-sided statistical tests)

(corrected $p < 0.05$, 1000 permutations, survived the cluster-based permutation test for multiple comparisons). Solid (painful condition) and dashed (non-painful condition) lines indicate the mean power across all channels for each time point, with shading representing the standard error.

Point 7. The discussion is difficult to read. I would like to suggest that the authors help the readers by adding title for each section (as they did for the result).

Response: We appreciate this helpful suggestion from Reviewer 2. However, due to the formatting requirements of the journal, the inclusion of subtitles in the Discussion section was not feasible. Instead, to improve the readability, we re-structured the *Discussion* to ensure clarity of the main topic in each section for the audience.

The revised *Discussion* was organized as: i) we summarized main findings of empathy-related oscillatory power and inter-regional interactions into an integrated neurodynamic model of human empathy (paragraphs #1 and #2), ii) presented our understanding of observed regional oscillatory power changes (paragraphs #3 and 4), iii) indicated the importance of observed inter-regional communications in unraveling the functional organization of empathy network (paragraphs #5-9), and iv) discussed other considerations and pointed towards new directions for future studies (paragraphs #10 and 11) and drew out the final conclusion (paragraph #12).

Point 8. Previous studies hinting at the dynamic nature of brain rhythms during empathy for pain should be quoted and briefly discussed even if they do not use iEEG (e.g. Betti et al, 2009; Zebarjadi et al, 2021)

Response: We thank Reviewer 2 for this suggestion. Accordingly, the relevant papers were cited in the revised manuscript (*Page 4 Line 91*).

Minor points:

- a) Some of the information in the analysis part repeated in the method section. I suggest avoiding presenting them twice and to leave the information sufficient to understand the analysis in the results section and all the details in the methodology.
- b) The results are presented in a clear and logical manner. However, the manuscript is very dense, presenting several analyses in a way that it is hard to follow. Some sections with many statistics could be organized into tables.
- d) Line 190: I would suggest the authors to state that the empathic responses to vicarious pain were comparable to healthy controls – more than representative of a general population.
- e) Line 123- 130: this sentence needs to be rephrased
- f) Line 140: felt painful
- g) Supplementary tables with the subjective data of patients and controls rather than just similarity indices may be useful

Response: We appreciated these helpful suggestions by Reviewer 2. Accordingly, we have removed redundant methodological details from the *Result* section (Point #a),

organized the full statistical reports into tables (Point #b; revised Table S1-3), modified some specific statements (Point #d; Page 8 Lines 214-216; Point #e; Pages 5-6 Lines 137-145; and Point #f; Page 6 Lines 154-155), and provided the subjective ratings of patients and healthy controls (Point #g; revised Table S5).

Point #d (revised Results, Page 8 Lines 214-216): “Thus, patients’ empathic responses to vicarious pain were comparable with those of healthy individuals.”

Point #e (revised Introduction, Pages 5-6 Lines 137-145): “In an effort to delineate the specific contributions of these spectral-temporal-spatial specific patterns to vicarious pain perception and identify important neural features, we further investigated how these neural features jointly contributed to the perception of others’ pain. Moreover, to assess the associations between critical neural features within the pain empathy network, we tested how these critical neural features were linked to empathy-related behavioral measures, including the strength of overall empathic responses and empathy-related subprocesses (i.e., evaluation of perceived pain intensity and one’s own unpleasantness) during perception of other’s pain.”

Point #f (revised Results, Page 6 Lines 154-155): “Following the picture viewing phase, patients were asked to judge whether the person depicted in the picture experienced pain.”

c) Line 511: it is not clear to me why one behavioral component has been removed from this analysis.

Response: We thank Reviewer #2 for pointing out this issue. Accordingly, we conducted the similar analysis on the other behavioral component (arousal levels). Specifically, we examined the correlation between the arousal level and ACC alpha oscillations/AI low-frequency oscillations and did not find any significant result ($ps > 0.05$). These results were reported in the revised Results (Page 21 Lines 552-556).

➤ **Responses to Reviewer #3's comments:**

This work utilizes intracranial electrophysiological recordings in humans to examine in detail the complex interplay between the anterior insula, the anterior cingulate cortex, the amygdala and the inferior frontal gyrus across different oscillation frequencies at a millisecond timescale during vicarious pain perception. Highly valuable iEEG data collected from pre-clinical epilepsy patients in a well-established pain judgment task was analyzed with advanced intra- and inter-regional neurophysiological analysis techniques and the results were presented in a highly organized and clear manner. In addition, the relative importance, the necessity, and the sufficiency of a wide array of neural activity markers were assessed via a decoding algorithm to construct a neurodynamic model of empathy for pain. Overall, this is a very challenging, and thus highly valuable work that enhances our understanding of the interactions between critical brain regions that underlie vicarious pain representations.

I have a number of comments/recommendations that I believe will increase the impact and the understandability of the manuscript.

General response: We thank Reviewer #3 for the positive evaluations of our manuscript. We highly appreciate Reviewer #3's constructive suggestions. Specific point-by-point responses are listed below:

Point 1. In the third paragraph of the Introduction section, immediately after discussing the limitations of studying empathic neural responses using fMRI and EEG/MEG and the advantages of using intracranial methods, the goal of the present study is introduced. This gives the feeling that vicarious pain responses have never been investigated before using intracranial recordings in the four regions of interest in the human brain. However, this is not true. The authors should refer to studies such as

Hutchison, W. D., Davis, K. D., Lozano, A. M., Tasker, R. R., & Dostrovsky, J. O. (1999). Pain-related neurons in the human cingulate cortex. *Nature neuroscience*, 2(5), 403-405.

Soyman, E., Bruls, R., Ioumpa, K., Müller-Pinzler, L., Gallo, S., Qin, C., ... & Gazzola, V. (2022). Intracranial human recordings reveal association between neural activity and perceived intensity for the pain of others in the insula. *Elife*, 11, e75197.

Response: This is an excellent point, we greatly appreciate the constructive suggestion from Reviewer 3. In the revised *Introduction* (*Page 4 Lines 98-105, Page 5 Lines 118-122*), we have now **reviewed** previous iEEG studies related to empathy for pain, including Hutchison et al., 1999; Mo et al., 2022; Soyman et al., 2022.

On one hand, we underscored that these iEEG studies have yielded **valuable insights** into the neural basis of empathy. Specifically, Hutchison et al. (1999) conducted single-neuron recordings of ACC and found that single neurons in the ACC responded when participants witnessed others' fingers being pin-pricked, providing electrophysiological evidence of the engagement of ACC in the processing of others' pain. Soyman et al. (2022) adopted iEEG to characterize the electrophysiological responses of insula to others' pain, with broadband activity of insula encoding other's

pain intensity. Mo et al. (2022) examined the recurrence-related characteristics of the default-mode network during resting-state and the relationship with a questionnaire score of empathy trait. On the other hand, we acknowledged their **limitations** such as focusing exclusively on a single brain region (the ACC in Hutchison et al., 1999; AI in Soyman et al., 2022), or examining empathy-irrelevant, non-task-specific resting-state instead of task-related and empathy-specific neural activity (i.e., associating empathy-related traits with resting-state in Mo et al., 2022).

Moreover, we clarified that **significant gaps** existed between findings from these studies and the understanding of multi-areal neural dynamics in vicarious pain perception. These gaps served as **motivation** for our current study, which aims to investigate both temporal order and spectral characteristics of neural oscillatory activity within empathy-related brain regions, as well as the rapid-scale information flow among these regions through recording iEEG signals at multiple empathy-related regions with millimeter and millisecond resolutions during perception of others' pain.

#Point 2. In the last paragraph of the Introduction section, different sentences use different tenses. For readability, please stick with the present or past tense consistently throughout the paragraph.

Response: We thank Reviewer #3 for this reminder. We have made revisions accordingly to ensure consistency in tenses within that paragraph (*Pages 5-6 Lines 134-148*). Throughout the entire manuscript, we also devoted special attention to the issue of tense during the revision.

Point 3. The fact that iEEG participants did not show any significant differences in accuracies and reaction times between the painful and non-painful stimuli during the pain judgment task is shown in sufficient detail. However, the average, the standard deviation, the minimum, and the maximum of these accuracies and reaction times for painful and non-painful stimuli are not shown in the Results or the Methods section. These values must be clearly reported so that the reader can grasp what type of performances the participants showed in the task.

Response: Following the reviewer's suggestion, we have now reported descriptive statistical details of response accuracies and response times for painful and non-painful stimuli in the *revised Table S4 (Table R2)*.

#Point 4. In the sentence starting at line 257, it is stated that a smaller sliding window was used for the analysis reported in the Supplementary Figure 3. Although, it is reported clearly in the earlier pages that the first analysis was conducted with a 100-ms window and this analysis is conducted with a 50-ms window, the reader has to go between several pages to fully grasp what is being changed in this analysis. The reader would substantially benefit if the authors clearly state here the change was from the earlier 100-ms to 50-ms time window.

Response: We thank Reviewer #1 for asking us to clarify this issue. It should indeed be clear to the reader what was done and how. Accordingly, in the revised manuscript,

we decided to only report this analysis in the *Methods* section and explicitly mentioned that the change made in this analysis, compared to the original one, lied in adjusting smoothing parameters (from a 100 ms to a 50 ms time window) (*Page 39 Lines 1091-1094*).

#Point 5. In the sentence starting at line 257, the authors state that the smaller sliding window analyses showed similar temporal profiles to the longer sliding time window. However, as seen in Supplementary Figure 3G, there was a very critical difference in that the latency of the IFG high-gamma responses were 160 ms, as opposed to the 60-ms latency in the analyses with the longer smoothing window. This latency difference challenges the neurodynamic model proposed in Figure 6, which present very early IFG responses followed by other brain regions. The authors must discuss this critical difference openly in the manuscript.

Response: Thank you, this is a valid point that we did not fully address in the original submission. Upon closer examination of the results using a smaller sliding window of 50 ms, we found that the early response of IFG high-gamma activity exhibited the same pattern and remained significant with a traditional threshold of $p < 0.05$ (Fig. R17), although it was insignificant at the stringent threshold of $p < 0.01$. In the revision, we have chosen to transparently report all these results, including both the significant results at $p < 0.01$ and $p < 0.05$ thresholds, in order to comprehensively present the pattern of IFG high-gamma activity and alleviate any confusion readers may have regarding the absence of early IFG high-gamma responses (revised Fig. S15).

Figure R17. Temporal profile of IFG high-gamma power with the power time series smoothed by a sliding window of 50 ms. We found the early increase of IFG high-gamma power within a traditional threshold of $p < 0.05$ (significant increases from 80 ms to 130 ms, 160 ms to 290 ms, and 390 ms to 460 ms, outlined with cross symbols), although it was not significant at the stringent threshold of $p < 0.01$ (significant increases from 160 ms to 290 ms and 390 ms to 460 ms, outlined with horizontal lines).

#Point 6. In the paragraph starting at line 308, the authors use sentences as “The ACC and amygdala synchronized to a greater degree in the painful than non-painful conditions...”. It is as if there was synchronization in both conditions and one condition was higher than the other one. In fact, the analyses show that there was no synchronization at all in the non-painful condition. All the sentences in this paragraph must be rephrased to capture this critical information in order to prevent such misinterpretations.

Response: We agreed with Reviewer #3's comments. Accordingly, we have revised all relevant sentences to ensure clarity and prevent any potential misinterpretations. Additionally, we have reported the significance of each condition (*Page 13-14 Lines 338-351*):

Page 13-14 Lines 338-351: "The comparison between the painful and non-painful conditions revealed significant suppression in beta-band power correlations between the ACC and AI (25-32 Hz, Fig. 3a; significant coupling in the non-painful condition, which became absent in the painful condition, Supplementary Fig. 4a, b), as well as between AI and the amygdala (18-24 Hz, Fig. 3b; significant coupling in both conditions, Supplementary Fig. 4c, d). Interestingly, between the amygdala and ACC during the perception of other's pain (vs. non-pain), there were higher power correlations within the lower frequency range of the beta band (18-22 Hz, Fig. 3c; significant ACC-amygdala coupling in the painful not non-painful conditions, Supplementary Fig. 4e, f) but lower power correlations within the upper frequency range of the beta band (25-30 Hz, Fig. 3c; significant ACC-amygdala coupling in the non-painful, but not painful, condition; Supplementary Fig. 4g, h). These results together suggested that the beta oscillation may act as a prominent mediator for inter-regional communications among the ACC, AI, and amygdala during perception of other's pain."

Point 7. In Figure 5D and the related analyses in the text, the statistical significance of the correlations between empathic strength and neural features are reported in detail. However, quite importantly, some of these correlations are positive and some are negative, which is not stated or discussed in the manuscript. Especially for neural features 4, 5, and 6, the direction of the correlations are quite surprising given the analyses reported in earlier sections of the manuscript. To take neural feature 6 as an example, the amygdala beta oscillation analyses reported in Figure 2K shows that these oscillations are significantly decreased during vicarious pain. Since empathic processing would increase during vicarious pain, one would expect a negative correlation between these two measures, whereas Figure 5D shows a positive correlation. I do understand that these ladder analyses are conducted on painful-nonpainful conditional difference scores, and thus do not argue that there is necessarily an analytic error here. But these nuanced differences in analyses must first be explicitly stated in the Results section and then also how they should be interpreted, as well as their potential explanatory power in terms of the neurodynamical processes underlying empathic processing, must be openly discussed in the Discussion section.

Response: We thank Reviewer #3 for the valuable suggestions which are helpful to enhance the readability of our results. In the revised *Result* section (*Page 19-20 Lines 511-530*), we have clearly stated how the neural features varied with the strength of empathy responses, providing our understanding of the correlations between empathy strength and neural features. We also discussed how results from the contrast and correlation analysis helped to understand the *qualitative* and *quantitative* role of

suppressed low-frequency power in the ACC (alpha) and AI in empathic response, and have accordingly updated our neurodynamic model of empathy (revised Fig. 6).

First, please note that we have updated our correlation results due to the modifications made in response to *Reviewer #1's Main Point #3b* and *Minor Point #12* and the implementation of linear-mixed effect models to control for individual variations. In the updated results, only two neural features (ACC alpha oscillations and AI low-frequency oscillations) were reliably associated with empathic strength (revised *Results, Page 20 Lines 519-525, revised Fig. 5d*). The correlations with other features reported in the original submission did not survive multiple correction under the examination of linear-mixed effect models. Therefore, in the revision, we only focused on the correlation results between empathy strength and these two features. The ACC alpha power and AI low-frequency power were negatively associated with empathy strength, indicating that stronger empathic responses were linked to greater suppression of ACC alpha oscillations and AI low-frequency oscillations. These correlation patterns were consistent with the patterns of decreased low-frequency power observed in the “painful vs. non-painful” contrast (Fig. 2e, f; Fig. R4A, B). The observation that ACC alpha power and AI low-frequency power were suppressed in the “painful vs. non-painful” contrast and negatively associated with empathy strength suggested that the suppressed low-frequency power in the ACC (alpha) and AI not only facilitated qualitative differentiation between others’ pain and non-pain (signaling the presence of other’s pain), but also quantitatively tracked the strength of empathic responses.

Furthermore, to aid in the interpretation of the functional meaning of low-frequency power in the ACC (alpha) and AI, we investigated their association with perceived pain intensity in others and one’s own level of unpleasantness (revised *Results, Page 20 Line 532-547*). We revealed a negative association between ACC alpha oscillations and perceived pain intensity (revised Fig. 5e). Consistent with this, a recent iEEG study showed that decreased ACC alpha power predicted higher pain states in participants (Shirvalkar et al., 2023). Thus, suppression of ACC alpha power was implicated in encoding both the intensity of pain experienced by oneself (Shirvalkar et al., 2023) and vicariously by others (current study). For the AI, we observed a negative correlation between AI low-frequency oscillations and perceived pain intensity (similarly reported by Soyman et al., 2022), but also a positive correlation with one’s own unpleasantness (revised Fig. 5f). This suggested that, during empathy for other’s suffering, the suppression of AI low-frequency power may enhance other-related processing (e.g., the perception of intensity of others’ pain) while inhibited self-related processing (e.g., the generation of stronger personal unpleasantness).

#Point 8. In the sentences starting at line 540, the authors report that the neural activity did not vary as a function of arousal levels. However, whether there was a significant difference between the painful and on-painful stimuli in these arousal levels is not reported. The results of such an analysis must be reported here.

Response: We took Reviewer #3's suggestion and compared the arousal levels between the painful and non-painful stimuli (results were reported in the revised *Methods*, *Pages 31-32 Lines 860-869*):

Page 20 Line 532-547: "In addition, to confirm that the neural findings did not result from possible differences in arousal levels between painful and non-painful stimuli, patients also provided ratings of arousal level for each stimulus ("How intense is your emotional response induced by this picture"; 0 = extremely calm, 100 = extremely strong). Although painful stimuli were associated with higher arousal levels than non-painful stimuli (difference: 55.98 ± 6.00 , $t_{15} = 9.33$, $p = 1.23 \times 10^{-7}$, Cohen's $d = 2.33$, 95% *CI*: 43.20, 68.76), the observed tempo-spectral patterns did not vary as a function of arousal levels (*Supplementary Fig. 9*). These results suggested that the observed neural effects were not due to potential differences in arousal levels between painful and non-painful stimuli."

#Point 9. The Discussion section would benefit substantially if the authors discuss their anterior insula findings in relation to the findings of the Soyman et al. (2022) study, which analyzed intracranial recordings from the insula while the participants engaged in a vicarious pain perception task similar to the one used in this study.

Response: This is a very valid point that we did not address in the initial submission. We sincerely appreciate this constructive suggestion and further probed the relationship between our findings and those of Soyman et al. (2022). In summary, we identified consistent findings and elucidated the distinct patterns of insula activity (detailed below). In the revision, i) we highlighted the consistent findings that both the current study and the study of Soyman et al. (2022) showed low-frequency insula power encoding perceived pain intensity (*Page 23 Lines 624-625*); ii) we acknowledged the different spectral patterns of insula observed in our results and theirs (Soyman et al., 2022) and discussed potential implications arising from these distinct findings (*Pages 27-28 Lines 745-763*).

➤ Consistent findings of insula

In the current study, we found a negative association between AI low-frequency power and the intensity of perceived pain in others. This finding aligned with Soyman and colleagues' finding that a low-frequency cluster within the insula (13-17 Hz) exhibited a negative correlation with the intensity of perceived pain in others (Soyman et al., 2022).

➤ Differences and potential insights

Our work focused on the anterior insula and revealed significant power changes specifically within low-frequency bands in AI (but not in the gamma and high-gamma

bands). In contrast, Soyman et al (2022) explored activity across both the anterior and posterior insula and found that broadband activity, rather than specific frequency bands, was involved in encoding intensity of perceived pain in others. These distinct spectral patterns may imply that the anterior and posterior insula have different response profiles during processing of other's pain. Alternatively, these differences may be attributed to variations in experimental design and stimulus types between our study and that of Soyman et al (2022). For example, while our study asked patients to make a dichotomous judgment on whether the person depicted in a static picture experienced pain or not, Soyman et al (2022) asked patients to rate the intensity of pain perceived in a person shown in a video clip using continuous evaluation. It would be interesting for future studies with larger sample size and trial number to further investigate if and how the experimental task modulates spectral patterns within different subregions of the insula as well as other empathy-related regions during processing others' pain.

Point 10. In the sentence starting at line 769, the authors state that the participants pressed the left or right button for reporting their answers for the presence or the absence of pain. Was the assignment of the buttons to the responses randomized across trials or participants? If not, wouldn't the systematic association of one motor behavior with one response lead to systematic biases in the neural responses?

Response: We thank Reviewer #3 for asking us to clarify this issue. We clearly stated here and in the revised *Method* (Pages 30-31 Lines 832-839).

All participants used their dominant right hand to indicate whether they thought the person depicted in the picture felt painful (or not) by pressing the index (middle) finger. This judgment-button and response finger was fixed and not randomized across trials or participants. However, this design *did not bias* participants' behavioral performances, as we found comparable response times and response accuracies between the painful and non-painful condition (response time: $t_{21} = -0.16$, $p = 0.874$; response accuracy: $t_{21} = 1.26$, $p = 0.221$).

Moreover, this design was also unlikely to influence the observed neural results. First, the motor cortex was not the region of interest in the current study. Furthermore, a crucial aspect of our experimental setup involved *separating* the stimulus presentation phase from the response phase (i.e., separate screens for stimulus display and pain judgment, Fig. 1a). Participants were instructed to carefully view the picture during the stimulus presentation phase and make their pain judgment after its disappearance (the response phase). This separation effectively isolated vicarious pain perception from any potential biasing effects of motor responses on neural responses to others' pain (reported in Page 30-31 Lines 836-839). Consistent with previous studies (Mu et al., 2008; Zhou & Han, 2021), our neural analysis solely focused on the stimulus presentation phase; thus, it is unlikely that participants' motor responses could have affected our observed neural effect.

Point 11. In the sentences starting at line 774, the authors state that the 20 pictures were shown once, which means that the maximum number of trials that could go into any analysis was 20. Furthermore, in the sentence starting at line 913, it is stated that any channel with more than 30% epochs removed from either painful or non-painful conditions was excluded. This means that it was possible for a channel to be included in the analyses if it had 7 painful and 7 non-painful trials. For electrophysiological analyses, these are surprisingly low numbers of trials. In addition, some of the analyses in the manuscript are conducted by taking the conditional difference between matched pairs of painful and non-painful stimuli. If unmatched trials were rejected, these analyses would be conducted with even lower numbers of matched trials. For each analysis, the author must report the mean, the standard deviation, the minimum, and the maximum number of trials (across channels) that were taken into account when computing that particular metric or conducting that analysis for transparent reporting of the analytical procedures.

Response: We thank Reviewer #3 for raising this question and corresponding suggestions. Accordingly, we i) provided evidence from the literature justifying the ***exceptionally high signal quality*** of iEEG as a means to illustrate the feasibility of utilizing limited number of trials (Ball et al., 2009; Parvizi & Kastner, 2018; Mercier et al., 2022); ii) took reviewer's suggestion to report detailed information of trial numbers for each analysis (Table R3, Supplementary Data 1); and iii) encouraged future studies to examine empathic neural responses with larger trial numbers in the revised Discussion (Page 28 Lines 759-763).

i) As pointed out the reviewer, the number of trials in our study was relatively limited compared to fMRI and scalp EEG studies. We made this choice due to i) the clinical and hospital constraints associated with intracranial EEG recording and ii) the practical consideration that patients were donating their time and energy during a challenging period (Parvizi & Kastner, 2018; Mercier et al., 2022). Therefore, we minimized the experimental duration (Mercier et al., 2022) and reduced the number of trials accordingly. However, this potential limitation in trial numbers can be compensated for by the ***exceptional signal-to-noise ratio*** (SNR) observed in iEEG data compared to fMRI or scalp EEG data (Parvizi & Kastner, 2018; Mercier et al., 2022). For example, when simultaneously recording scalp EEG and iEEG signals, Ball and colleagues (2009) found that the signal quality of iEEG data was ***20 to over 100 times better*** than that of scalp EEG. Furthermore, recent iEEG studies have demonstrated that, with the high SNR of iEEG data, particularly in functionally specialized brain regions, stimulus-evoked neural responses can be ***reliably observed using fewer trials*** (Mercier et al., 2022; Lachaux, 2023). For example (Fig. R1, to avoid going back the Fig. R1, we pasted here again, adapted from Mercier et al., 2022, Fig. 11), broadband and high-frequency neural activity can be reliably observed with just 2 trials while maintaining consistent patterns across a range of 2 to 59 trials.

Figure R1. Neural responses with different number of averaged trials. The broadband activity (A) and the magnitude envelope of high-frequency activity (40-150 Hz, B) showed similar patterns across a range of 2 to 59 trials (adapted from Mercier et al., 2022, Fig. 11).

ii) Moreover, as per the recommendation of Reviewer #3, we have included comprehensive descriptive statistics of trial numbers for each analysis (including mean, standard deviation, and range) in a Supplemental file (Table R3, Supplementary Data 1). In summary, each analysis encompassed an average of approximately 90% of trials in every brain region or region-pair (Table R3).

iii) We admitted that a larger number of trials will be beneficial and recommended using datasets with larger trial numbers to further examine observed empathy-related neural features in the revised *Discussion* (Page 28 Lines 759-763).

#Point 12. The MNI coordinates of all channels included in the final analyses must be reported in a supplementary table or file. In addition, this file must have, for each channel, the MNI coordinates of the nearest white-matter neighbor reference channel used in the bipolar montage analyses. In the Methods section, around the sentence starting at line 899, the authors must clearly state, in this bipolar montage referencing, whether the same nearest white-matter reference channel was used for channels that were placed in different regions in the brain. The purpose of the bipolar analyses is to overcome the potential problem of increasing the estimated correlations between different brain regions that can stem from having the same reference signal for those brain regions. Thus, the authors must clearly state that indeed different reference channels are used for channels in different brain regions in the bipolar montage.

Response: We appreciate the reviewer's suggestions. Accordingly, in the revised

Method (Page 35 Lines 978-981), we reported that a majority of cross-regional channel pairs (an average of 84.09% of pairs) utilized different white-matter channels as references for their respective brain regions, when applying the closest-white-matter referencing scheme. Additionally, following Reviewer #3's recommendations, we included a supplementary table for the MNI coordinates of all channels and their corresponding nearest white-matter neighbor reference channels for each ROI (Table R4, Supplementary Data 2).

#Point 13. In the sentence starting at line 1039, the authors state that transfer entropy was computed at a lag of 10 ms. Considering that the spectral power data was down-sampled to 100 Hz and a 100-ms sliding time window was used for sliding, responses at a 10-ms lag would be heavily influenced by a wide temporal window that intersects at a substantial amount of earlier and later time points in the two channels. The transfer entropy analysis must be conducted at various lags starting from 0 ms to 100 ms to assess the sensitivity of the reported results.

Response: We thank Reviewer #3 for this helpful suggestion. Following the reviewer's recommendation, we conducted transfer entropy (TE) analysis at various lags. Since we used a sliding window of 50 ms instead of 100 ms to compute the transfer entropy, we conducted the TE analysis with lag values ranging from 10 ms to 50 ms in increments of 10 ms. This range (i.e., 10% of the analysis window length) aligned with that used in directionality analyses in previous studies (Oehrn et al., 2018; Ten Oever et al., 2021). The results obtained from this analysis showed that the observed pattern in transfer entropy remained stable as the lag increased (Fig. R18 D-F, revised Fig. S16, reported in Page 41 Lines 1148-1152). In addition, even when computing TE based on non-normalized data, we still observed similar TE patterns with increasing lags (Fig. R18 G-I). These results suggested that our transfer entropy results were relatively robust and not significantly affected by changes in lags.

Figure R18. Transfer entropy (TE) analysis between ACC, AI, and amygdala. (A-C) When computing TE at a lag of 10 ms (which was reported in the original submission), we found main effects of pain for ACC-AI (A. reduced TE in both directions during perception of other's pain between ACC and AI), interaction effect for AI-amygdala TE (B. processing of others' pain specifically suppressed beta information flow from the amygdala to AI) and main effects of pain for ACC-amygdala TE (C. enhanced TE in both directions during perception of other's pain between ACC and amygdala). (D-I) The main effects of pain for ACC-AI and ACC-amygdala TE and interaction effect for AI-amygdala TE remained when lag values ranging from 10 ms to 50 ms based on minimally smoothed data (D-F) or non-smoothed data (G-I). Significant main effect of pain is highlighted with orange circles and significant interaction effect is highlighted with pink circles. Amy = amygdala. ** $p < 0.05$, *** $p < 0.01$, NS, not significant.

Table R1. Channel/channel-pair information for each patient

Patient	Number of channels				Number of channel pairs					
	ACC	AI	Amy	IFG	ACC -AI	ACC -Amy	AI -Amy	ACC -IFG	AI -IFG	Amy -IFG
sub01		4	3	5			12		20	15
sub02		2								
sub03			7							
sub04			8							
sub05	4	6	2	2	24	8	12	8	12	4
sub06	5	4	4	3	20	20	16	15	12	12
sub07	5	2		8	10			40	16	
sub08	4	3		5	12			20	15	
sub09		5	3	5			15		25	15
sub10			3							
sub11		9	4	13			36		117	52
sub12	1	2	4	15	2	4	8	15	30	60
sub13		2	5	7			10		14	35
sub14		10	4	2			40		20	8
sub15		6	3				18			
sub16		8	4	3			32		24	12
sub17	8	11	5	7	88	40	55	56	77	35
sub18			3							
sub19	13	6		5	78			65	30	
sub20		9	4				36			
sub21		4		8					32	
sub22		5	2	3			10		15	6
Total	40	98	68	91	234	72	300	219	459	254

Table R2 *Descriptive statistics of behavioral performances*

Behavioral indices		Non-painful condition	Painful condition
Response accuracy (%)	Mean \pm SE	81.82 \pm 3.58	87.73 \pm 3.15
	Minimum	50	60
	Maximum	100	100
Response time (s)	Mean \pm SE	1.02 \pm 0.12	1.00 \pm 0.10
	Minimum	0.28	0.33
	Maximum	2.99	1.69

Table R3 *Descriptive statistics of trial number across channels or channel-pairs*

Main analysis		Trial numbers
Time-frequency analysis		
AI	Mean \pm SE	19.43 \pm 0.07
	Minimum	17
	Maximum	20
ACC	Mean \pm SE	19.63 \pm 0.11
	Minimum	17
	Maximum	20
Amygdala	Mean \pm SE	19.51 \pm 0.07
	Minimum	18
	Maximum	20
IFG	Mean \pm SE	19.64 \pm 0.09
	Minimum	16
	Maximum	20
Power correlation analysis or Transfer entropy analysis		
ACC-AI	Mean \pm SE	19.12 \pm 0.06
	Minimum	16
	Maximum	20
ACC-amygdala	Mean \pm SE	18.92 \pm 0.09
	Minimum	17
	Maximum	20
AI-amygdala	Mean \pm SE	19.01 \pm 0.04
	Minimum	16
	Maximum	20
Phase-amplitude coupling analysis		
ACC-IFG	Mean \pm SE	19.31 \pm 0.05
	Minimum	17
	Maximum	20
AI-IFG	Mean \pm SE	19.37 \pm 0.04
	Minimum	16
	Maximum	20
Amygdala-IFG	Mean \pm SE	19.12 \pm 0.08
	Minimum	14
	Maximum	20
Correlation analysis with empathy-related ratings		
ACC alpha oscillations	Mean \pm SE	9.58 \pm 0.12
	Minimum	7
	Maximum	10
ACC beta oscillations	Mean \pm SE	9.58 \pm 0.12
	Minimum	7

	Maximum	10
AI	Mean \pm SE	9.43 ± 0.08
low-frequency oscillations	Minimum	8
	Maximum	10
Amygdala beta oscillations	Mean \pm SE	9.39 ± 0.09
	Minimum	8
	Maximum	10
ACC-amygdala (low-)beta coupling	Mean \pm SE	8.92 ± 0.09
	Minimum	7
	Maximum	10
ACC-IFG PAC	Mean \pm SE	9.24 ± 0.06
	Minimum	7
	Maximum	10
AI(alpha)-IFG PAC	Mean \pm SE	9.22 ± 0.05
	Minimum	6
	Maximum	10
Amygdala-IFG PAC	Mean \pm SE	8.94 ± 0.09
	Minimum	5
	Maximum	10

Table R4 Channel coordinate within the ACC, AI, amygdala and IFG for all patients in the MNI space

Patient ID	The coordinate of included channel for ROIs			The coordinate of the corresponding nearest white-matter neighbor reference channel		
Channels for the ACC						
5	-6	21	-22	-11	30	-9
5	-7	23	-19	-11	30	-9
5	-8	26	-16	-11	30	-9
5	-10	28	-13	-11	30	-9
6	-12	38	-9	-38	-2	-26
6	-12	40	-5	-44	-2	3
6	-13	41	-1	-44	-2	3
6	-13	43	3	-44	-2	3
6	-14	45	7	-44	-2	3
7	4	36	14	18	39	27
7	6	37	16	18	39	27
7	9	37	19	18	39	27
7	12	38	22	18	39	27
7	15	38	24	18	39	27
8	2	43	12	17	47	17
8	6	44	13	17	47	17
8	10	45	14	17	47	17
8	13	46	16	17	47	17
12	-4	19	-17	-9	21	-12
17	6	38	5	15	47	15
17	8	40	8	15	47	15
17	10	43	10	15	47	15
17	13	45	12	15	47	15
17	5	26	23	15	34	31
17	8	28	25	15	34	31
17	10	30	27	15	34	31
17	13	32	29	15	34	31
19	-7	20	37	-15	20	35
19	-9	21	40	-15	20	35
19	0	39	1	-16	37	17
19	-3	39	4	-16	37	17
19	-6	39	7	-16	37	17
19	-3	17	31	-15	20	35
19	-5	18	34	-15	20	35
19	-8	38	9	-16	37	17
19	-11	38	12	-16	37	17
19	-14	38	15	-16	37	17

Patient ID	The coordinate of included channel for ROIs			The coordinate of the corresponding nearest white-matter neighbor reference channel		
19	-6	22	29	-15	20	35
19	-9	21	31	-15	20	35
19	-12	21	33	-15	20	35
Channels for the AI						
1	-33	21	-4	-38	-6	-26
1	-37	20	-3	-38	-6	-26
1	-39	5	4	-38	-6	-26
1	-43	4	5	-50	-7	-25
2	36	-2	16	47	0	19
2	40	-1	17	47	0	19
5	-37	12	-11	-41	0	-25
5	-37	15	-8	-41	0	-25
5	-36	17	-5	-31	32	14
5	-35	20	-2	-31	32	14
5	-34	22	1	-31	32	14
5	-34	25	4	-31	32	14
6	-33	10	1	-44	-2	3
6	-37	11	1	-44	-2	3
6	-37	-5	0	-44	-2	3
6	-40	-3	2	-44	-2	3
7	35	8	3	19	20	36
7	38	8	5	19	20	36
8	36	15	1	25	43	-3
8	39	16	3	25	43	-3
8	43	17	5	25	43	-3
9	38	12	-8	29	26	4
9	36	15	-6	29	26	4
9	34	17	-3	29	26	4
9	33	20	-1	29	26	4
9	31	23	2	29	26	4
11	-38	3	-6	-39	1	-24
11	-38	6	-3	-39	1	-24
11	-38	9	0	-39	1	-24
11	-38	11	4	-39	1	-24
11	-37	12	-4	-39	1	-24
11	-39	11	-1	-39	1	-24
11	-37	0	3	-45	-22	-4
11	-39	-2	5	-45	-22	-4
11	-42	-4	8	-45	-22	-4
12	-32	8	8	-32	6	4
12	-31	10	11	-31	12	14

Patient ID	The coordinate of included channel for ROIs			The coordinate of the corresponding nearest white-matter neighbor reference channel		
13	-35	9	-15	-37	0	-24
13	-38	7	-15	-37	0	-24
14	44	13	-9	36	-2	1
14	42	14	-6	36	-2	1
14	41	16	-3	36	-2	1
14	40	17	1	36	-2	1
14	39	19	4	34	25	18
14	38	20	7	34	25	18
14	36	22	11	34	25	18
14	35	23	14	34	25	18
14	39	-1	3	36	-2	1
14	37	-7	11	33	-6	10
15	-35	8	-4	-30	17	16
15	-34	10	-1	-30	17	16
15	-34	11	2	-30	17	16
15	-33	13	6	-30	17	16
15	-32	14	9	-30	17	16
15	-31	16	13	-30	17	16
16	34	12	-13	36	5	-25
16	33	13	-9	33	3	-24
16	32	14	-6	33	3	-24
16	31	16	-3	27	23	14
16	31	17	0	27	23	14
16	30	19	4	27	23	14
16	29	20	7	27	23	14
16	28	22	10	27	23	14
17	35	6	-10	37	1	-25
17	35	7	-6	37	1	-25
17	35	9	-3	34	17	16
17	35	10	1	34	17	16
17	35	12	5	34	17	16
17	35	15	12	34	17	16
17	25	16	-18	22	30	-12
17	27	18	-15	22	30	-12
17	35	10	-5	37	1	-25
17	37	12	-4	34	17	16
17	40	14	-2	34	17	16
19	-32	17	-9	-16	37	17
19	-35	18	-7	-16	37	17
19	-38	19	-5	-16	37	17
19	-41	20	-3	-19	37	20

Patient ID	The coordinate of included channel for ROIs			The coordinate of the corresponding nearest white-matter neighbor reference channel		
19	-40	-1	-2	-19	37	20
19	-42	0	0	-19	37	20
20	37	12	-11	42	-1	-24
20	36	14	-8	42	-1	-24
20	35	16	-5	28	28	17
20	34	18	-2	28	28	17
20	33	20	1	28	28	17
20	32	21	4	28	28	17
20	31	23	7	28	28	17
20	30	25	10	28	28	17
20	29	26	14	28	28	17
21	42	-3	-1	47	0	4
21	34	12	-2	26	27	-5
21	36	14	0	26	27	-5
21	39	17	1	26	27	-5
22	-37	7	-8	-47	0	-21
22	-37	10	-6	-47	0	-21
22	-37	13	-3	-47	0	-21
22	-36	17	-1	-47	0	-21
22	-36	20	1	-47	0	-21
Channels for the amygdala						
1	-26	-5	-27	-38	-6	-26
1	-30	-6	-26	-38	-6	-26
1	-34	-6	-26	-38	-6	-26
3	-16	-2	-21	-37	-5	-22
3	-20	-3	-22	-37	-5	-22
3	-27	-4	-22	-37	-5	-22
3	-31	-5	-21	-37	-5	-22
3	-34	-5	-22	-37	-5	-22
3	21	-1	-26	36	1	-28
3	29	0	-27	36	1	-28
4	15	-5	-19	34	-3	-21
4	19	-5	-19	34	-3	-21
4	23	-4	-20	34	-3	-21
4	-17	-6	-17	-36	-7	-17
4	-20	-7	-17	-36	-7	-17
4	-24	-7	-17	-36	-7	-17
4	-28	-7	-17	-36	-7	-17
4	-32	-7	-17	-36	-7	-17
5	-27	-3	-26	-34	-1	-26
5	-30	-2	-26	-34	-1	-26

Patient ID	The coordinate of included channel for ROIs			The coordinate of the corresponding nearest white-matter neighbor reference channel		
6	-24	-6	-22	-38	-2	-26
6	-27	-5	-23	-38	-2	-26
6	-31	-4	-24	-38	-2	-26
6	-34	-3	-25	-38	-2	-26
9	26	-3	-28	36	-1	-27
9	29	-2	-28	36	-1	-27
9	33	-2	-27	36	-1	-27
10	-24	-6	-23	-34	-3	-24
10	-28	-5	-23	-34	-3	-24
10	-31	-4	-24	-34	-3	-24
11	-22	-6	-22	-35	0	-24
11	-25	-4	-22	-35	0	-24
11	-29	-3	-23	-35	0	-24
11	-32	-1	-23	-35	0	-24
12	-20	-3	-19	-34	-4	-19
12	-23	-3	-19	-34	-4	-19
12	-27	-4	-19	-34	-4	-19
12	-30	-4	-19	-34	-4	-19
13	-19	-11	-17	-34	-11	-18
13	-22	-10	-18	-34	-11	-18
13	-25	-8	-20	-34	-11	-18
13	-28	-6	-21	-34	-2	-23
13	-31	-4	-22	-34	-2	-23
14	22	-1	-26	35	1	-29
14	25	-1	-26	35	1	-29
14	29	0	-27	35	1	-29
14	32	0	-28	35	1	-29
15	-23	-3	-23	-47	-10	-26
15	-27	-2	-24	-47	-10	-26
15	-30	0	-24	-47	-10	-26
16	19	-6	-20	30	1	-23
16	22	-5	-21	30	1	-23
16	24	-3	-22	30	1	-23
16	27	-1	-22	30	1	-23
17	24	-3	-21	37	1	-25
17	27	-2	-22	37	1	-25
17	30	-1	-23	37	1	-25
17	34	0	-24	37	1	-25
17	23	-11	-19	37	-10	-20
18	-27	-3	-25	-44	-18	-18
18	-30	-1	-25	-44	-18	-18

Patient ID	The coordinate of included channel for ROIs			The coordinate of the corresponding nearest white-matter neighbor reference channel		
18	-33	1	-25	-44	-18	-18
20	20	-7	-24	33	-3	-24
20	23	-6	-24	33	-3	-24
20	26	-5	-24	33	-3	-24
20	29	-4	-24	33	-3	-24
22	-31	-8	-24	-37	-5	-23
22	-34	-7	-23	-37	-5	-23
Channels for the IFG						
1	-41	19	-3	-38	-6	-26
1	-45	17	-3	-50	-7	-25
1	-49	16	-3	-50	-7	-25
1	-53	14	-3	-50	-7	-25
1	-56	13	-3	-50	-7	-25
5	-33	27	7	-31	32	14
5	-32	29	10	-31	32	14
6	-40	12	1	-44	-2	3
6	-44	14	2	-44	-2	3
6	-47	15	2	-44	-2	3
7	41	7	7	22	10	40
7	44	7	9	22	10	40
7	47	7	12	22	10	40
7	50	7	14	22	10	40
7	53	6	17	22	10	40
7	56	6	19	22	10	40
7	59	6	21	22	10	40
7	62	6	24	22	10	40
8	46	18	6	25	43	-3
8	49	19	8	25	43	-3
8	53	19	10	25	43	-3
8	56	20	12	25	43	-3
8	59	21	14	25	45	0
9	32	39	-19	29	26	4
9	35	39	-18	29	26	4
9	39	39	-17	29	26	4
9	43	40	-17	29	26	4
9	46	40	-16	29	26	4
11	-38	14	7	-39	1	-24
11	-38	17	11	-39	1	-24
11	-39	19	14	-39	1	-24
11	-39	22	18	-39	1	-24
11	-42	10	2	-39	1	-24

Patient ID	The coordinate of included channel for ROIs			The coordinate of the corresponding nearest white-matter neighbor reference channel		
11	-44	10	4	-39	1	-24
11	-47	9	7	-42	3	-25
11	-49	8	10	-45	-22	-4
11	-52	8	13	-45	-22	-4
11	-54	7	16	-45	-22	-4
11	-57	7	19	-45	-22	-4
11	-59	6	22	-45	-22	-4
11	-62	6	25	-45	-22	-4
12	-29	26	4	-26	26	2
12	-32	27	6	-26	26	2
12	-35	28	8	-26	26	2
12	-37	28	10	-26	26	2
12	-40	29	12	-26	26	2
12	-43	30	14	-31	17	24
12	-46	31	16	-31	17	24
12	-29	38	-14	-20	24	-3
12	-32	38	-13	-26	26	2
12	-36	39	-12	-26	26	2
12	-39	39	-11	-26	26	2
12	-42	39	-10	-26	26	2
12	-46	40	-9	-26	26	2
12	-49	40	-7	-26	26	2
12	-53	41	-6	-26	26	2
13	-27	32	-12	-37	0	-24
13	-30	33	-9	-37	0	-24
13	-33	33	-7	-37	0	-24
13	-36	33	-4	-37	0	-24
13	-38	33	-2	-37	0	-24
13	-41	33	1	-37	0	-24
13	-44	34	4	-37	0	-24
14	59	2	17	33	-6	10
14	62	2	20	33	-6	10
16	40	46	-14	27	23	14
16	43	47	-13	27	23	14
16	46	48	-11	27	23	14
17	41	33	3	31	41	2
17	43	35	6	32	43	5
17	45	38	8	32	43	5
17	47	40	11	32	43	5
17	50	42	13	32	43	5
17	57	25	10	34	17	16

Patient ID	The coordinate of included channel for ROIs			The coordinate of the corresponding nearest white-matter neighbor reference channel		
17	59	27	11	34	17	16
19	-44	21	-1	-19	37	20
19	-47	22	1	-19	37	20
19	-50	22	2	-19	37	20
19	-53	23	4	-19	37	20
19	-56	24	6	-19	37	20
21	30	32	6	26	27	-5
21	50	2	6	47	0	4
21	53	3	9	55	4	11
21	61	7	16	58	6	14
21	42	20	3	26	27	-5
21	45	23	5	26	27	-5
21	48	26	6	55	4	11
21	50	29	8	58	6	14
22	-36	24	3	-47	0	-21
22	-36	27	6	-47	0	-21
22	-36	30	8	-47	0	-21

References

Abrams, D. A. et al. Inter-subject synchronization of brain responses during natural music listening. *Eur. J. Neurosci.* **37**, 1458-1469 (2013).

Allsop, S.A. et al. Corticoamygdala transfer of socially derived information gates observational learning. *Cell* **173**, 1329-1342 (2018).

Aru, J. et al. Untangling cross-frequency coupling in neuroscience. *Curr. Opin. Neurobiol.* **31**, 51-61 (2015).

Avenanti, A., Buetti, D., Galati, G. & Aglioti, S. M. Transcranial magnetic stimulation highlights the sensorimotor side of empathy for pain. *Nat. Neurosci.* **8**, 955-960 (2005).

Ball, T., Kern, M., Mutschler, I., Aertsen, A. & Schulze-Bonhage, A. Signal quality of simultaneously recorded invasive and non-invasive EEG. *NeuroImage* **46**, 708-716 (2009).

Berman, J.I. et al. Variable bandwidth filtering for improved sensitivity of cross-frequency coupling metrics. *Brain Connect.* **2**, 155-163 (2012).

Bichot, N. P., Heard, M. T., DeGennaro, E. M. & Desimone, R. A source for feature-based attention in the prefrontal cortex. *Neuron* **88**, 832-844 (2015).

Bisenius, S. et al. Predicting primary progressive aphasia with support vector machine approaches in structural MRI data. *NeuroImage Clin.* **14**, 334-343 (2017).

Caggiano, V., Fleischer, F., Pomper, J. K., Giese, M. A. & Thier, P. Mirror neurons in monkey premotor area F5 show tuning for critical features of visual causality perception. *Curr. Biol.* **26**, 3077-3082 (2016).

Canolty, R.T. & Knight, R.T. The functional role of cross-frequency coupling. *Trends Cogn. Sci.* **14**, 506-515 (2010).

Chen, C. H., Hung, K. S., Chung, Y. C. & Yeh, M. L. Mind-body interactive qigong improves physical and mental aspects of quality of life in inpatients with stroke: A randomized control study. *Eur. J. Cardiovasc. Nurs.* **18**, 658-666 (2019).

Chen, G. et al. Untangling the relatedness among correlations, part I: nonparametric approaches to inter-subject correlation analysis at the group level. *NeuroImage* **142**, 248-259 (2016).

Chen, S. et al. Theta oscillations synchronize human medial prefrontal cortex and

amygdala during fear learning. *Sci. Adv.* **7**, eabf4198 (2021).

Chen, Y. C., Chen, C. C., Decety, J. & Cheng, Y. Aging is associated with changes in the neural circuits underlying empathy. *Neurobiol. Aging* **35**, 827-836 (2014).

Chu, C. et al. Does feature selection improve classification accuracy? Impact of sample size and feature selection on classification using anatomical magnetic resonance images. *Neuroimage* **60**, 59-70 (2012).

Cohen, M. X. *Analyzing neural time series data: theory and practice*. (MIT press, Cambridge, 2014).

Combrisson, E. & Jerbi, K. Exceeding chance level by chance: The caveat of theoretical chance levels in brain signal classification and statistical assessment of decoding accuracy. *J. Neurosci. Methods* **250**, 126-136 (2015).

Conroy, B. R., Singer, B. D., Guntupalli, J. S., Ramadge, P. J. & Haxby, J. V. Inter-subject alignment of human cortical anatomy using functional connectivity. *NeuroImage* **81**, 400-411 (2013).

de Greck, M. et al. Neural substrates underlying intentional empathy. *Soc. Cogn. Affect. Neurosci.* **7**, 135-144 (2012).

Donner, T. H. & Siegel, M. A framework for local cortical oscillation patterns. *Trends Cogn. Sci.* **15**, 191-199 (2011).

Fallon, N., Roberts, C. & Stancak, A. Shared and distinct functional networks for empathy and pain processing: a systematic review and meta-analysis of fMRI studies. *Soc. Cogn. Affect. Neurosci.* **15**, 709-723 (2020).

Fan, Y. & Han, S. Temporal dynamic of neural mechanisms involved in empathy for pain: an event-related brain potential study. *Neuropsychologia* **46**, 160-173 (2008).

Fan, Y. et al. Multivariate examination of brain abnormality using both structural and functional MRI. *NeuroImage* **36**, 1189-1199 (2007).

Fan, Y., Duncan, N. W., de Greck, M. & Northoff, G. Is there a core neural network in empathy? An fMRI based quantitative meta-analysis. *Neurosci. Biobehav. Rev.* **35**, 903-911 (2011).

Feng, C. et al. Social hierarchy modulates neural responses of empathy for pain. *Soc. Cogn. Affect. Neurosci.* **11**, 485-495 (2016).

Fries, P. A mechanism for cognitive dynamics: neuronal communication through

neuronal coherence. *Trends Cogn. Sci.* **9**, 474-480 (2005).

Gonzalez-Liencre, C., Brown, E. C., Tas, C., Breidenstein, A. & Brüne, M. Alterations in event-related potential responses to empathy for pain in schizophrenia. *Psychiatry Res.* **241**, 14-21 (2016).

Griffiths, B. J. et al. Directional coupling of slow and fast hippocampal gamma with neocortical alpha/beta oscillations in human episodic memory. *Proc. Natl. Acad. Sci.* **116**, 21834-21842 (2019).

Han, H. & Jiang, X. Overcome support vector machine diagnosis overfitting. *Cancer Inform.* **13**, CIN-S13875 (2014).

Han, S., Fan, Y. & Mao, L. Gender difference in empathy for pain: an electrophysiological investigation. *Brain Res.* **1196**, 85-93 (2008).

Huang, Y., Hu, P. & Deng, H. Empathic concern induction modulates behavioral ratings and neural responses to harm-related moral judgment: An event-related potentials study. *Behav. Brain Res.* **446**, 114397 (2023).

Hughes, G. On the mean accuracy of statistical pattern recognizers. *IEEE Trans. Inf. Theory* **14**, 55-63 (1968).

Hülsemann, M. J., Naumann, E. & Rasch, B. Quantification of phase-amplitude coupling in neuronal oscillations: comparison of phase-locking value, mean vector length, modulation index, and generalized-linear-modeling-cross-frequency-coupling. *Front. Neurosci.* **13**, 573 (2019).

Hutchison, W. D., Davis, K. D., Lozano, A. M., Tasker, R. R. & Dostrovsky, J. O. Pain-related neurons in the human cingulate cortex. *Nat. Neurosci.* **2**, 403-405 (1999).

Hyon, R., Kleinbaum, A. M. & Parkinson, C. Social network proximity predicts similar trajectories of psychological states: evidence from multi-voxel spatiotemporal dynamics. *NeuroImage* **216**, 116492 (2020).

Jensen, O. & Colgin, L.L. Cross-frequency coupling between neuronal oscillations. *Trends Cogn. Sci.* **11**, 267-269 (2007).

Kam, J. W. et al. Default network and frontoparietal control network theta connectivity supports internal attention. *Nat. Hum. Behav.* **3**, 1263-1270 (2019).

Knoll, L. J., Magis-Weinberg, L., Speekenbrink, M. & Blakemore, S. J. Social influence on risk perception during adolescence. *Psychol. Sci.* **26**, 583-592 (2015).

Kriegeskorte, N., Simmons, W. K., Bellgowan, P. S. & Baker, C. I. Circular analysis in systems neuroscience: the dangers of double dipping. *Nat. Neurosci.* **12**, 535-540 (2009).

Lachaux, J. P. How Many Data Do I Need for an iEEG Study? Treasure Maps and the Status of Variability. In *Intracranial EEG: A Guide for Cognitive Neuroscientists*, 125-142 (2023).

LaConte, S., Strother, S., Cherkassky, V., Anderson, J. & Hu, X. Support vector machines for temporal classification of block design fMRI data. *NeuroImage* **26**, 317-329 (2005).

Legendre, G., Andrillon, T., Koroma, M. & Kouider, S. Sleepers track informative speech in a multitalker environment. *Nat. Hum. Behav.* **3**, 274-283 (2019).

Levitt, J. et al. Pain phenotypes classified by machine learning using electroencephalography features. *NeuroImage* **223**, 117256 (2020).

Li, J. et al. Anterior-posterior hippocampal dynamics support working memory processing. *J. Neurosci.* **42**, 443-453 (2022).

Libkuman, T. M., Otani, H., Kern, R., Viger, S. G. & Novak, N. Multidimensional normative ratings for the international affective picture system. *Behav. Res. Methods* **39**, 326-334 (2007).

Lin, C., Keles, U. & Adolphs, R. Four dimensions characterize attributions from faces using a representative set of English trait words. *Nat. Commun.* **12**, 5168 (2021).

Linn, K. A. et al. Control-group feature normalization for multivariate pattern analysis of structural MRI data using the support vector machine. *NeuroImage* **132**, 157-166 (2016).

Mann, K., Gallen, C. L. & Clandinin, T. R. Whole-brain calcium imaging reveals an intrinsic functional network in *Drosophila*. *Curr. Biol.* **27**, 2389-2396 (2017).

Manssuer, L. et al. Integrated Amygdala, Orbitofrontal and Hippocampal Contributions to Reward and Loss Coding Revealed with Human Intracranial EEG. *J. Neurosci.* **42**, 2756-2771 (2022).

Maris, E. & Oostenveld, R. Nonparametric statistical testing of EEG- and MEG-data. *J. Neurosci. Meth.* **164**, 177-190 (2007).

Mercier, M. R. et al. Advances in human intracranial electroencephalography research, guidelines and good practices. *NeuroImage* **260**, 119438 (2022).

Mo, J. et al. Neural underpinnings of default mode network on empathy revealed by intracranial stereoelectroencephalography. *Psychiatry Clin.* **76**, 659-666 (2022).

Morelli, S. A. & Lieberman, M. D. The role of automaticity and attention in neural processes underlying empathy for happiness, sadness, and anxiety. *Front. Hum. Neurosci.* **7**, 160 (2013).

Mu, Y., Fan, Y., Mao, L. & Han, S. Event-related theta and alpha oscillations mediate empathy for pain. *Brain Res.* **1234**, 128-136 (2008).

Mukamel, E. A. et al. A transition in brain state during propofol-induced unconsciousness. *J. Neurosci.* **34**, 839-845 (2014).

Nieuwenhuis, M. et al. Classification of schizophrenia patients and healthy controls from structural MRI scans in two large independent samples. *NeuroImage* **61**, 606-612 (2012).

Oehrn, C. R. et al. Direct electrophysiological evidence for prefrontal control of hippocampal processing during voluntary forgetting. *Curr. Biol.* **28**, 3016-3022 (2018).

Oehrn, C. What Are the Pros and Cons of ROI Versus Whole-Brain Analysis of iEEG Data? In *Intracranial EEG: A Guide for Cognitive Neuroscientists*, 475-486 (2023).

Pallarés, V. et al. Extracting orthogonal subject-and condition-specific signatures from fMRI data using whole-brain effective connectivity. *NeuroImage* **178**, 238-254 (2018).

Parvizi, J. & Kastner, S. Promises and limitations of human intracranial electroencephalography. *Nat. Neurosci.* **21**, 474-483 (2018).

Pereira, F., Mitchell, T. & Botvinick, M. Machine learning classifiers and fMRI: a tutorial overview. *NeuroImage* **45**, S199-S209 (2009).

Quandt, F. et al. Single trial discrimination of individual finger movements on one hand: a combined MEG and EEG study. *NeuroImage* **59**, 3316-3324 (2012).

Shao, Y. & Lunetta, R. S. Comparison of support vector machine, neural network, and CART algorithms for the land-cover classification using limited training data points. *ISPRS J. Photogramm. Remote Sens.* **70**, 78-87 (2012).

Shirvalkar, P. et al. First-in-human prediction of chronic pain state using intracranial neural biomarkers. *Nat. Neurosci.* **26**, 1090-1099 (2023).

Siegel, M., Donner, T. H. & Engel, A. K. Spectral fingerprints of large-scale neuronal

interactions. *Nat. Rev. Neurosci.* **13**, 121-134 (2012).

Snyder, A. C., Morais, M. J., Willis, C. M. & Smith, M. A. Global network influences on local functional connectivity. *Nat. Neurosci.* **18**, 736-743 (2015).

Sonkusare, S. et al. Frequency dependent emotion differentiation and directional coupling in amygdala, orbitofrontal and medial prefrontal cortex network with intracranial recordings. *Mol. Psychiatry* **28**, 1636-1646 (2023).

Soyman, E. et al. Intracranial human recordings reveal association between neural activity and perceived intensity for the pain of others in the insula. *Elife* **11**, e75197 (2022).

Stangl, M. et al. Boundary-anchored neural mechanisms of location-encoding for self and others. *Nature* **589**, 420-425 (2021).

Stelzer, J., Chen, Y. & Turner, R. Statistical inference and multiple testing correction in classification-based multi-voxel pattern analysis (MVPA): random permutations and cluster size control. *NeuroImage* **65**, 69-82 (2013).

Sterpenich, V. et al. Reward biases spontaneous neural reactivation during sleep. *Nat. Commun.* **12**, 4162 (2021).

Ten Oever, S., Sack, A. T., Oehr, C. R. & Axmacher, N. An engram of intentionally forgotten information. *Nat. Commun.* **12**, 6443 (2021).

Timmers, I. et al. Is empathy for pain unique in its neural correlates? A meta-analysis of neuroimaging studies of empathy. *Front. Behav. Neurosci.* **12**, 289 (2018).

Valizadeh, S. A., Riener, R., Elmer, S. & Jäncke, L. Decrypting the electrophysiological individuality of the human brain: Identification of individuals based on resting-state EEG activity. *NeuroImage* **197**, 470-481 (2019).

Vandana, C. P. & Chikkamannur, A. A. Feature selection: An empirical study. *Int. J. Eng. Technol.* **69**, 165-170 (2021).

Varoquaux, G. Cross-validation failure: Small sample sizes lead to large error bars. *Neuroimage* **180**, 68-77 (2018).

Weston, J. et al. Feature selection for SVMs. In *Proceedings of the 13th International Conference on Neural Information Processing Systems*, 647-653 (2000).

Wu, H. et al. Identifying patients with cognitive motor dissociation using resting-state temporal stability. *NeuroImage* **272**, 120050 (2023).

Xu, X., Zuo, X., Wang, X. & Han, S. Do you feel my pain? Racial group membership modulates empathic neural responses. *J. Neurosci.* **29**, 8525-8529 (2009).

Yang, Z. et al. Individualized psychiatric imaging based on inter-subject neural synchronization in movie watching. *NeuroImage* **216**, 116227 (2020).

Zandvoort, C. S. & Nolte, G. Defining the filter parameters for phase-amplitude coupling from a bispectral point of view. *J. Neurosci. Methods* **350**, 109032 (2021).

Zhang, M. et al. Glutamatergic synapses from the insular cortex to the basolateral amygdala encode observational pain. *Neuron* **110**, 1-16 (2022).

Zhang, R. et al. Temporal-spatial characteristics of phase-amplitude coupling in electrocorticogram for human temporal lobe epilepsy. *Clin. Neurophysiol.* **128**, 1707-1718 (2017).

Zheng, J. et al. Amygdala-hippocampal dynamics during salient information processing. *Nat. Commun.* **8**, 1-11 (2017).

Zheng, J. et al. Multiplexing of theta and alpha rhythms in the amygdala-hippocampal circuit supports pattern separation of emotional information. *Neuron* **102**, 887-898 (2019).

Zhou, Y. & Han, S. Neural dynamics of pain expression processing: Alpha-band synchronization to same-race pain but desynchronization to other-race pain. *NeuroImage* **224**, 117400 (2021).

REVIEWER COMMENTS

Reviewer #1 (Remarks to the Author):

General Comments:

The authors have spent great efforts on addressing the concerns and questions from the last round of review. There is no doubt that the manuscript has been improved from the last version. However, there are still major concerns, including the limited trial number issues, and some methodological problems, that haven't been fully addressed. This greatly affects the reliability of the reported results and interpretation of the findings. Besides, great improvement in writing is still needed for the current version of the manuscript, with precise descriptions of the results, methods, and understandable interpretation of the current findings. I don't think the manuscript right now fits the high standard of Nature Communication.

Major issues:

Concerns about the low trial number. I appreciate the authors' efforts in addressing this issue. However, I don't think this issue is fully addressed with the evidence and justifications that the authors provided. Authors claim that "potential limitation in trial numbers can be compensated for by the exceptional SNR observed in iEEG data compared to fMRI or scalp EEG data". I agree that iEEG has stronger SNR compared to fMRI or scalp EEG data, but it doesn't prove the point that the low trial number is sufficient for providing reliable results. Authors quoted Fig. 11 from Mercier et al, 2022 (also adapted as Figure. R1) to show that averaged neural responses did not change significantly with different trial numbers. Note that, the results in Fig. 11 from Mercier et al, 2022 are ERPs recorded in the primary auditory cortex in response to pure tones. It is known that sensory areas, like the primary auditory cortex, tend to have consistent neural responses to simple sensory inputs, which enables signal trial decoding analyses for Brain-Computer Interface application. However, such trial-by-trial consistency is much lower in higher cognitive regions during complex cognitive tasks. With a limited trial number (10 trials per condition in the current study), the observed averaged results can be easily driven by specific 1 or 2 trials. To avoid this potential bias, it is helpful for authors to generate similar figures like Fig. R1, using their collected data to show the single trial liability for empathy-related power change, power correlation, transfer entropy, and phase-amplitude coupling.

Problematic methodology of decoding analysis.

First, it is confusing to me that why the authors built two classification models. As mentioned in the Methods Section decoding analysis, authors first built a classification model using 70% of the data as a feature-identifying dataset and 30% of data as a decoding dataset. Then the authors built another classification model using SVM along with a 5-fold cross-validation procedure. What is the relationship between these two models?

Second, the use of a 5-fold cross-validation procedure is not valid. K-fold cross-validation is beneficial to avoid overfitting problems. It should be used to evaluate the robustness of a model during training but for generating testing results. For example, the dataset should be split into a training dataset and a testing dataset first. Then the 5-fold cross-validation procedure applies to the training dataset only to assess the model performance and finalize the model structure before introducing the novel testing data to the model to assess its generalization ability.

Third, the limited trial number in this study for SVM decoding analysis is concerning. I agree with the authors that SVM requires fewer trial numbers compared with other multi-layer perceptron neural networks. However, 10 trials per condition still seem to be a challenging number. In Figure R2, which the authors adapted from Shao and Lunetta 2012, the minimum training sample size is 20 trials per class, not even including the number of testing trials. The authors use a random resampling approach to generate a simulated dataset and test the relationship between decoding accuracy and trial numbers. This approach assumes that the newly collected data will follow the same distribution of the existing data, which is problematic especially when the existing dataset has so limited sample size that might not represent enough observational scenarios. For example, when you throw a dice twice and get a number 1 and a number 2 (real observation), if you use these two real observations to resample and generate more data, most likely your simulated data will be a bunch of 1 and 2. However, the real distribution of throwing a dice should cover numbers 1 to 6, which can never be simulated with limited real observations.

Some descriptions or interpretations of the results are not precise.

The term of “spatial integration” occurs several times in the text, e.g., in line 68. What do authors refer to?

The description of directionality analysis is not clear. For example, in line 363, what does “beta information transfer” mean?

Line 363, “suggesting reduced beta information transfer in both directions”.. Reduced beta information transfer relative to what?

Line 365, “the effect of pain was direction sensitive”... what is the meaning of direction sensitive? Please just describe the result itself.

Line 372, “enhanced information transmission between ACC and amygdala”... What information is transmitted here?

Line 672, what results in the study support the claim of “two DISTINCT inter-regional communication mechanisms”?

Line 714, “we observed weaker functional communications between the ACC and AI”, weaker compared to what?

Some of the questions are not fully addressed

As pointed out earlier, in the current version of the paper, it is still not clear how to understand the results from the directionality analyses. The authors mentioned that “the directionality analysis served as a supplementary analysis to power coupling analysis and was not the primary focus of the current study.” If this is the case, what is the point to include in the main figure, and how does that help better understand the primary power coupling analysis?

As pointed out earlier, the interpretation of the results is not clear. I appreciate that the authors made significant changes to the discussion section. However, the current version is extremely long, and still very hard for readers to understand the main findings of the paper. Not to mention the underlying potential explanation of the results, how that links to previous findings in other studies, and what specific questions inspired by the current findings for future studies.

Reviewer #2 (Remarks to the Author):

The authors have taken into appropriate consideration our remarks and changed the ms accordingly.

Reviewer #3 (Remarks to the Author):

The revised manuscript "Intracranial EEG signals disentangle multi-areal neural dynamics of vicarious pain perception" addressed all my concerns and I don't have any remaining comments or questions.

Reviewer #4 (Remarks to the Author):

I co-reviewed this manuscript with one of the reviewers who provided the listed reports as part of the Nature Communications initiative to facilitate training in peer review and appropriate recognition for co-reviewers.

General response: We are very grateful to you and other reviewers for the effort and time you put into reviewing our manuscript. We are delighted to learn that other reviewers were fully satisfied with our revised manuscript, and reviewer#1 only has a few remaining points that have not been fully addressed in the last round. Accordingly, we have provided further clarifications, additional analyses, and supporting results below to address these points. Specifically, we conducted *a series of trial-wise analyses*, which consistently demonstrated the reported neural patterns across the majority of trials. These analyses suggested that *the observed effects were reliable and not driven by only a few specific trials* (supporting results detailed in **Figure R1-R3**). We are pleased to report that these analyses underscore both the reliability of our findings and their consistency across trials. Additionally, we clarified methodological details and issues related to our decoding analysis to avoid oversimplification or potential confusion. We revised descriptions of our results to ensure clarity and accuracy. In terms of the discussion section, we made efforts to simplify our language in order to make it more intuitive and easy to understand. We also provided a summary of our revised *Discussion* which clarifies how we discussed the key findings by explaining their significance, discussing their connection with existing literature, and proposing future research questions.

➤ **Point-to-point response**

#Point 1. “Major issues: Concerns about the low trial number. I appreciate the authors’ efforts in addressing this issue. However, I don’t think this issue is fully addressed with the evidence and justifications that the authors provided. Authors claim that “potential limitation in trial numbers can be compensated for by the exceptional SNR observed in iEEG data compared to fMRI or scalp EEG data”. I agree that iEEG has stronger SNR compared to fMRI or scalp EEG data, but it doesn’t prove the point that the low trial number is sufficient for providing reliable results. Authors quoted Fig. 11 from Mercier et al, 2022 (also adapted as Figure. R1) to show that averaged neural responses did not change significantly with different trial numbers. Note that, the results in Fig. 11 from Mercier et al, 2022 are ERPs recorded in the primary auditory cortex in response to pure tones. It is known that sensory are, like the primary auditory cortex, tend to have consistent neural responses to simple sensory inputs, which enables signal trial decoding analyses for Brain-Computer Interface application. However, such trial-by-trial consistency is much lower in higher cognitive regions during complex cognitive tasks. With a limited trial number (10 trials per condition in the current study), the observed averaged results can be easily driven by specific 1 or 2 trials. To avoid this potential bias, it is helpful for authors to generate similar figures like Fig. R1, using their collected data to show the single trial liability for empathy-related power change, power correlation, transfer entropy, and phase-amplitude coupling.”

Response: To address Reviewer 1’s concerns, we conducted *a series of trial-wise analyses*. We are pleased to report that these analyses provided *compelling evidence for the reliability* of our main findings (detailed below). Specifically, we examined the consistency of our neural patterns across different trial numbers and presented empirical evidence to support high trial-by-trial consistency in our neural data (**Figure R1A-E**;

Figure R2A-D; Figure R3A-D). Furthermore, we checked the neural pattern for each single trial and performed sensitivity analyses and demonstrated that our main findings were not “driven by specific 1 or 2 trials” (Figure R1F-O, Figure R2E-L, Figure R3E-L).

➤ **Reliability of our main findings**

The reliability of observed effects was recognized as *the consistency of measurements* (Lavrakas, 2008; Elliott et al., 2020; Noble et al., 2020). Specifically, in terms of trial-wise reliability, it refers to the extent to which a measure produces similar patterns across trials (Muller et al., 2014; Ter Wal M et al., 2021). The measure is considered reliable if it generates similar patterns across different trials. Accordingly, we conducted trial-wise analysis for each main finding to assess the reliability and consistency of our main neural findings across trials.

First, we examined neural responses with varying number of averaged trials. A high trial-by-trial consistency would manifest as similar patterns when including all or a subset of trials. We investigated the conditional differences in neural activity between painful and non-painful conditions by including different number of trials in the analyses. Notably, even when considering partial trials (e.g., 40%, 60%, or 80% of the total trials), we observed that the reported neural findings remained statistically significant (Figure R1A-E, replicating the results shown in Figure 2e-h; Figure R2A-D, replicating the patterns presented in Figure 3a-c; Figure R3A-D, replicating the results shown in Figure 4a-c). **These findings underscore both the reliability of our main findings and their consistency across trials.**

Second, in order to mitigate the potential influence of a limited number of specific trials on our neural results, we examined neural patterns at the single-trial level. Remarkably, we consistently observed the reported neural patterns across the majority of trials (Figure R1F-J, Figure R2E-H, and Figure R3E-H). Moreover, to assess the impact of extreme values on our results, we performed a sensitivity analysis by excluding trials with extreme conditional differences. If our findings were driven solely by 1-2 specific trials, removing these extreme trials would result in the absence of the reported neural patterns. However, this set of analyses demonstrated that all identified neural findings remained unchanged (Figure R1K-O, replicating the results shown in Figure 2e-h; Figure R2I-L, replicating the findings reported in Figure 3a-c; Figure R3I-L, replicating the findings shown in Figure 4a-c).

With these lines of evidence, we were confident in the robustness of our neural findings, which demonstrated consistent patterns across trials and could not be attributed to a mere subset of specific trials. We included the single-trial neural patterns (Figure R1F-J, Figure R2E-H, and Figure R3E-H) in the revised supplementary materials (Fig. S10-12) to show the consistent patterns across trials.

Figure R1. Consistent spectro-temporal power patterns in the AI, ACC, amygdala, and IFG across trials as shown in main Fig. 2. (A-E) The conditional differences of spectro-temporal power remained statistically significant even when considering partial trials (40%, 60% or 80%). Orange (violet) bars indicate normalized power for painful (non-painful) conditions. (F-J) The reported neural patterns were consistently observed across the majority of trials. Each orange (painful) and violet (non-painful) dot indicate single-trial normalized power. (K-O) Excluding extreme trials did not change the spectro-temporal power results. Orange (violet) split-half violin plots indicate the probability density of the averaged normalized power in the painful (non-painful) conditions.

Figure R2. Consistent power correlations patterns between ACC, AI, and amygdala across trials as shown in main Fig. 3. (A-D) The conditional differences of power correlations remained statistically significant when including partial trials (40%, 60% or 80%). Orange (violet) bars indicate Fisher-z-transformed power correlation values for painful (non-painful) conditions. (E-H) The reported neural patterns were consistently observed across the majority of trials. Each orange (painful) and violet (non-painful) dot indicate single-trial Fisher-z-transformed power correlation values. (I-L) The power correlation results remained unchanged when excluding extreme trials. Orange (violet) split-half violin plots indicate the probability density of the averaged Fisher-z-transformed power correlation values in the painful (non-painful) conditions.

Figure R3. Consistent phase-amplitude coupling (PAC) patterns ACC/AI/amygdala and IFG across trials as shown in main Fig. 4. (A-D) The conditional differences of PAC values remained statistically significant when including partial trials (40%, 60% or 80%). Orange (violet) bars indicate Fisher-z-transformed PAC values for painful (non-painful) conditions. **(E-H)** The reported neural patterns were consistently observed across the majority of trials. Each orange (painful) and violet (non-painful) dot indicate single-trial Fisher-z-transformed PAC values. **(I-L)** The PAC results remained unchanged when excluding extreme trials. Orange (violet) split-half violin plots indicate the probability density of the averaged Fisher-z-transformed PAC values in the painful (non-painful) condition.

Point 2.

2a. “First, it is confusing to me that why the authors built two classification models. As mentioned in the Methods Section decoding analysis, authors first built a classification model using 70% of the data as a feature-identifying dataset and 30% of data as a decoding dataset. Then the authors built another classification model using SVM along with a 5-fold cross-validation procedure. What is the relationship between these two models?”

2b. “Second, the use of a 5-fold cross-validation procedure is not valid. K-fold cross-validation is beneficial to avoid overfitting problems. It should be used to evaluate the robustness of a model during training but for generating testing results. For example, the dataset should be split into a training dataset and a testing dataset first. Then the 5-fold cross-validation procedure applies to the training dataset only to assess the model performance and finalize the model structure before introducing the novel testing data to the model to assess its generalization ability.”

Response: To avoid potential pitfalls in decoding analysis, as previously highlighted in the literature (e.g., the problems of overfitting and information leakage; Kriegeskorte et al., 2009; Pereira et al., 2009; Varoquaux et al., 2017), we employed a relatively complex analysis pipeline. We thank you for raising these questions, which prompted us to revise the description of decoding methodology to avoid oversimplification and potential confusion or misunderstanding.

In the revised *Method* (*Pages 42-43, the Decoding analysis section*), we have rephrased the explanation of our decoding approach. Specifically, we have clarified our analysis pipeline, and provided clearer details about our 70%-30% data split procedure and cross-validation procedure.

➤ **70%-30% data split procedure** (*Pages 42, Para 2-4*)

Please allow us to clarify that we did not construct two models. We employed the 70%-30% data split procedure to randomly split our data into two independent sub-datasets, which were separately used to perform feature selection and model construction. Specifically, we randomly split our data into two independent sub-datasets, with 70% of the data as the feature-selection dataset, and the remaining 30% of the data as the decoding dataset. It should be noted that the feature-selection dataset was only used to identify informative features, rather than constructing classification models. The decoding dataset was only used to construct the classification model to decode stimulus types. The use of a separate sub-dataset, instead of the full dataset, to select relevant features has been suggested as a way to avoid potential problems of information leakage (Kriegeskorte et al., 2009; Pereira et al., 2009; Varoquaux et al., 2017).

➤ **Cross-validation procedure** (*Pages 43, Para 2*)

We agree with your perspectives on the description of the K-fold cross-validation procedure. Our approach indeed was consistent with what you described, and we apologize for any lack of clarity in our previous description of the cross-validation procedure that may have caused misunderstanding of our cross-validation procedure. In fact, our utilization of the cross-validation procedure aligns with your viewpoints

and is consistent with previous studies (Gulli et al., 2019; Prioix et al., 2020; Wang et al., 2023; Benisty et al., 2024). We have now provided a clear elaboration on the objective and workflow of our cross-validation procedure in the revised *Methods* (Pages 43, Para 2).

Point 2c. “Third, the limited trial number in this study for SVM decoding analysis is concerning. I agree with the authors that SVM requires fewer trial numbers compared with other multi-layer perceptron neural networks. However, 10 trials per condition still seem to be a challenging number. In Figure R2, which the authors adapted from Shao and Lunetta 2012, the minimum training sample size is 20 trials per class, not even including the number of testing trials. The authors use a random resampling approach to generate a simulated dataset and test the relationship between decoding accuracy and trial numbers. This approach assumes that the newly collected data will follow the same distribution of the existing data, which is problematic especially when the existing dataset has so limited sample size that might not represent enough observational scenarios. For example, when you throw a dice twice and get a number 1 and a number 2 (real observation), if you use these two real observations to resample and generate more data, most likely your simulated data will be a bunch of 1 and 2. However, the real distribution of throwing a dice should cover numbers 1 to 6, which can never be simulated with limited real observations.”

Response: We thank you for raising this concern, which motivates us to take a closer examination of the impact of trial number on our decoding results. In preparing the revision, we conducted additional analyses and presented empirical evidence that the current number of trials had only a minimal impact on the robustness of the results. Specifically, we repeated the decoding analysis by including partial trials in the analyses to investigate the influence of smaller trial number. We found that our classification model consistently maintained its ability to discriminate between painful and non-painful stimuli above chance across the range of 8 to 20 trials (all $ps < 0.001$; **Figure R4**). This result demonstrated the robustness of our classification model, as successful decoding remained even with smaller trial numbers.

In addition, we observed a significant increase in decoding accuracy as the trial number increased: the decoding accuracy showed a significant improvement for every increment of two trials (**Figure R4**; highlighted by the grey shadow) when the trial number was below 16. Critically, when the training sample size reached 16 or above, further increases in training samples did not lead to a significant improvement in decoding performance (all $ps > 0.05$; **Figure R4**; highlighted by the yellow shadow). This plateau of decoding accuracy suggested that the model has converged, providing evidence supporting the sufficiency of our trial number to achieve stable, significant decoding performance. These results strengthened our confidence in the reliability and robustness of our decoding results.

Figure R4. Decoding accuracy with partial trials. The purple solid-dotted line indicates the decoding accuracy across a range of 8 to 20 trials with error bars representing standard error. The black dash-dotted line indicates the statistical threshold of $p < 0.05$ for each sample size. Grey (orange) shadow highlights the range of sample size in which the improvement of decoding accuracy was (in)significant for every increment of two trials. * $p < 0.05$, ** $p < 0.01$, *** $p < 0.001$, NS, insignificant

Minor Suggestions #1. “Some descriptions or interpretations of the results are not precise.

- a) The term of “spatial integration” occurs several times in the text, e.g., in line 68. What do authors refer to?
- b) The description of directionality analysis is not clear. For example, in line 363, what does “beta information transfer” mean?
- c) Line 363, “suggesting reduced beta information transfer in both directions”... Reduced beta information transfer relative to what?
- d) Line 365, “the effect of pain was direction sensitive”... what is the meaning of direction sensitive? Please just describe the result itself.
- e) Line 372, “enhanced information transmission between ACC and amygdala”... What information is transmitted here?
- f) Line 672, what results in the study support the claim of “two DISTINCT inter-regional communication mechanisms”?
- g) Line 714, “we observed weaker functional communications between the ACC and AI”, weaker compared to what?”

Response: We greatly appreciate the reviewer’s careful reading of our manuscript. Accordingly, we have removed confusing terms mentioned in point #a and point #f from the revised manuscript. We also have revised all relevant sentences to ensure precise descriptions of our results and mitigate any potential confusions (points #b,c,d,e, Page 14 Para. 3; point #g, Page 25 Para.4).

Minor Suggestion #2. “As pointed out earlier, in the current version of the paper, it is still not clear how to understand the results from the directionality analyses. The authors mentioned that “the directionality analysis served as a supplementary analysis to power coupling analysis and was not the primary focus of the current study.” If this is the case, what is the point to include in the main figure, and how does that help better understand the primary power coupling analysis?”.

Response: We appreciate your suggestion. Following this suggestion, we have now presented the results of the directionality analysis in the revised supplementary figure (Figure S5) instead of being included in the main figure. In addition, we have removed the corresponding discussions on the results of the directionality analysis from the *Discussion* section, this may also contribute to enhancing the comprehensibility of the *Discussion*.

Minor Suggestion #3. “As pointed out earlier, the interpretation of the results is not clear. I appreciate that the authors made significant changes to the discussion section. However, the current version is extremely long, and still very hard for readers to understand the main findings of the paper. Not to mention the underlying *potential explanation* of the results, how that *links to previous findings* in other studies, and what specific questions inspired by the current findings for *future studies*.”

Response: We thank you for raising this issue. Accordingly, we have made efforts to simplify our language in order to make it more intuitive and easy to understand. Specifically, we have revised the *Discussion* section to provide a clearer understanding of how we discussed the key findings by explaining their significance, discussing their connection with existing literature, and proposing future research questions. For your convenience, below is a summary of the structure of our revised discussion section highlighting how we discussed our main findings in these three aspects.

Specifically, in the revised *Discussion*, we first *summarized the main findings* (*Discussion* Para. #1) and *proposed a neurodynamic model* of human empathy for pain (*Discussion* Para. #2, Fig. 6). We then discussed the key findings (*Discussion* Para. #3-9), *providing explanations for our key results*, discussing *how our findings are linked to the literature*, and/or *suggesting questions for future studies to further address*. Specifically,

Key finding 1 (presented in Fig. 2e-f, i-j; *Discussion* Para. #3): spectro-temporally shared and distinct profiles of the key empathy related regions ACC and AI. *These findings answered questions that remained unclear in previous fMRI studies of empathy*, i.e., whether the neural dynamics of the ACC and AI contribute to different aspects of empathic experiences. Moreover, *we specified alpha oscillatory suppression in the ACC and AI as the shared oscillatory feature of first-hand and vicarious experiences of physical pain*. On the other hand, *together with previous human and animal studies*, we suggested *distinct functional roles of beta oscillation in ACC (encoding pain intensity)*

and AI (encoding own unpleasantness).

Key finding 2 (presented in Fig. 2g, k; *Discussion* Para. #4): Our findings of decreased amygdala beta oscillations, and coupling with ACC/IFG during vicarious pain perception answered the question whether the amygdala engaged in the processing of human empathy and fill the gap between previous animal electrophysiological findings and human fMRI studies. We interpreted the late amygdala response and coupling with other regions as reflecting late, integrated neural representation of others' pain resulting from interactions with other regions. We also encouraged future research to directly test this possibility.

Key finding 3 (presented in Fig. 3, 4; *Discussion* Para. #5-7): We pointed out that our study provided initial electrophysiological understanding of empathy-related inter-regional communications and the functional organization of the empathy network. Specifically, in line with previous animal studies, we observed the engagement of the ACC-amygdala and AI-amygdala circuits. We also identified a new mode of inter-regional communication related to empathy, which provided new directions for future research about empathy-related circuits (*Discussion* Para. #5). Moreover, we interpreted our main findings of increased and decreased inter-regional communications as increased inter-regional information integration and reduced mutual distractions or functional specialization (*Discussion* Para. #6-7), taking the main findings of ACC-AI coupling (Fig. 3) and ACC-IFG coupling (Fig. 4) as specific examples.

In addition, we also discussed how we minimize potential influences to enhance the reliability of our findings (*Discussion* Para. #8). We pointed out potential limitations and proposed research questions that could be addressed by future studies (*Discussion* Para. #9-10).

Please allow us to highlight that, although not exhaustive of all reported results, we have provided comprehensive discussions on key findings that might be of particular interest in the field. We would be happy to expand on any specific points recommended by the reviewer for further improvement.

References

- Elliott, M. L., Knodt, A. R., Ireland, D., Morris, M. L., Poulton, R., Ramrakha, S., ... & Hariri, A. R. (2020). What is the test-retest reliability of common task-functional MRI measures? New empirical evidence and a meta-analysis. *Psychological science*, *31*(7), 792-806.
- Gulli, R. A., Duong, L. R., Corrigan, B. W., Doucet, G., Williams, S., Fusi, S., & Martinez-Trujillo, J. C. (2020). Context-dependent representations of objects and space in the primate hippocampus during virtual navigation. *Nature neuroscience*, *23*(1), 103-112.
- Kriegeskorte, N., Simmons, W. K., Bellgowan, P. S., & Baker, C. I. (2009). Circular analysis in systems neuroscience: the dangers of double dipping. *Nature neuroscience*, *12*(5), 535-540.
- Lavrakas, P. J. (2008). *Encyclopedia of survey research methods*. Sage publications.
- Muller, L., Reynaud, A., Chavane, F., & Destexhe, A. (2014). The stimulus-evoked population response in visual cortex of awake monkey is a propagating wave. *Nature communications*, *5*(1), 3675.
- Noble, S., Scheinost, D., & Constable, R. T. (2021). A guide to the measurement and interpretation of fMRI test-retest reliability. *Current Opinion in Behavioral Sciences*, *40*, 27-32.
- Pereira, F., Mitchell, T., & Botvinick, M. (2009). Machine learning classifiers and fMRI: a tutorial overview. *Neuroimage*, *45*(1), S199-S209.
- Proix, T., Delgado Saa, J., Christen, A., Martin, S., Pasley, B. N., Knight, R. T., ... & Giraud, A. L. (2022). Imagined speech can be decoded from low-and cross-frequency intracranial EEG features. *Nature communications*, *13*(1), 48.
- Ter Wal, M., Linde-Domingo, J., Lifanov, J., Roux, F., Kolibius, L. D., Gollwitzer, S., ... & Wimber, M. (2021). Theta rhythmicity governs human behavior and hippocampal signals during memory-dependent tasks. *Nature Communications*, *12*(1), 7048.
- Varoquaux, G., Raamana, P. R., Engemann, D. A., Hoyos-Idrobo, A., Schwartz, Y., & Thirion, B. (2017). Assessing and tuning brain decoders: cross-validation, caveats, and guidelines. *NeuroImage*, *145*, 166-179.
- Wang, S., Falcone, R., Richmond, B., & Averbeck, B. B. (2023). Attractor dynamics reflect decision confidence in macaque prefrontal cortex. *Nature Neuroscience*, *26*(11), 1970-1980.